# Structural variation in 1,019 diverse humans based on long-read sequencing

Siegfried Schloissnig[1,19], Samarendra Pani[2,3,19], Jana Ebler[2,3], Carsten Hain[4], Vasiliki Tsapalou[4], Arda Söylev[2,3], Patrick Hüther[1,5], Hufsah Ashraf[2,3], Timofey Prodanov[2,3], Mila Asparuhova[1,6], Hugo Magalhães[2,3], Wolfram Höps[7], Jesus Emiliano Sotelo-Fonseca[8,9], Tomas Fitzgerald[10], Walter Santana-Garcia[10], Ricardo Moreira-Pinhal[11,12], Sarah Hunt[10], Francy J. Pérez-Llanos[13,14], Tassilo Erik Wollenweber[13], Sugirthan Sivalingam[15], Dagmar Wieczorek[15], Mario Cáceres[11,12,16], Christian Gilissen[7], Ewan Birney[10], Zhihao Ding[17], Jan Nygaard Jensen[17], Nikhil Podduturi[17], Jan Stutzki[18], Bernardo Rodriguez-Martin[4,8,9 ✉], Tobias Rausch[4 ✉], Tobias Marschall[2,3 ✉] & Jan O. Korbel[4,10 ✉]

Genomic structural variants (SVs) contribute substantially to genetic diversity and human diseases[1–4], yet remain under-characterized in population-scale cohorts[5]. Here we conducted long-read sequencing[6] in 1,019 humans to construct an intermediate-coverage resource covering 26 populations from the 1000 Genomes Project. Integrating linear and graph genome-based analyses, we uncover over 100,000 sequence-resolved biallelic SVs and we genotype 300,000 multiallelic variable number of tandem repeats[7], advancing SV characterization over short-read-based population-scale surveys[3,4]. We characterize deletions, duplications, insertions and inversions in distinct populations. Long interspersed nuclear element-1 (L1) and SINE-VNTR-Alu (SVA) retrotransposition activities mediate the transduction[8,9] of unique sequence stretches in 5′ or 3′, depending on source mobile element class and locus. SV breakpoint analyses point to a spectrum of homology-mediated processes contributing to SV formation and recurrent deletion events. Our open-access resource underscores the value of long-read sequencing in advancing SV characterization and enables guiding variant prioritization in patient genomes.

SVs make up most polymorphic base pairs in the genome[4], and are causally implicated in numerous common and rare diseases[1,2]. A subset of SVs exhibit strong population stratification[10,11]. Recently, the Human Pangenome Reference Consortium (HPRC) released a draft pangenome from 44 diploid long-read sequencing (LRS) assemblies generated from multiple genomic platforms, and demonstrated how this graph-based reference enhances SV discovery[12]. Although LRS is increasingly used in disease research and diagnostics[13–16], population-scale datasets with comprehensive global representation remain limited[5]. Addressing this gap would be critical to facilitate widespread community access to rare and ancestry-specific genetic variation, especially for variants such as insertions, which are typically underrepresented in short-read sequencing datasets[17]. These data are expected to be instrumental to facilitate human diversity and disease research, including variant prioritization in genomic medicine.

Here we applied Oxford Nanopore Technologies (ONT) LRS to analyse SVs from the 1000 Genomes Project (1kGP) sample collection[18], which allows open and unrestricted public data access, data sharing and reuse. To allow inclusion of a wide diversity of haplotypes, we performed ONT-based genomic sequencing of over a thousand human samples to intermediate coverage. To benefit from recent advances in interpreting genetic variation in a pangenome[12], we established methods that leverage linear and graph-based pangenomic references for LRS-based SV discovery, and engineered a computational framework using graph augmentation for SV genotyping. Our resulting pangenomic dataset constitutes a comprehensive collection of DNA sequence-resolved SV alleles, encompassing a variant frequency spectrum from common to rare across 26 diverse human populations. This resource provides a global reference of genetic variation using LRS technology—facilitating studies of SV biology and human disease, and yielding insight into allelic architecture, mechanistic

[1]Research Institute of Molecular Pathology (IMP), Vienna BioCenter (VBC), Vienna, Austria. [2]Institute for Medical Biometry and Bioinformatics, Medical Faculty and University Hospital Düsseldorf, Heinrich Heine University Düsseldorf, Düsseldorf, Germany. [3]Center for Digital Medicine, Heinrich Heine University Düsseldorf, Düsseldorf, Germany. [4]European Molecular Biology Laboratory (EMBL), Genome Biology Unit, Heidelberg, Germany. [5]Institute of Molecular Biology (IMB), Mainz, Germany. [6]Institute of Molecular Biotechnology of the Austrian Academy of Sciences (IMBA), Vienna BioCenter (VBC), Vienna, Austria. [7]Department of Human Genetics, Radboud University Medical Center, Nijmegen, The Netherlands. [8]Centre for Genomic Regulation (CRG), The Barcelona Institute of Science and Technology, Barcelona, Spain. [9]Universitat Pompeu Fabra (UPF), Barcelona, Spain. [10]European Molecular Biology Laboratory, European Bioinformatics Institute, Cambridge, UK. [11]Institut de Biotecnologia i de Biomedicina, Universitat Autònoma de Barcelona, Bellaterra, Spain. [12]Research Programme on Biomedical Informatics (GRIB), Hospital del Mar Research Institute, Barcelona, Spain. [13]Biological and Medical Research Center (BMFZ), Medical Faculty and University Hospital Düsseldorf, Heinrich Heine University Düsseldorf, Düsseldorf, Germany. [14]West German Genome Center (WGGC), Medical Faculty and University Hospital Düsseldorf, Heinrich Heine University Düsseldorf, Düsseldorf, Germany. [15]Institute of Human Genetics, Medical Faculty and University Hospital Düsseldorf, Heinrich Heine University Düsseldorf, Düsseldorf, Germany. [16]ICREA, Barcelona, Spain. [17]Global Computational Biology and Digital Sciences (gCBDS), Boehringer Ingelheim Pharma GmbH & Co. KG, Biberach an der Riss, Germany. [18]BI X GmbH, Ingelheim am Rhein, Germany. [19]These authors contributed equally: Siegfried Schloissnig, Samarendra Pani. ✉e-mail: bernardo.rodriguez@crg.eu; tobias.rausch@embl.de; tobias.marschall@hhu.de; jan.korbel@embl.de

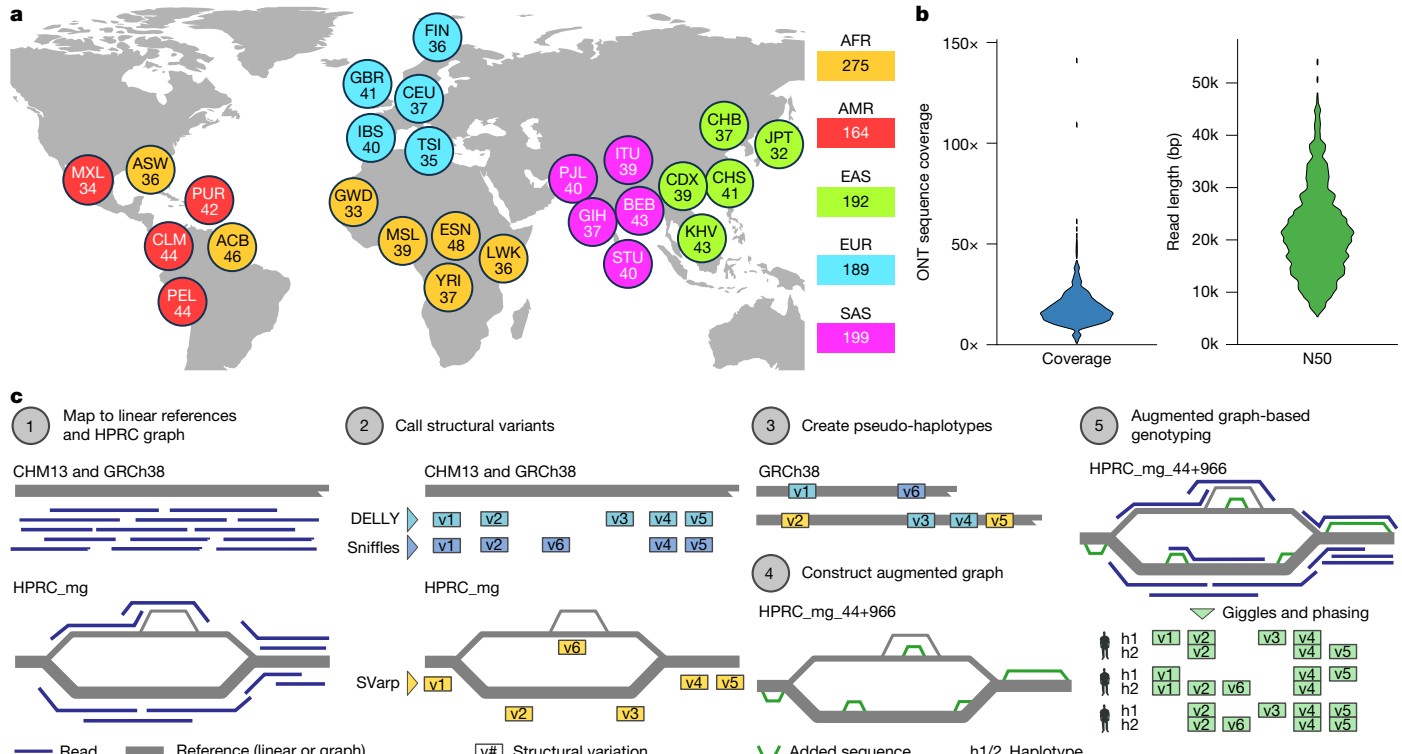

**Fig. 1 | LRS and SAGA. a**, Breakdown of self-identified geographical ancestries for 1,019 long-read genomes representing 26 geographies (that is, populations) from 5 continental regions. The three-letter codes used are equivalent to those used in the 1kGP phase III[18] and are resolved in Supplementary Table 2. **b**, ONT sequence coverage per sample, expressed as fold-coverage (left), and N50 read length in base pairs (right). **c**, Schematic of the SAGA framework for graph-aware discovery and genotyping of SVs using a pangenome graph augmentation approach. Basemap in **a** from Natural Earth data (https://www.naturalearthdata.com).

origin, mutational recurrence and population distribution of SV classes.

## LRS and graph-based SV discovery

We selected samples from the 1kGP collection, subjecting size-selected more than or equal to 25-kilobase pair (kb) DNA fragments to LRS (Supplementary Fig. 1 and Supplementary Table 1). After quality filtering (Methods), our study cohort comprises 1,019 genomes (Fig. 1a), sequenced to a median coverage of 16.9× with a median N50 read length of 20.3 kb (Fig. 1b and Supplementary Fig. 2). These samples are from 26 self-reported population groups and span five continental areas, including 189 from European (EUR) donors, 192 from East Asia (EAS), 199 from South Asia (SAS), 275 from Africa (AFR) and 164 from the Americas (AMR).

To allow for graph-aware SV discovery and genotyping in these data, we devised the SV analysis by graph augmentation (SAGA) framework (Fig. 1c). Outlined below, SAGA integrates read mapping to both linear and graph references, followed by graph-aware SV discovery and genotyping at population scale.

## Long-read alignment

In the initial step of SAGA, we performed read alignment against both linear (GRCh38, and the Telomere-to-Telomere (T2T) reference, denoted CHM13) and graph (the minigraph HPRC reference graph, denoted HPRC_mg) genomic references (Methods). Comparative analyses of the alignments show that HPRC_mg and CHM13 outperform GRCh38 (with average mapping identities increased by more than 0.5%), whereas the size distribution of alignment gaps indicates that HPRC_mg comprises a more comprehensive collection of mobile element insertions (MEIs) and deletions (Supplementary Fig. 3). Using CHM13, we find that on average 93.6% of the genome exhibits a coverage of 5× or more per sample (Supplementary Fig. 4). Furthermore, using WhatsHap (Methods) we haplotype-phased single nucleotide polymorphisms (SNPs) using our ONT dataset, and find that these show excellent concordance with phased SNPs from a short-read-based 1kGP study[3], as evidenced by median switch error rates of only 0.69% in children from the six parent–offspring trios included in our dataset, and 1.32% for unrelated (and parental) samples (Supplementary Fig. 5 and Supplementary Note 1).

## SV discovery

To allow for comprehensive SV discovery, we used LRS-based SV callers tailored to linear reference genomes. We used Sniffles and DELLY (Methods), applying both to GRCh38 and CHM13. Callset integration across both linear references yielded an average number of 15,301 and 21,529 SVs per sample for Sniffles and DELLY, respectively.

To complement these classical SV discovery methods, we additionally harnessed the graph-aware SVarp algorithm (Methods), which allows SV discovery in haplotype contexts represented in HPRC_mg yet absent from a linear reference. SVarp searches for SV patterns from graph-aligned reads, and then performs local long-read assembly to reconstruct 'SV sequence contigs' (svtigs). To maximize the accuracy of svtig assembly, we applied SVarp to a subset of our sample set (*n* = 967 genomes) previously sequenced at high coverage with short reads[3]. This allowed efficient haplotype assignment ('haplo-tagging') of 69.9% of the ONT reads using phased SNPs. Using this haplo-tagged read information, SVarp constructed on average 1,145 svtigs per genome not previously contained in the HPRC_mg graph (Supplementary Fig. 6).

## Graph augmentation

To enable non-redundant SV callset integration across all SV callers, we augmented the HPRC_mg graph by incorporating further bubbles[12]

representing new SV allelic sequences. To achieve this, SAGA encompasses a pseudo-haplotype construction and a graph augmentation step (Fig. 1c). Pseudo-haplotype construction produces chromosome-wide, haplotype-like sequences that incorporate the discovered SV alleles in a non-overlapping manner (Methods). Graph augmentation then uses the minigraph tool (Methods) to integrate these pseudo-haplotypes into the graph. Our study augments the HPRC_mg graph (originally representing 44 samples) with SVs from all 967 LRS samples considered for comprehensive SV discovery. This process yields 'HPRC_mg_44+966', a pangenome representing SVs from 1,010 individuals (considering one sample in our resource (HG01258) is part of the HPRC_mg graph already; Extended Data Fig. 1 and Supplementary Fig. 7). HPRC_mg_44+966 comprises 220,168 bubbles altogether, compared with 102,371 bubbles present in HPRC_mg (Supplementary Figs. 8 and 9).

Of the 117,797 new bubbles represented in HPRC_mg_44+966, 105,744 (90%) are at least 1 kb away from the nearest bubble in the original HPRC_mg graph. These are presumed to reflect SVs not previously represented in HPRC_mg (Supplementary Fig. 10). To evaluate the quality of HPRC_mg_44+966, we mapped ONT reads from the HG00513 sample onto this graph. We observe improved alignment metrics, with a gain of 33,208 aligned reads and a further 152.5 megabases (Mb) of aligned bases compared with alignment onto HPRC_mg (Supplementary Table 3), suggesting the augmented graph provides enhanced SV analysis capabilities.

### SV genotyping and phasing

Unified SV genotypes are a prerequisite for relevant downstream analyses with population-scale variant datasets, including population genetic and disease studies. As the final step of SAGA, we therefore use Giggles, a genotyping tool that harnesses graph-aligned long reads for SV genotyping (Methods). We genotyped the set of 967 samples of HPRC_mg_44+966 with available LRS data, yielding genotypes for 167,291 primary SV sites.

Consistent haplotype phasing elevates the value of SV resources, facilitating allele-specific analyses and investigation of haplotype structures. We used a recently generated CHM13 haplotype reference panel[19] constructed from short-read SNP calls to carry out statistical SV phasing using SHAPEIT5 (Methods). We find that 164,571 (98.4%) of the genotyped SV sites are successfully phased. These comprise our final SAGA-based SV callset, and include 65,075 deletions, 74,125 insertions and 25,371 'putatively complex' sites, for which both the reference and alternative allele are larger than 1 base pair (bp) (Supplementary Note 2). On the basis of analysing the respective graph bubble structures (Methods), 107,005 SVs in the phased SV callset are classified as biallelic, and the remaining ($n$ = 57,566) multiallelic.

### Resource quality assessment

Comparison of this SV callset with SVs discovered from multi-platform genome assemblies recently constructed by the Human Genome Structural Variation Consortium (HGSVC)[20] suggests a genome-wide false discovery rate (FDR) of 15.55% for deletions and 15.89% for insertions. The FDR varies by SV size: SVs ≥ 250 bp show considerably lower FDR (deletions: 6.91%, insertions: 8.12%) than SVs < 250 bp (deletions: 19.14%, insertions: 19.57%; Extended Data Fig. 2 and Supplementary Figs. 11 and 12). The smallest SVs largely comprise tandem repeat variation, whose divergent representation between graph- and assembly-based callsets can complicate both SV discovery and comparative analysis[21]. MEIs—an SV class exhibiting well-defined allele architectures—exhibit a particularly low FDR (0.85–6.75%; Supplementary Fig. 13), whereas mobile element deletions show 1.94% FDR (Supplementary Note 3). Comparison with multi-platform assemblies[20] also yielded autosomal sensitivity estimates, which vary by SV class, allele frequency and size (Extended Data Fig. 2 and Supplementary Figs. 14 and 15). Sensitivity is particularly high for MEIs (ranging from 84.3% to 90.6%). The genome-wide true positive rate is 64.36% for deletions and 67.33%

for insertions after genotyping (Extended Data Fig. 2 and Methods). Reflecting our intermediate-coverage study design, more common SVs exhibit an improved true positive rate over rarer SVs, with 79.59% for deletions and 83.24% for insertions seen for minor allele frequency (MAF) ≥ 0.1.

We additionally examined the primary SV genotypes. SV allele frequencies after phasing are in excellent agreement with the primary SV genotypes (Supplementary Figs. 16 and 17). We used data from the six parent–offspring trios to identify 'Mendelian inconsistencies', which could derive from de novo SV formation events or genotyping errors. The average rate of such inconsistencies for biallelic SVs is 3.87% for deletions, 4.44% for insertions and 4.10% for putatively complex sites, implying high genotype accuracy (Supplementary Tables 4–12). For multiallelic sites the average Mendelian inconsistency is 15.1%, reflecting previously reported challenges in genotyping such variant sites[4,18]. Our estimates align with strong performance against Hardy–Weinberg equilibrium checks (Extended Data Fig. 3 and Supplementary Figs. 18 and 19) and genotypes are largely robust to coverage and read length variations (Supplementary Note 4). Moreover, our SV sites intersect 69.5% of insertions and 64.9% of deletions called from short-read sequencing in the same samples[3] (Supplementary Fig. 20). At these intersecting sites, the genotype concordance is 98.7% for deletions (non-reference genotype concordance: 77.6%) and 96.8% for insertions (non-reference concordance: 79.0%) compared with Illumina-based genotypes[3], in support of high genotype accuracy.

## The SV landscape across 26 ancestries

We analysed the degree to which this SV dataset grows cumulatively after each new genome is added (Fig. 2a and Supplementary Fig. 9). We observe pronounced saturation effects[22] for common SVs when adding new samples to the pangenome graph, with AFR samples disproportionally increasing SV yield consistent with higher levels of genetic variation in these populations[18,22].

When compared with previous population-scale genome sequencing studies in 1kGP samples[3,4], our resource captures more variants across the SV size spectrum, with SVs ranging from 50 to several hundred base pairs poorly captured using short reads[17]. The median SV count per sample is 23,969 for AFR samples (19,297 for other ancestries, non-AFR; Supplementary Note 4), compared with 9,963 and 8,540 for AFR and non-AFR, respectively, detected through short reads when subsetting to the same samples[3]. The proportionally largest gain is seen for insertions, which our resource captures with increased sensitivity across insertion sizes (Fig. 2b and Supplementary Fig. 21). By comparison, large deletions are detected with increased abundance using short reads[17] (Fig. 2b). Furthermore, facilitated by the read length in our resource, we observe a more than tenfold increase in insertion sites with sequenced-resolved SV alleles (Fig. 2b). For deletions, a considerable fraction of which can be resolved through split-read analysis using short reads[3,4,23], our resource increases the nucleotide-resolved SV count by 40% (from 46,895 to 65,812). These analysis results are consistent with the improved characteristics of LRS in resolving SVs[17,22,24].

### Rare SVs from diverse ancestries

Analysing the allele frequency spectrum from our augmented pangenome shows a broad spectrum of SV alleles, from common to rare (Fig. 2c, Extended Data Fig. 3 and Supplementary Fig. 22). We observe a considerable enhancement in capturing SVs with an allele frequency ≤ 2% over HPRC_mg, with the vast majority of rare SVs stemming from samples newly incorporated into HPRC_mg_44+966 (Supplementary Fig. 23). Across our resource, most SVs are rare (59.3% have an MAF < 1%), and, with the current sample size, are typically detected in a single continental cohort (AFR, AMR, EAS, EUR or SAS). By comparison, at an allele frequency ≥ 2.5%, most SVs are seen in at least two continental groups (Extended Data Fig. 3). Multiallelic SV loci generally

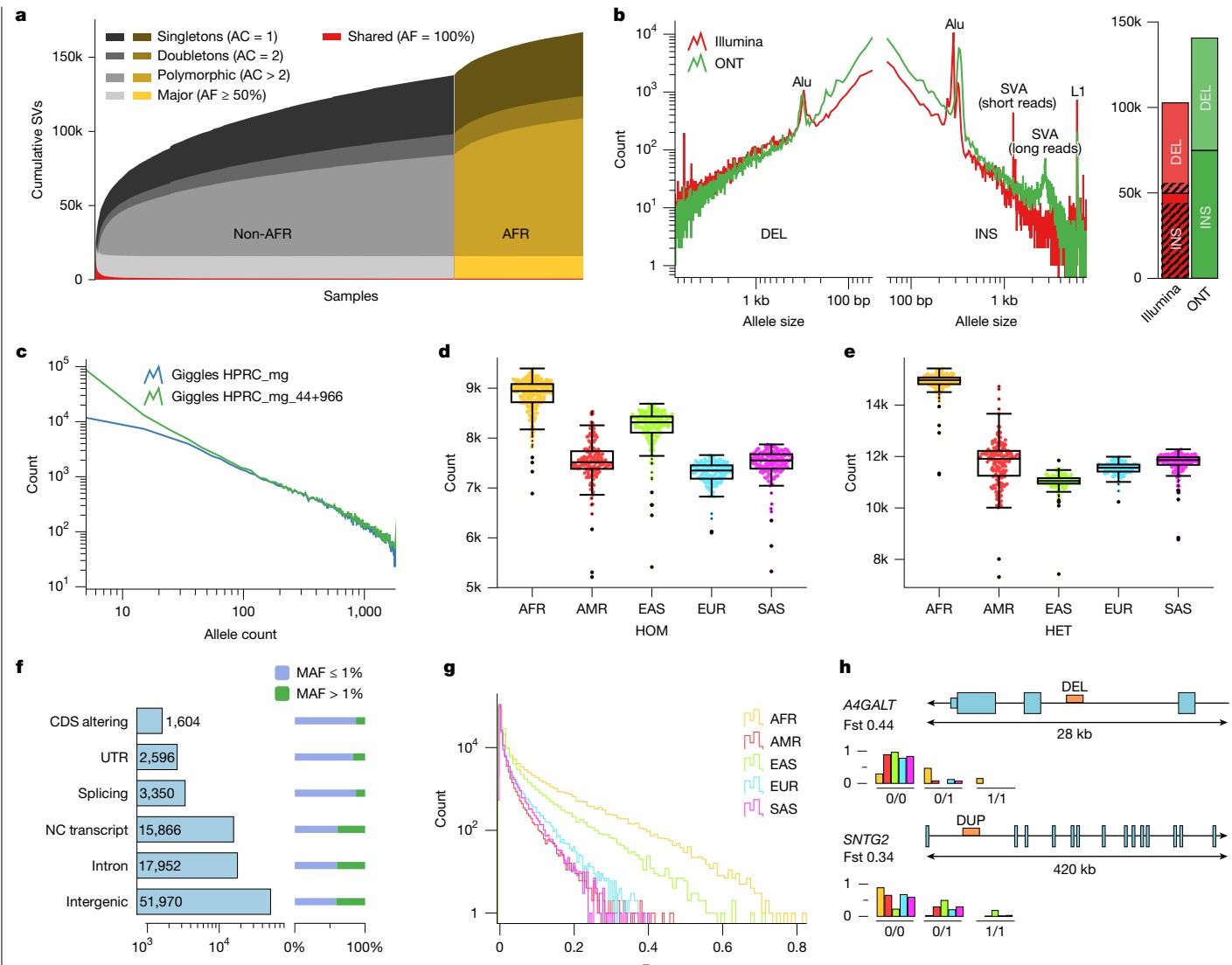

**Fig. 2 | Callset properties and SV landscape in different geographical ancestries. a**, Cumulative number of unique SVs from the SAGA framework when adding individual long-read genomes, from left to right. The rate of SV discovery slows with each new sample added. Colours denote singletons, doubletons, SVs seen with an allele count of more than 2 (polymorphic), as well as major and shared alleles. **b**, Left, SV length distributions in population-scale 1kGP SV callsets, versus the cumulative count of SV sites, with a comparison of ONT sequencing (ONT; *n* = 967 samples) with the short-read-based analysis of the 1kGP cohort[3] (Illumina) subsetted to the same (*n* = 967) samples. Right, the 1kGP ONT callset is outnumbering short-read-based SV calls both for deletions (DEL, upper sections) and insertions (INS, lower sections). SVs previously unresolved by their sequence, using short reads[3], are depicted by shaded areas. Notably, LRS provides exact MEI lengths, in contrast to the length approximations reported in short-read-based studies[3] (Supplementary Fig. 29). **c**, SV allele count

(*x* axis) relative to the count of SV sites (*y* axis), constructed by genotyping the original HPRC graph (HPRC_mg) and the augmented graph (HPRC_mg_44+966) using Giggles. Consistent with the considerably smaller panel size used, HPRC_mg under-represents rare alleles. **d,e**, Homozygous (HOM) SV count (**d**) and heterozygous (HET) SV count (**e**) per sample stratified into self-identified geographies (*n* = 967; whiskers extend to points that are within 1.5 × interquartile range from the upper or the lower quartiles). **f**, Impact of SVs on distinct genomic features. SVs affecting sequences of genes occur at lower MAF compared with non-coding regions. **g**, Frequency polygons of SV Fst values per continental population (histograms stratifying Fst for bi- and multiallelic are in Supplementary Fig. 28). **h**, Deletion (DEL) and duplication (DUP) in the intragenic space of two medically relevant genes exhibiting differentiation in AFR and EAS samples, respectively. AC, allele count; AF, allele frequency; CDS, coding sequence; NC, non-coding; UTR, untranslated region.

have a higher propensity to be shared across continents than biallelic SVs, with most multiallelic SVs with allele frequency ≥1.5% being shared (Supplementary Fig. 24), which is potentially explained by recurrent rearrangements at these loci. Furthermore, the relative increase in SVs in AFR samples is more pronounced for heterozygous than for homozygous SVs, reflecting the greater genetic diversity in AFR samples[22] (Fig. 2a,d,e and Supplementary Fig. 25).

## Population characteristics by SV class

Unlike previous 1kGP studies that have placed a focus on characterizing sequence-resolved deletions[3,4], we find comparable population

characteristics between SVs primarily identified as deletions and insertions (Supplementary Fig. 19). Measuring the degree to which SVs are in linkage disequilibrium with nearby SNPs, we find that 62.6% of deletions and 62.9% of insertions with at least 1% MAF are in linkage disequilibrium with nearby SNPs (*r*² ≥ 0.5) (Extended Data Fig. 3 and Supplementary Fig. 26). These metrics increase to 89.8% (deletions) and 91.7% (insertions), respectively, in Genome-in-a-Bottle (GiaB) 'high-confidence regions' (Methods), which are more depleted of repeats and thus less likely subject to recurrent SV or gene conversion events[25]. Both larger insertions and deletions are rarer in the population than smaller SVs (Supplementary Fig. 27), possibly explained by stronger negative

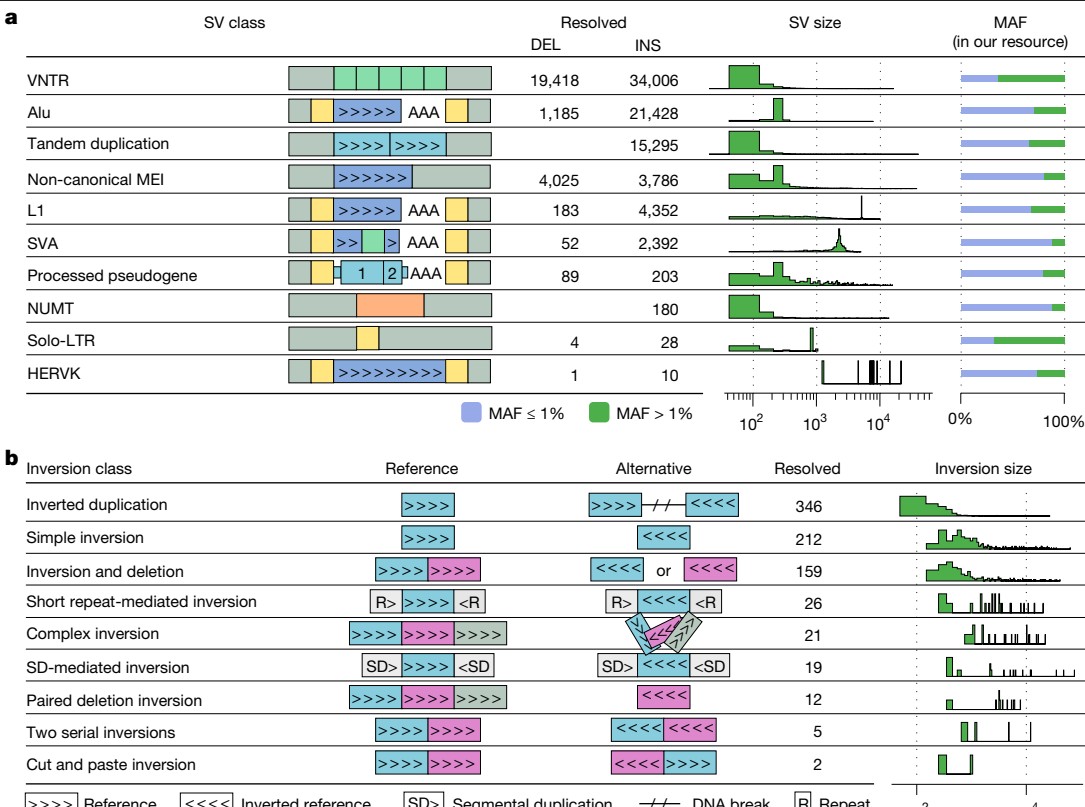

**Fig. 3 | Prevalence of distinct SV classes in our SAGA-based data resource.**
**a**, VNTRs (here shown as identified with the SAGA framework), duplications and insertions, as classified by SVAN. The SV class is denoted by label and ideogram, and the number of resolved class members is given for insertions (INS; insertion relative to the reference) and deletions (DEL; deletion relative to reference), in addition to size distribution and percentage of low- and high-frequency alleles. Repeat units for VNTRs and internal VNTR sequence for SVA insertions are depicted using green boxes. TSDs flanking retrotransposition insertion events and LTRs flanking HERVK are represented as yellow boxes. Numbers for interspersed and complex duplications are shown in Supplementary Table 17. Alu, L1 and SVA refer to canonical retrotransposition events. **b**, Inversion classes identified (twin priming events excluded from this display). For each class, the reference structure is contrasted with the alternative allele, the number of resolved inversions is shown and the inversion size distribution is shown (in bp).

selection against variants affecting functional sequence. Consistent with this notion, SVs affecting functional elements show significantly reduced MAF ($P < 7.4 \times 10^{-235}$; Kolmogorov–Smirnov test; Fig. 2f).

## Geographical stratification of SVs

A principal component analysis of SV calls, as well as an SV-based admixture analysis, show groupings of samples corresponding well with donor ancestry[4] (Extended Data Fig. 3). Using fixation indices (Fst) to quantify SV differentiation by continent, the allelic diversity of AFR (followed by EAS) is clearly observable (Fig. 2g and Supplementary Fig. 28). We find evidence for differentiation (Fst > 0.2) for 8,597 SVs (Supplementary Table 13), of which 105 encompass a GiaB-classified medically relevant gene[26], and thus are of potential interest for disease studies (Supplementary Table 14). Examples are a deletion and duplication affecting the intragenic regions of *A4GALT* (MAF 14%) and *SNTG2* (MAF 18%), enriched in AFR and EAS, respectively (Fig. 2h). We further note a complex SV near *LAMB1* exhibiting enrichment in AMR, and particularly among samples with self-identified ancestry of Peruvians in Lima (PEL) (Supplementary Fig. 29). Additionally, we observe a strong correlation between 1-Mb-window averaged SV-based and SNP-based Fst values (Pearson's $P < 4.0 \times 10^{-16}$), deviations of which potentially serve as indicators for genomic areas with SV-driven differentiation. Requiring a confidence level of more than 5 s.d. ($P < 0.6 \times 10^{-6}$), we find 11 regions in which the differentiation is likely to be SV-driven (Supplementary Table 15), including a deleted region near gene *ARHGAP24* with the highest MAF of 27.3% (Fst = 0.31) seen in the Mende in Sierra Leone (MSL).

## Resolved spectrum of SV classes

We devised the SV annotator (SVAN), an algorithm that leverages allelic representations and genomic annotations to classify SVs into distinct classes (Methods). SVAN classifies 72,346 insertions (96.0%) of our pangenomic resource (Fig. 3a and Extended Data Fig. 3), offering a detailed perspective of this variant class. SVAN additionally classifies 21,301 (32.2%) deletions, and 14,681 (57.1%) of all putatively complex sites (Fig. 3a), the latter of which resolve into 11,030 further insertions and 3,651 further deletions.

## Duplications and variable number tandem repeats

In total, 17,029 (19.5%) of the insertions in our pangenomic resource are duplications, of which most (15,295, 89.8%) represent tandem duplications (Fig. 3). Additionally, 34,006 (39.0%) insertions and 19,418 (26.2%) deletions are classified as variable number tandem repeats (VNTRs), an SV class relevant to human diversity and disease[27] (Fig. 3a). Unlike for other SV classes, most VNTRs (50.4%) in our SAGA-based SV callset represent multiallelic SV sites, which poses a challenge to graph construction tools, as finding the correct alignment can be difficult if several SV alleles at the same locus differ in their representation by only a few base pairs[21]. Therefore, to allow us to systematically capture VNTR complexity, we additionally genotyped VNTR sites across our resource using the vamos tool (Methods). A detailed analysis of these genotypes shows 739 biallelic and 369,685 multiallelic VNTRs, considerably surpassing estimates derived from

graph-based analysis, and thus providing a refined estimate of allelic VNTR diversity. In-depth comparison against multi-platform whole-genome assemblies[20] shows a strong concordance in polymorphic repeat unit counts, supporting the robustness of the vamos-based VNTR genotypes (Supplementary Fig. 30). Unlike genome assemblies at present limited to small sample sizes, our resource enables population-scale analysis of extreme repeat unit counts, illustrated by the 1% and 99% percentiles of the repeat unit distribution (Extended Data Fig. 4). The utility of these genotypes is exemplified by our analysis of repeat unit variation at the *PLIN4* and *ABCA7* loci implicated in late-onset diseases (Extended Data Fig. 4, Supplementary Table 16 and Supplementary Note 5).

### Mobile elements and nuclear mitochondrial DNA segments

We next focused on the distinct classes of insertions classified by SVAN, most of which are the product of the activity of mobile elements (Fig. 3a). These include 31,302 non-reference MEIs, including 23,212 Alu, 4,851 L1 and 3,239 SVA insertions (Extended Data Fig. 5 and Supplementary Table 18), which reflects an increase of 20% (for Alu), 166% (L1) and 179% (SVA), respectively, over short-read-based analysis in the same samples[3]. We further identify 3,813 reference MEIs primarily called as deletions (classified into 3,122 Alu, 460 L1 and 231 SVA events, respectively). Using SVAN, we classify MEIs as canonical (84.3%; 29,592) and non-canonical (15.73%; 5,523). Thereby, canonical MEIs show a conformation consistent with target-primed reverse transcription[28] or twin priming[29], whereas non-canonical ones are 3′ truncated (34.4%; 1,901), lack poly(A) tails (66.4%; 3,665) or have internal rearrangements (52.8%; 2,916). Non-canonical MEIs may have arisen through alternative integration mechanisms[30–32], or be the product of SVs occurring after the integration event. Canonical MEIs typically show distinctive hallmarks of retrotransposition[33], including target site duplications (TSDs, median: 21 bp) and poly(A) tracts (median: 30 bp) seen for 98.7% and 100% of the events (Extended Data Fig. 5), respectively. Consistent with previous studies[22,34–36], most (72.5%; 3,288) of the canonical L1 insertions represent 'dead-on-arrival' copies owing to either truncation or inversion at their 5′ ends (Extended Data Fig. 5). This contrasts with canonical Alu and SVA for which 74.4% and 72.9% of the insertions, respectively, are full-length. We find that full-length SVAs show a broad size distribution, ranging from 1 to 6 kb, largely owing to the extensive variability observed in their hexameric repeat and VNTR regions (Extended Data Fig. 5).

We also identify 203 non-reference processed pseudogenes, as well as 89 reference polymorphic processed pseudogenes[37,38] (Extended Data Fig. 5). Most of these (73.9%; 216) are monoexonic, with the remaining pseudogenes comprising between 2 and 21 exons. We further identify evidence for human endogenous retrovirus (HERV) activity, with 10 insertions classified as HERVK and 28 solo-long terminal repeats (LTRs) (Extended Data Fig. 6). Manual inspection of these events shows that 82% (23 of 28) of solo-LTRs correspond to complete LTR5_Hs events. With respect to HERVK, five of the insertions encompass the proviral sequence flanked by two LTRs, four have a single LTR with the second residing on the human reference (CHM13) and one is heavily 5′-truncated. Comparisons with previous reports[35,39] show that although only around 22% (5 of 23) of the solo-LTRs in our study were not previously reported, most (70%; 7 of 10) of the HERVK insertions were not described previously.

Finally, SVAN classifies 180 insertions as nuclear mitochondrial DNA segments (NUMTs)[40]. The median length of these insertions is 126 bp, with 15 NUMTs larger than 1 kb, including a 14.8-kb insertion comprising 90% of the mitochondrial length identified in one carrier (Supplementary Fig. 31d).

### Inversions

Inversions pose considerable challenges for detection by sequencing owing to their balanced copy-number state, which limits their indirect identification through coverage depth analysis. Additionally, difficulties in sequence alignment, particularly arising from repetitive sequences flanking inversion breakpoints, further complicate their accurate ascertainment[36]. The notable N50 read length of this ONT dataset led us to explore its potential for inversion discovery, for which we noted that performing read re-alignment at regions exhibiting clustered mismatches enhances inversion detection accuracy (Supplementary Note 6 and Supplementary Figs. 32 and 33). We devised an inversion discovery workflow leveraging these insights (Methods), identifying 491 inversions which were confirmed by semi-manual (dotplot-based) analysis (Supplementary Fig. 34). We added to these inversions further SVs that were primarily defined as insertions, yet reclassified by SVAN— including 311 inverted duplications and 1,047 inversions at the 5′-ends of L1 insertions probably resulting from twin priming[29] (Extended Data Fig. 5)—yielding 1,849 inversion calls overall (Fig. 3b). After omitting SVs previously genotyped with Giggles as well as SVs annotated as inverted duplications, the GeONTIpe tool (Methods) successfully determined genotypes for 78% of the inversions.

We next conducted detailed analyses of the set of 733 inversions outside MEI events, focusing on their size distribution, allelic structure and complexity (Methods). Inverted duplications represent the most common class, with 277 events detected showing a median length of 284 bp (Fig. 3b and Supplementary Fig. 35). Additionally, we find 257 balanced (or 'simple') inversions with a median length of 1,565 bp, which are further broken down into 45 inversions bordered by various repeat classes in inverted orientation, implying formation through homology-directed repair (HDR) processes or non-allelic homologous recombination (NAHR)[41].

We further find an abundance of complex inversions with multiple breakpoints[4,36] (Fig. 3b). These include 159 inversions with an adjacent deletion, and 40 more complex structures, comprising five SVs characterized by two serial inversions; two instances of 'cut-and-paste' inversions with the locus structure characterized by an excised, inverted and inserted segment into a nearby location; 12 inversion sites exhibiting two flanking deletions; and 21 particularly complex inversions with a large number of breakpoints and in part incompletely resolved allelic structure. These events are likely to originate from a DNA replication-associated process, such as microhomology-mediated break-induced replication[42].

### Polymorphic L1 and SVA transductions

MEI transductions represent an SV generation process in which a full-length mobile element, denoted as source or progenitor locus, mobilizes flanking DNA sequences during retrotransposition[43]. This process can result in the integration of a mobile element and a companion non-repetitive sequence situated in 5′ or 3′ relative to the progenitor copy[8,9]. Harnessing our pangenomic resource, we analysed polymorphic transduction events among all sequence-resolved MEI sequences. We annotate 878 transductions, with 466 (8.8%) of the 5,311 L1 insertions and 412 (11.9%) of the 3,470 SVA insertions exhibiting a transduction (Supplementary Tables 19 and 20). In addition to these transduction events containing a companion mobile element segment (denoted 'partnered transductions'), a further set of 48 transductions are truncated resulting in the integration of the transduced sequence alone ('orphan transductions').

Consistent with previous studies[8,9,22], we find that the relative proportion of 5′ or 3′ partnered transductions differs between MEI families. Most (82.2%; 350) L1-mediated transductions are in 3′, whereas SVAs generate 5′ and 3′ transductions at similar ratios, with 218 (53.7%) detected in 5′ and 188 (46.3%) in 3′ (Supplementary Table 20). Although SVA-mediated 3′ transductions are significantly longer in size than 5′ transductions ($P = 7.2 \times 10^{-6}$; two-tailed Mann–Whitney $U$ test), we observe no significant length difference for L1s (Extended Data Fig. 5). Furthermore, SVA-mediated 3′ transductions are significantly longer

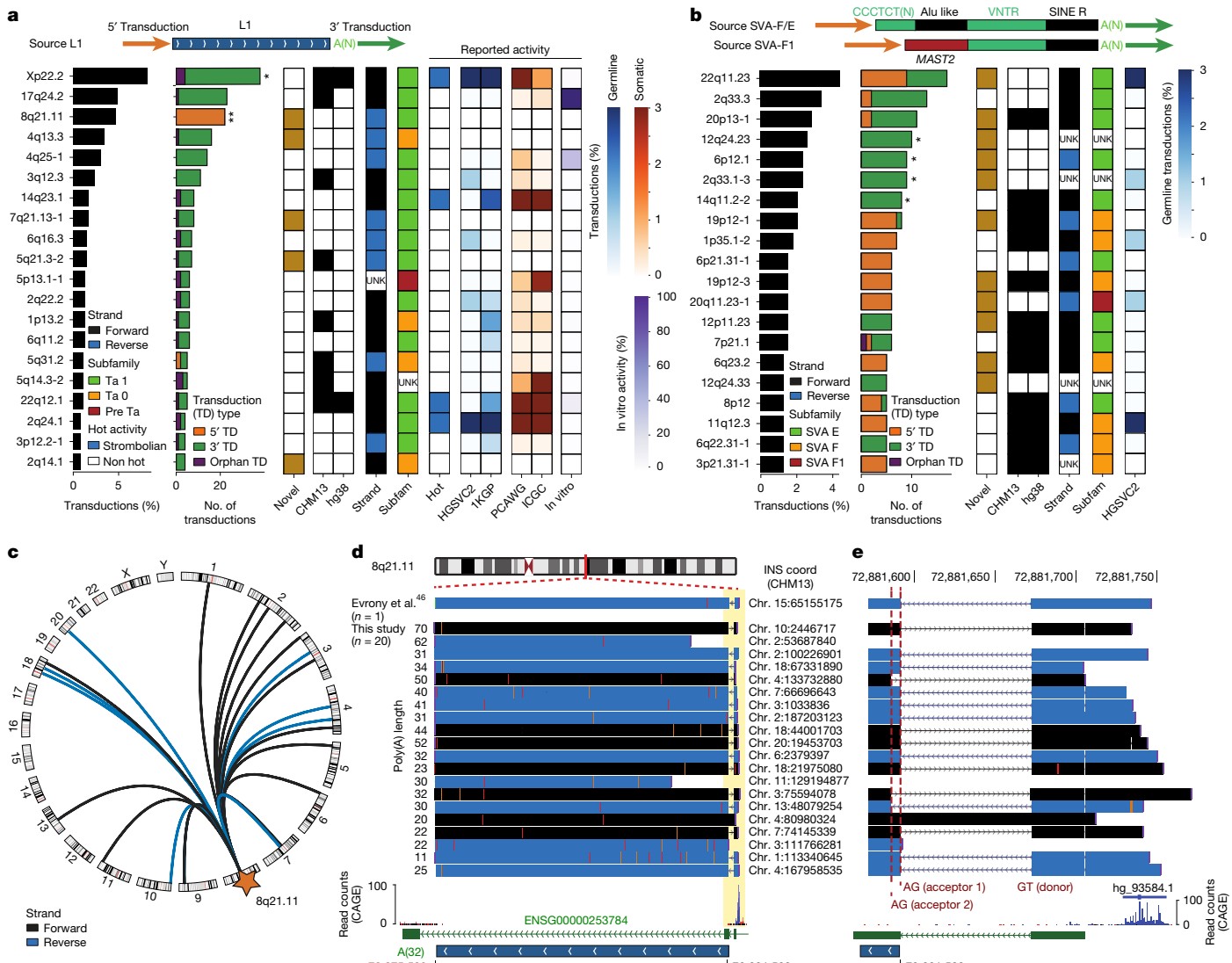

**Fig. 4 | Polymorphic landscape of L1 and SVA transductions. a,b,** Contribution relative to the total number of transductions (5′, 3′ and orphan) for the 20 most active L1 (**a**) and SVA (**b**) progenitors. Source elements are annotated by their presence/absence in the reference, orientation and subfamily. Source element novelty, hot activity status and previously reported activity estimates are based on in vitro assays and transduction tracing (Methods) and are shown as heat maps. Transduction 5′ and 3′ bias was assessed using a two-sided exact binomial test followed by multiple testing correction with Benjamini–Hochberg. Xp22.2 ($P_{adj} = 0.04$) and 8q21.11 ($P_{adj} = 3.9 \times 10^{-15}$) source L1s exhibit a significant 3′ and 5′ bias, respectively. All significant SVAs are 3′ biased, namely 12q24.23 ($P_{adj} = 0.01$), 6p12.1 ($P_{adj} = 0.02$), 2q33.1-3 ($P_{adj} = 0.02$) and 14q11.2-2 ($P_{adj} = 0.04$). Adjusted $P$ values for significantly biased loci are represented adjacent to each

bar as follows: *$P_{adj} < 0.05$; **$P_{adj} < 0.005$. **c,** Circos plot showing the integration positions for the 22 instances of 5′ transductions mediated by the 8q21.11 element. **d,** Alignment of inserts containing 5′ transductions at the source L1 region, including a single somatic transduction event reported in ref. 46 in the brain. Inserts are coloured according to whether they align in forward (black) or reverse (blue). Splicing between the full-length L1 and an upstream exon leading to 5′ transductions, highlighted in yellow. **e,** Magnification showing that the 5′ transductions initiate at a strong promoter located upstream, followed by canonical splicing between the first and second exon of *ENSG00000253784*, in addition to a second acceptor splice site within the L1 body. Transcription initiation is supported by an annotated transcription start site (hg_93584.1) and CAGE read counts[60].

compared with L1 ($P = 0.005$; Extended Data Fig. 5). This suggests the existence of family-specific determinants for 5′ and 3′ transductions.

We leveraged the transduced sequences as barcodes to identify their progenitor loci, showing that L1 transductions originate from a limited set of 208 L1 source elements. Among these, 20 highly active source L1s are responsible for 43.9% (205) of all L1 transductions identified (Fig. 4a), with most (14 of 20) of them belonging to the youngest Ta-1 subfamily. By comparison, we detect 176 source SVA elements, with 20 loci alone mediating 36.9% (152) of all SVA transductions (Fig. 4b). All these highly active SVAs belong to the recent human-specific SVA-E and SVA-F subfamilies, with the exception of the source element at 20q11.23-1, which contains an exonic sequence from the *MAST2* gene—a signature of the SVA-F1 subfamily[44]. Only a small subset—24% and 17% of

the source L1s and SVAs, respectively—were previously reported to be active on the basis of genomic or in vitro studies (Fig. 4ab and Methods).

Of the 208 L1 progenitors, 156 exhibit only 3′ transductions. This includes the most active source L1 locus, a Ta-1 element residing at Xp22.2 on the reference that mediates 38 transductions, all in 3′ (Supplementary Tables 19 and 20). By comparison, the third most active L1 element at 8q21.11 exhibits 22 transductions, with all of them in its 5′ resulting in a 5′ transduction bias (adjusted $P$ value ($P_{adj}$) = $3.9 \times 10^{-15}$, FDR-adjusted two-tailed binomial test; Fig. 4a,c). A previous report indicates the presence of a strong promoter located upstream of this source L1 (ref. 45), which has been associated with aberrant splicing leading to a somatic 5′ transduction in the embryonic brain[46]. Our analyses of 22 independent germline 5′ transduction events arising from this

source locus suggest that all probably originate from the same mechanism, involving splicing of the first exon of the *ENSG00000253784* long non-coding RNA gene with two alternative acceptor splice sites located within the source L1 (Fig. 4d,e). This provides a mechanistic explanation for the 5′ transduction bias seen for this source element, and indicates that L1 progenitors can hijack flanking regulatory elements leading to transduction bias in the germline.

By contrast, SVA progenitors show a pronounced locus-specific pattern of transduction activity, with 14 among the 20 most active source SVAs showing solely 5′ or 3′ transductions (Fig. 4b), respectively. Of these, four SVA source elements exhibit significant bias towards 3′ transductions (5% FDR; Methods)—including progenitors at 12q24.23, 2q33.1-3, 6p12.1 and 14q11.2-2. In summary, detailed analyses of our resource show family- and locus-specific patterns for L1- and SVA-mediated transductions in the germline.

## Genomic breakpoint homology landscape

Prompted by the comprehensive set of nucleotide-resolved SVs, we comprehensively investigated SV breakpoint junctions, examining 66,198 deletions and 75,238 insertions from our SAGA resource (Fig. 5a and Methods). In examining insertions by class, we find that VNTRs and tandem duplications present extensive breakpoint homology (49.7% and 89.8%, respectively), with the homologous sequences flanking the SV frequently mirroring the inserted element in length (Extended Data Fig. 7 and Supplementary Fig. 36). VNTRs typically form by processes such as replication slippage, HDR and NAHR, which involve DNA sequence homology[47]. But similar to simple tandem duplications, the allelic structure of VNTRs generates homologies at the SV flanks independently of the mechanism of formation, thus necessitating separate consideration of these two SV classes when analysing breakpoint junctions

Leveraging the annotations provided by SVAN, our analysis of breakpoint junctions verifies that most MEIs exhibit TSDs of 10–20 bp at the respective insertion site (Fig. 5b,c). When analysing SVs not annotated as VNTR, tandem duplication or MEI, 35.0% of deletions and 28.7% of insertions exhibit homologous flanks exceeding 50 bp, indicative of SV formation through HDR processes. A considerable subset thereof—10.8% of the deletions and 6.7% of the insertions—are flanked by more than or equal to 200 bp of homology, and are probably mediated by NAHR[48].

We further identify several clusters of SVs flanked by Alu, L1 and LTR elements annotated in the reference genome (Fig. 5b,c,e), the formation of which is likely to be mediated by HDR- and NAHR-driven transposable element-mediated rearrangement (TEMR)[49]. Among these, we find Alu-flanked SVs to be much more common in deletions ($n = 3,260$) than in insertions ($n = 80$). This group of SVs predominantly harbours pairs of full-length Alu elements at their flanks (89.3%), visible as a breakpoint homology length peak of 295 bp (Fig. 5b). Notably, these presumably Alu element-mediated SVs show a wide distribution of SV lengths, ranging from about 300 bp up to 20.4 kb for deletions and 9.5 kb for insertions. AluY and AluSx elements in all combinations constitute 23.5% of all SV-flanking Alu pairs. The use of the different Alu subfamilies at breakpoint junctions highly correlates with their counts in the reference genome (Supplementary Fig. 37)—with the exception of the AluJ family, one of the oldest Alu families, whose members appear several times less frequently at the flanks of SVs than expected by their reference genome count (for example, with a 24-fold reduction seen for AluJb).

We also find some L1-flanked SVs, which similar to Alu-flanked SVs are much more common in deletions ($n = 219$) than in insertions ($n = 1$). These L1–L1 pairs mediate relatively large SVs up to 62.9 kb in size (Fig. 5c). Pairs of L1PA family members are most often represented, partaking in 76.4% of L1–L1-mediated SVs. L1 elements at deletion flanks are typically truncated (median length: 1.7 kb), with only 5.0% of L1 sequence homology-mediated deletions flanked by full-length L1

elements. We additionally observe three distinct clusters corresponding to LTR-flanked deletions and insertions, reflecting SVs involving members of the HERVK (deletion: 39, insertion: 3), HERVH (deletion: 111, insertion: 9) and mammalian apparent LTR retrotransposon (deletion: 46, insertion: 4) families. Apart from 30 events subjected to truncation, the homology lengths correspond to the length of the LTRs flanking the respective integration site, with the SV length corresponding to the size of the viral integration plus one LTR, indicating LTR recombination[50]. Determination of the ancestral state of SV alleles through sequence alignment to the chimpanzee genome confirms that the vast majority of TEMRs indeed represent deletions (Supplementary Note 7).

Moreover, we find 59 segmental duplication (SD)-mediated deletions (Fig. 5f), which are up to 31.8 kb in size and exhibit a homology length of up to 14.9 kb. We note that our approach is expected to exhibit decreased sensitivity in identifying SVs flanked by SDs of a similar length or larger than the N50 read length, suggesting that larger SD-mediated SVs are underrepresented.

We next analysed 30,449 deletions not categorized into a particular subclass by SVAN and lacking homology of at least 50 bp at their breakpoints. Of these, 2,812 (9.2%) exhibit blunt-ended breakpoint junctions whereas 25,311 (83.1%) exhibit microhomology of 1–15 bp in size, indicative of SV formation by non-homologous end-joining, alternative end-joining or microhomology-mediated break-induced replication[51]. The remaining 2,326 (7.6%) events show microhomology of 16–49 bp in size, and are likely to have formed through homology-independent processes as well. We further find that inverted duplications rarely show homology at their breakpoints (8.8%), with most exhibiting microhomology of 1–15 bp at their respective breakpoint junction, suggesting a replicative formation origin[51] of this SV class.

Last, we analysed the distribution of SV breakpoint homologies with increasing SV allelic length. Notably, this distribution lacks appreciable inflection points, notwithstanding peaks in homology length corresponding to the sizes of mobile elements at the deletion flanks (Fig. 5c,d). This implies that homology-mediated SV formation processes use a wide variety of flanking repeat lengths, rather than reflecting distinctive peaks as might be expected from a minimal processing length for homologous recombination[52].

## Recurrence of TEMR-mediated deletions

Given the abundance of mobile element sequences in the genome sharing sequence homology, we postulated that occasionally SVs may have formed recurrently in humans, facilitated by the same flanking element pair. Although precise mutation rate estimates will require larger cohorts sequenced by LRS, we systematically screened for short- to intermediate-length homology-mediated deletions, for which the flanking homology length (200–9,000 bp) would be compatible with flanking mobile elements (that is, TEMR) or small- to intermediate-sized SDs, shortlisting 42 potentially recurrent candidates (Methods). We searched for SNPs for which haplotypes with all four combinations of both SNP alleles with the deletion being absent/present are observed (termed recurrence-indicating SNPs). On the basis of the presence of such SNPs on both sides of the deletion for further quality control, we selected six sites with robust evidence for recurrence, all of which represent TEMRs. One of these involves L2/LINE sequences, whereas the remaining five involve paired *Alu* elements (Fig. 5g, Extended Data Fig. 8 and Supplementary Figs. 38–43).

We illustrate this phenomenon for an 806-bp deletion at 12p13.3 mediated by an AluSx–AluY pair (Fig. 5g). After clustering haplotypes using SNPs in a 100-kb window centred around the deletion (Methods and Extended Data Fig. 8), the four major emerging groups recapitulate the geographical ancestries—with Group 1 mostly comprising EAS samples, Groups 2 and 4 mostly AFR samples and Group 3 mostly SAS samples. The SV allele is present in different haplotype backgrounds of Groups 2 and 3, as well as in an outlier haplotype that clusters with Group 4 which

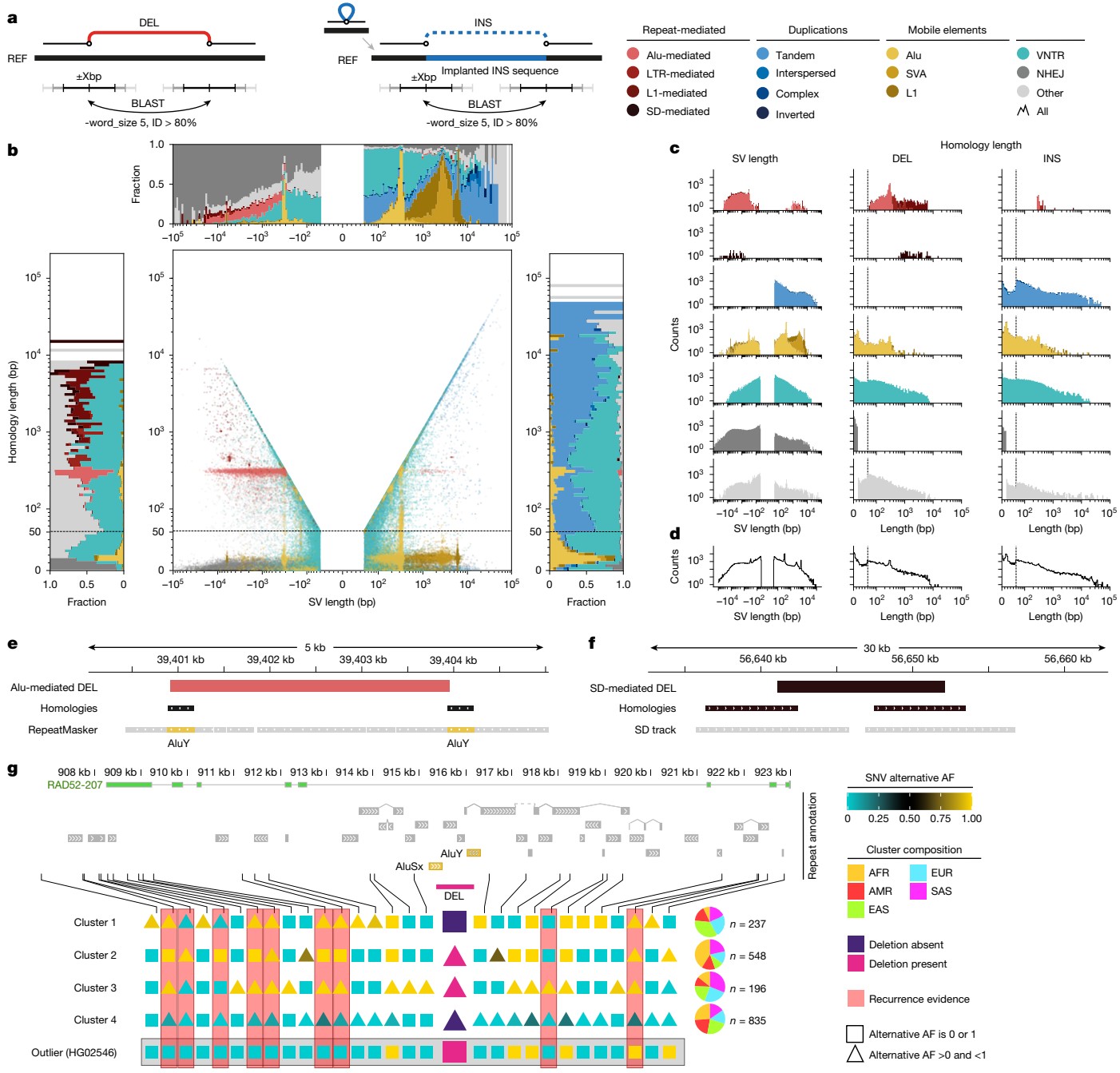

**Fig. 5 | Breakpoint homology landscape and deletion recurrence. a**, Approach for SV breakpoint junction analysis, in the case of primary insertions (INS) achieved by implanting SV sequences into CHM13 (denoted REF). DEL, primary deletion. **b**, SV length versus homology length (DELs depicted with negative length; INS with positive length). Marginal plots show the size-binned fraction of SV classes perpendicular to both axes, depicting DEL and INS at the left and right, respectively. SVs flanked by repeats are shown in different shades of red (mobile element) and black (SD). SVs exhibiting less than or equal to 15 bp of microhomology or blunt-ended breakpoints are in dark grey. Further colouring denotes: duplications (shades of blue), mobile elements (shades of yellow) and VNTRs (cyan). SVs not classified are in light grey. For visualization purposes, scales of scatter plot axes in **a** and *x* axes in **b** are linear up to 50 bp (representing microhomology) and logarithmic afterwards (representing homology); this split is denoted by the dashed line. **c**, SV and homology length distribution for distinct SV classes. **d**, Homology length trend lines for INS and DEL classes combined. **e**,**f**, Schematic view showing two repeat-mediated DELs, an Alu-mediated DEL (**e**) and an SD-mediated DEL (**f**). **g**, An 806-bp DEL at 12p13.3 mediated by an AluSx–AluY pair with evidence for recurrence. For visualization purposes, a consensus haplotype in a 20-kb window centred around the DEL is represented for each cluster. Clusters 1–4 were obtained from SNP-based clustering of the haplotypes in a 100-kb window centred around the DEL. Pie charts represent continental ancestries. Squares are used to represent an allele frequency of 0 (yellow) or 1 (cyan) within a cluster, whereas triangles represent allele frequency values in between. A haplotype from HG02546, grouping with cluster 4, is shown as an outlier. Red bars: SNPs indicating deletion recurrence. NHEJ, non-homologous end-joining.

is otherwise composed of haplotypes not carrying the deletion. We find nine recurrence-indicating SNPs (Fig. 5g, vertical red bars) characteristic of either recurrent deletions or extensive local recombination near the deletion. Given their proximity and association with specific haplotype groups, the most plausible explanation is locally recurrent TEMRs, with scenarios based on homologous recombination on either side of the event seeming less likely. These data imply a relevant contribution of TEMRs to recurrent SV formation in the human genome.

## Discussion

We show that intermediate-coverage LRS enables the generation of comprehensive population-scale SV catalogues comprising common and rare alleles. We predict our resource to serve as benchmarking data for tool development, supporting the continued development of graph-based genetic variant characterization methods[53–55].

Mobile elements are accountable for 1.32–3.14 Mb (median: 2.26 Mb) of the sequence variation detected per individual genome, with MEIs, TEMRs, transductions and processed pseudogenes contributing. This represents a considerable portion of the 4.78–9.88 Mb (median: 7.35 Mb) of sequence variation per genome attributed to SVs on the basis of the SAGA framework (Supplementary Table 21 and Supplementary Figs. 44 and 45). We show a subset of these have arisen recurrently in humans, opening up possibilities to study linkage disequilibrium patterns surrounding these events, and SV recurrence rates, in the future.

With respect to L1 and SVA transductions, whether the transduction occurs at the 5′ or 3′ end is influenced by mobile element class and locus. For instance, although L1s primarily mediate 3′ transductions[9], the L1 progenitor at 8q21.11 exclusively produces 5′ transductions in our population-scale dataset and in the embryonic brain[46]—a phenomenon probably driven by promoter hijacking[45]. Similarly, the 13q12.3 source L1, previously reported to mediate 3′ transductions in brain tissue[45], shows consistent behaviour in germline genomes, which implies shared mechanisms across cell types. A recent report suggests that strong polyadenylation sites downstream to source L1 loci can mediate 3′ transduction in cancer[56]. Altogether, these observations support the notion that transduction is influenced by locus-specific sequence determinants, such as regulatory elements.

Looking forward, we anticipate that our resource will support research on the phenotypic implications of SVs. A comparison with the Genome Aggregation Database (gnomAD)[57], a genetic variation repository with population-level allele frequencies, shows that most (50.9%) of the insertions from our resource were not previously represented in gnomAD. These include 8,077 Alu (34.8%), 2,586 L1 (52.9%) and 1,269 SVA element (47.1%) insertions. For deletions, 14.5% were not previously reported, reflecting the capabilities of short reads in deletion discovery[3,4,23] (Supplementary Note 3). To explore the potential utility of our resource for variant prioritization, we analysed deep PacBio HiFi LRS data generated from four patients with rare diseases (Supplementary Note 8). Initial filtering using multi-platform whole-genome assemblies[20] left an average of 386 candidate SVs per patient. Incorporating our SAGA resource reduces these candidate SV lists by 54.7–56.4%, resulting in 159–187 candidate SVs remaining per patient (Extended Data Fig. 9 and Supplementary Figs. 46–48). We also evaluated the extent by which previously validated disease-causing SVs are filtered out through our resource, by analysing 31 published PacBio-sequenced rare disease genomes, comprising 35 validated causal SVs[14]. Of these validated causal SVs, only two were filtered out using our resource (Supplementary Note 9). These analyses show effective filtering of candidate SVs in patients, with causal SVs sensitively retained, using our LRS population-scale resource.

To further illustrate practical applications of our resource, we additionally explored the genotyping of medically relevant genes in complex loci of the genome. This was facilitated by the use of Locityper, a tool designed to address the challenges of accurately genotyping regions difficult to ascertain. We analysed 270 genes thought to represent challenging loci yet crucial for understanding genetic underpinnings of disease (Methods and Supplementary Table 22). Leveraging our resource, we find a high level of genotyping accuracy in the 1,019 samples of our study, particularly in regions fraught with structural complexity, considerably surpassing genotyping with short reads (Extended Data Fig. 10 and Supplementary Note 10).

The haplotype data derived from our ONT resource are also anticipated to facilitate long-read-based DNA methylation analyses (Supplementary Fig. 45 and Supplementary Note 11), as well as to enhance variant imputation efforts, including in genomic areas previously difficult to analyse. To explore the latter aspect, a companion paper to this study demonstrates the usage of our ONT data resource for imputation and genome-wide association studies[58]. In summary, these observations suggest broader value of our dataset for ascertaining disease-relevant genetic variation.

Although the data and computational methods developed in our study mark a step forward in cost-effective population-scale SV characterization, remaining challenges include increasing discovery power and sequence consensus quality for extremely rare SVs. Large inversions, centromeric regions, high sequence identity SDs and multiallelic VNTRs represent a particular challenge to graph-based approaches[21] (Supplementary Fig. 12 and Supplementary Note 3). Resolving these complex regions comprehensively at population scale will necessitate expanding multi-platform assembly approaches[20] to more samples. Future improvements in graph algorithms should address needs for scalable analyses while considering all polymorphisms, including SNPs, indels and short tandem repeats.

As SAGA facilitates the integration of intermediate-coverage sequence data, we foresee broader adoption of our approach in large cohorts, which could accelerate advancements in population and disease research. Thereby, a coupled approach enhancing sample size at intermediate coverage and increasing sequencing depth on sample subsets seems particularly promising. With this in mind, our group has coordinated sample set design, as well as our public data release, with efforts piloting genome sequence assembly in smaller 1kGP sample subsets[12,22,59]. This approach to open data sharing is guided by the principles of the 1kGP and is inspired by the potential to combine intermediate- and high-coverage techniques to advance the completion of the catalogue of human genomic variation encompassing the entire 1kGP cohort in the near future. Moving forward, it will be crucial to embed these findings into efforts aimed at understanding the genetic underpinnings of diseases, promoting equitable and global progress in precision medicine.

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

# Methods

## Samples and DNA extraction

We initially selected 1,064 samples from the 1kGP collection, sourcing genomic DNA from Coriell and discarding low-quality DNA. The study complied with all relevant regulations for work with human participants. DNA was extracted from lymphoblastoid cell lines and resuspended in TE buffer (10 mM Tris, pH 8.0, 1 mM EDTA).

## Quality control before sequencing

DNA concentrations were determined with the Quant-iT dsDNA Broad-Range assay Kit (Q33130), using Varioskan LUX multimode microplate reader (VL0000D0). The purity evaluation of genomic DNA was performed on a DeNovix DS-11 Series Spectrophotometer. Optical density 260/280 and 260/230 ratio of 1.8 and 2 was maintained, respectively, for sequencing. Fragment length was measured on the Femto Pulse system (Agilent, M5330AA) using the Genomic DNA 165 kb Ladder Fast Separation assay with a separation time of 70 min (Agilent, FP-1002-0275).

## Size selection

All DNA samples were size selected using the Circulomics Short Read Eliminator Kit (Circulomics, SS-100-101-01). According to supplier information, the kit uses size-selective precipitation to reduce the amount of DNA fragments below 25 kb in length. The kit was used according to the manufacturer's recommendations (handbook v.2.0, 07/2019). Briefly, 60 µl of Buffer SRE was added to the sample tube (60-µl volume), gently mixed and the tube centrifuged at 10,000g for 30 min at room temperature. After supernatant removal, two washing steps were performed with 200 µl of 70% ethanol and a centrifugation at 10,000g for 2 min at room temperature. Finally, 50 µl Buffer EB was added and the tube was incubated at 50 °C for 30 min, followed by overnight incubation at room temperature to ensure efficient DNA elution.

## LRS

Sequencing library preparation was carried out following the general guidelines from ONT, with modifications proposed by New England Biolabs (NEB) to ensure high-yield data generation and long-fragment sequencing. For library preparation, the following reagents were used: Ligation Sequencing Kit (ONT, SQK-LSK110), NEBNext Companion Module for ONT Ligation Sequencing (NEB, E7180S) and AMPure XP beads (made in-house by the Molecular Biology Service, Research Institute of Molecular Pathology). A DNA amount of 3 µg as input material was transferred into a 0.2-ml thin-walled PCR tube and the total volume was adjusted to 48 µl with nuclease-free water (Thermo Fisher, AM9937). DNA fragments were repaired and end-prepped as follows: 3.5 µl of NEBNext FFPE DNA Repair Buffer, 2 µl of NEBNext FFPE DNA Repair Mix, 3.5 µl of NEBNext Ultra II End Prep Reaction Buffer and 3 µl NEBNext Ultra II End Prep Enzyme Mix were added to each tube. After mixing and spinning down, the samples were incubated at 20 °C for 30 min, followed by a second incubation at 65 °C for 5 min. The prolonged incubation time allowed recovery of longer fragments. The solution from each tube was then transferred to a clean 1.5-ml Eppendorf DNA LoBind tube (Eppendorf) for clean-up. First, 60 µl of AMPure XP Beads was added to each tube. The samples were then incubated on a HulaMixer sample mixer (Thermo Fisher Scientific, 15920D) for 5 min at room temperature. Bead clean-up was performed with two washing steps on a magnetic rack, each time pipetting off the supernatant and adding 200 µl of freshly prepared 70% ethanol. The pellet was resuspended in 61 µl of nuclease-free water and incubated for 5 min at room temperature. Tubes were placed on a magnetic rack to collect the final eluate (1 µl was then taken out for quantification). For adaptor ligation and clean-up, 60 µl of DNA from the previous step was combined with 25 µl of Ligation Buffer, 10 µl of NEBNext Quick T4 DNA Ligase and 5 µl of Adapter Mix in a 1.5-ml Eppendorf DNA LoBind tube. The reaction was then incubated for 20 min at room temperature. A second AMPure bead clean-up step was carried out by adding 40 µl of bead solution to each tube, followed by incubation on a HulaMixer for 5 min at room temperature. After pipetting off the supernatant on a magnet rack, the beads were washed twice with 250 µl of Long Fragment Buffer. Finally, the supernatant was discarded, and the pellet was resuspended in 25 µl of Elution Buffer EB and incubated for 10 min at 37 °C to collect the final library. Samples were quantified using a Qubit fluorimeter and diluted appropriately before loading onto the flow cells. The final mass loaded on the flow cells was determined on the basis of the molarity, dependent on average fragment size. Sequencing was carried out using FLO-PRO002 (R9.4.1) flow cells from ONT on the PromethION 48. The sequencing run was stopped after 24 h, the flow cell was washed using the Flow cell wash kit XL (ONT, EXP_WSH004-XL) and then the library was reloaded.

## Base calling and adaptor trimming

Guppy v.6.2.1 was used for base calling the Fast5 input files in 'sup' accuracy mode with adaptor trimming and read splitting disabled and the output was converted to FASTA. For adaptor trimming and read splitting, we subsequently used Porechop[61] v.0.2.4 on the generated FASTA files in chunks of 300,000 reads with default parameters.

## Reference genome alignments

Reads were aligned to the GRCh38 (ref. 62) and CHM13 (ref. 63) linear reference genomes as well as the prebuilt human genome graph, denoted HPRC_mg, which was previously constructed by the HPRC[12] (from HPRC year-1 samples; https://doi.org/10.5281/zenodo.6983934). We selected HPRC_mg as the pangenomic reference as it represents SVs while omitting SNPs and less than 50-bp indels[12], thus comprising a compact graph structure facilitating analyses at the scale of a thousand long-read genomes. For the GRCh38 and CHM13, we used minimap2 (ref. 64) to map the ONT reads using the options '-a -x map-ont --rmq=yes --MD --cs -L'. Samtools[65] was used to sort the alignments and convert to CRAM. Multiple ONT runs for the same sample were tagged using different read-groups. Minigraph[21] v.0.20-r559 was used to map the ONT reads against the HPRC_mg graph genome using the options '--vc -cx lr'. During alignment, the '--vc' flag enables the output of the alignments in vertex coordinates and the 'lr' option enables long-read mapping. The resulting graph alignments in GAF format[21] were sorted using gaftools[66] and compressed using bgzip.

## Sample and alignment quality control

Using SNP calls of the high-coverage short-read data from the 1kGP cohort[3], we implemented rigorous quality control measures to effectively eliminate sample swaps and cross-contaminations. We first genotyped all SNPs from the short-read haplotype reference panel with an allele count greater than or equal to 6 in all samples using bcftools[67]. Individual VCF files were merged using bcftools[67] into a multi-sample VCF file that was then combined with the short-read haplotype reference panel. We then used VCFtools[68] to calculate a relatedness statistic of the long-read sequenced samples compared with the short-read sequenced samples using the '--relatedness2' option. This analysis identified a sample swap between HG01951 and HG01983, which we relabelled afterwards. We excluded all samples that seemed to be cross-contaminated during library preparation, namely HG00138, HG02807, HG02813, HG02870, HG02888, HG02890, HG03804 and HG03778. Alignments were analysed using NanoPack[69] to determine median and N50 read length and genome coverage (Supplementary Figs. 2, 4 and 49). To compare linear and graph genome alignments in terms of percentage identity, number of aligned reads (bases) and largest Cigar I and D event, we used lorax[70] (Supplementary Fig. 3).

## SV discovery using linear references

We used Sniffles[71] v.2.0.7 and an LRS-optimized version of DELLY[72] (v.1.1.7) to discover SVs using linear reference genomes. For Sniffles,

we converted CRAM files to BAM format using samtools and then calculated for each sample candidate SVs and the associated SNF file using Sniffles. We then used Sniffles population-calling mode on all SNF files to generate two multi-sample VCF files, one for GRCh38 and one for CHM13.

Similarly, we also used DELLY's population-calling mode which first calls SVs by sample using the new long-read (lr) subcommand. We then merged all candidate SV sites using DELLY merge with the options '-p -a 0.05 -v 3 -c' to select PASS sites that are precise at single-nucleotide resolution with a minimum variant allele frequency of 5% and a minimum coverage of 3×. We then genotyped this SV site list in all samples using delly lr and merged the results by id with bcftools merge using the option '-m id'. We then applied sansa (https://github.com/dellytools/sansa), a newly developed, multi-sample SV annotation method that detects SV duplicates on the basis of SV allele and genotype concordance, to remove redundant SV sites. The parameters for the sansa markdup subcommand were '-y 0 -b 500 -s 0.5 -d 0.3 -c 0.1' to mark SV duplicates for sites that show an SV size ratio greater than 0.5, a maximum SV allele divergence of 30%, a maximum SV breakpoint offset of 500 bp and a minimum fraction of shared SV carriers of 10%. After removing duplicates, we generated the final multi-sample VCFs for GRCh38 and CHM13.

To ensure specificity and facilitate genome graph augmentation, we further generated for Sniffles and DELLY separately a consensus callset of SVs shared between the GRCh38 and CHM13 reference genomes. We therefore lifted the GRCh38 callsets to CHM13 using bedtools[73] and the liftOver tool[74] with the GRCh38 to CHM13 chain file. We then compared the lifted VCF with the original CHM13 VCF file to identify shared SVs using sansa's compvcf subcommand with the options '-m 0 -b 50 -s 0.8 -d 0.1' to identify SVs that have a size ratio greater than or equal to 0.8, a maximum SV breakpoint offset of 50 bp and an SV allele divergence of at most 10%. As the SV allele divergence filter requires a local assembly (for example, DELLY's consensus SV allele sequence), we did not apply this filter to Sniffles. For genome graph augmentation, we subset the final VCFs to deletions and insertions only. All inversion-type SVs called by DELLY and Sniffles using the minimap2 alignments were integrated separately in the inversion analysis.

## Inversion analysis

We developed a multi-tiered analytical pipeline to comprehensively ascertain inversions on the basis of LRS data. By inspecting previously known inversions in our dataset, along with simulating a range of small inversions (less than 1 kb) with a coverage closely mirroring our dataset (median 17×), we discovered that minimap2 frequently misaligned reads in small inversion regions, leading to increasing error rates in those genomic locations. To allow capturing of these inversions, we examined strategies for inversion discovery by simulating inversions of varied sizes (Supplementary Note 6). We generated simulated ONT reads from these augmented genomes, using SURVIVOR[75], mimicking the sequencing coverage in our resource. Using pysamstats (https://github.com/alimanfoo/pysamstats), we first calculated the mismatch rate per base pair in 50-bp intervals, initially in the simulated datasets to tune parameters and subsequently in the real data to identify candidate inverted regions for re-alignment. This analytical process showed that most genomic regions maintained a mismatch rate below 20%; regions surpassing this rate were identified as having an unexpectedly high mismatch rate and selected for re-alignment with NGMLR[71] (Supplementary Fig. 50). Post-exclusion of telomeric and centromeric regions, as well as of misaligned regions exceeding 1 kb in size, was then conducted to restrict the number of regions requiring remapping, thereby enhancing computational efficiency. The selected genomic segments underwent re-alignment using NGMLR and we then interrogated all realigned regions with DELLY[72] to discover previously missed inversions. The final inversion calls in the remapped regions from all samples were consolidated using SURVIVOR merge[75]. For the rest of the

regions not requiring re-alignment, inversion calling was conducted using both Sniffles and DELLY. Manual verification of true versus false positive calls was performed by examining dot plots and Integrative Genomics Viewer (IGV)-like plots generated with wally[70] for each candidate inversion location for the largest ten reads per candidate region, ensuring the accuracy of our findings. Ultimately, we generated a final comprehensive inversion callset by merging all unique instances from each dataset with 'bedtools merge' (v.2.31.1)[73]. As the inversion analysis was conducted on the GRCh38 reference genome, regions were subsequently lifted over to the T2T-CHM13+Y genome, applying a 90% base remapping threshold to retain a region.

To evaluate our inversion dataset against two previous studies on 1kGP samples[3,36], we used 'bedtools intersect' (v.2.31.1)[73], defining inversions as 'known' if they exhibited a minimum of 50% reciprocal overlap with inversions from either previous dataset. Analysis against a previous study delineating inversions through whole-genome assembly[36] shows the efficacy of our methodology in detecting a diverse range of inversions, both repeat-mediated and non-repeat-mediated: our results showed that 65% of non-repeat-associated inversions and 41% of SD-mediated inversions were successfully identified. Furthermore, we refined our comparison to a 1kGP-derived short-read inversion dataset[3], for which we included only those inversions from the comparison dataset with quality scores of 30 or higher, to ensure the accuracy of our comparative analysis. This approach showed an overlap of 289 inversions, or 36.5% concordance (median size of 530 bp).

Regarding the flanking repeats in repeat-rich inverted regions, we conducted a detailed analysis by manually inspecting the repeat types and their orientations at inversion breakpoints. Repeat data were acquired from the RepeatMasker track and the SD annotations of the CHM13 reference (obtained from https://github.com/marbl/CHM13); an inversion was classified as repeat-mediated if it was bracketed by repeats in reverse orientation relative to each other, detected through dotplot analysis.

Inversion genotyping was conducted using the GeONTIpe pipeline (commit: 1b5db07) (https://github.com/RMoreiraP/GeONTIpe), which identifies reads spanning inversion breakpoints and determines sequence order and orientation using probes positioned on both sides of the breakpoints. We focused on those inversions not genotyped by Giggles, by excluding inverted duplications and twin priming events. Multiple probe sets were tested, and for each inversion, a validated set of probes was generated and verified on dotplot-confirmed inversion carrier samples. Once the expected orientation was confirmed, the validated set of probes was applied to genotype the rest of the samples. All regions classified as 'inverted duplications' were excluded because of pipeline limitations, yielding a final dataset of 520 inversions. Of these, 407 inversions (78.3%) were successfully genotyped, with both haplotypes identified in 393 cases.

Notably, five inversions exclusively exhibited the inverted haplotype in all samples, suggesting either an inversion in the reference genome or a potential assembly error. In four cases, samples expected to have the inverted haplotype were classified as low confidence, with no other samples exhibiting the inversion. For a further five cases, manual analysis showed either a misclassification as an inverted duplication or the presence of complex structural rearrangements in the putative reads with the inversion.

## Phasing with the ONT reads

To conduct phasing, we first pursued phasing experiments with the ONT reads to check how well they compare to the statistical phasing done previously in high-coverage 1kGP short-read data generated by the New York Genome Center (NYGC)[3]. The NYGC phased VCFs and the NYGC raw genotypes were used. Using the NYGC raw genotypes, the phasing was done by WhatsHap[76] (v.2.0) in three different ways: phasing with only the ONT reads (from hereon referred to as long-read phasing), trio phasing and trio phasing with the ONT reads (from hereon referred

to as long-read–trio phasing). The trio phasing and long-read–trio phasing was conducted for the six complete family trios (family IDs: 2418, CLM16, SH006, Y077, 1463 (paternal side), 1463 (maternal side)) for which our resource has long-read data. The long-read phasing was conducted for all of the 967 samples in the intersection of our 1,019 sample set and the NYGC sample set of 3,202 (Supplementary Fig. 7 and Supplementary Table 23). The phasing was pursued for all autosomes, and each chromosome was phased separately to allow parallel processing. The commands used are as follows:

- Long-read phasing: whatshap phase -o <output phased vcf> --chromosome <chromosome ID> --sample <sample name> -r <reference fasta> <NYGC raw genotypes for sample> <ONT CRAM for sample>
- Trio phasing: whatshap phase -o <output phased vcf> --chromosome <chromosome ID> -r <reference fasta> --ped <pedigree data for the NYGC samples> <NYGC raw genotypes for samples in a family>
- Long-read–trio phasing: whatshap phase -o <output phased vcf> --chromosome <chromosome ID> -r <reference fasta> --ped <pedigree data for the NYGC samples> <NYGC raw genotypes for samples in a family> <ONT CRAM for samples in a family>.

### Comparison of the WhatsHap phased VCFs against NYGC statistical phasing

The phased VCFs produced by WhatsHap were compared against the NYGC statistical phasing using WhatsHap's compare function. The commands are as follows:

- For the samples without trio data: whatshap compare --sample <sample name> --names longread,nygc --tsv-pairwise <pairwise tsv file> --tsv-multiway <multiway tsv name> <input longread phased vcf> <input nygc statistical phasing vcf>
- For the samples with trio data: compare --sample <sample name> --names trio,longread,trio-longread,nygc --tsv-pairwise <pairwise tsv file> --tsv-multiway <multiway tsv name> <input trio phased vcf> <input longread phased vcf> <input longread trio phased vcf> <input nygc statistical phasing vcf>.

### Haplotype tagging of ONT reads

The ONT reads were haplotype-tagged (or haplo-tagged) using WhatsHap[76] (v.2.0). The NYCG phased VCF[3] was used as the reference for tagging the reads. The command used to tag the reads was: whatshap haplotag --skip-missing-contigs --reference <reference fasta> --sample <sample name> --output-haplotag-list <output file> --output /dev/null <NYGC phased VCF> <ONT CRAM>.

Although the main output of whatshap haplotag is a tagged alignment file, downstream tools used in this study required only a file containing the tag for each read which is given in --output-haplotag-list. Owing to the presence of pseudo-autosomal regions (PARs) in the phased VCF, the command was altered for the male samples. Instead of providing the entire NYGC phased VCF, the non-PAR records were removed and the haplo-tagging was performed. After haplo-tagging, the list of reads aligning to the non-PAR on chr. X were extracted, assigned manually as the maternal haplotype and added to the haplotag list.

### SV discovery from the graph

The aim of SVarp[77] is to discover SVs on graph genomes, including for haplotypes missing in a linear reference. SVarp (commit: 0acba75) calls novel phased variant sequences, called svtigs, rather than a variant callset, which we later use in the graph augmentation step. To discover phased SV assemblies (svtigs) on top of the pangenome graph, we used haplotag read information and the alignment file (that is, GAF alignments) as input to the SVarp algorithm using <svarp -a GAF-FILE -s 5 -d 500 -g GFA-FILE --fasta READS-FASTA-FILE -i SAMPLE_NAME --phase HAPLOTAG-FILE> command. With 967 samples, we found a total of 1,108,850 variants (approximately 1,145 per sample).

To find specific SV breakpoint loci with respect to a linear reference genome, we used the PAV tool[22] to call SV breakpoints, using svtigs that SVarp generated as input. This yielded 1,258,880 and 1,241,252 SVs relative to the CHM13 and GRCh38 linear genomes, respectively, that are more than 50 bp. However, we realized that some svtigs give rise to multiple SVs in the output of PAV. To ensure that variants called from the same svtig end up on the same pseudo-haplotypes in the graph augmentation step, we generated a script to combine records arising from the same svtig into a single VCF record. This is achieved by connecting multiple smaller such SVs into a single SV record through adding reference sequence in-between. This yielded 564,661 and 562,311 SVs relative to CHM13 and GRCh38, respectively.

The single-sample VCFs (relative to GRCh38) generated with PAV from the SVarp svtigs were merged into a multi-sample VCF using bcftools merge[67] (v.1.18) and then post-processed using truvari collapse[78] (v.4.1.0). The latter step merges SV records probably representing the same event into a single record, removing redundancy. This reduced the number of SVs from 451,942 to 215,209. Finally, we filtered the resulting VCF by keeping only records present in at least two samples. This filtered set contained 70,932 SVs.

### Graph augmentation

We developed a pipeline to add extra variants found by DELLY, Sniffles and SVarp across the 1kGP ONT samples to the minigraph graph so that they can be genotyped by Giggles (v.1.0)[79]. The main idea is to construct so-called 'pseudo-haplotypes' by implanting sets of non-overlapping variant calls into a reference genome and then adding them to the graph using minigraph[21] (Extended Data Fig. 1). Our pipeline consists of the following steps. At first, we remove variant calls that fall into the centromere regions and mark the respective region in the reference genome by masking the sequence by Ns using the tool bcftools maskfasta (v.1.18). We used the GRCh38 reference genome and centromere annotations obtained from the UCSC genome browser. In the next step, we generate the pseudo-haplotypes as follows. Each of the SV discovery callsets (DELLY, Sniffles, SVarp) contains variants overlapping across samples. Thus, inserting all of them into one reference genome will fail. Therefore, we first group variants of each callset into sets of non-overlapping variants, and then generate a consensus sequence for each of these sets by implanting the variants into the reference genome using bcftools consensus (v.1.18). As a result, we obtain a whole-genome consensus sequence for each of these sets, which we call the pseudo-haplotypes. For the DELLY calls, we obtained 26 such pseudo-haplotypes, for Sniffles we got 69 and for SVarp we generated 117 pseudo-haplotypes. In total, these pseudo-haplotypes carry 154,319 DELLY SVs, 128,688 Sniffles SVs and 70,813 SVs detected by SVarp. In the last step, we insert all of these newly constructed genome sequences into the graph using the minigraph tool (v.0.20). Thereby, the minigraph algorithm incorporates a new SV allele only if it is sufficiently different (that is, shifted by more than or equal to 50 bp) from SV alleles already represented in the graph[21]. We first inserted all SVarp haplotypes, then all DELLY haplotypes and finally all Sniffles haplotypes into the graph using the command: minigraph -cxggs -t32 <minigraph> <SVarp Pseudo-Haplotype FASTAs> <DELLY Pseudo-Haplotype FASTAs> <Sniffles Pseudo-Haplotype FASTAs> > augmented-graph.gfa.

In the augmented graph, the number of bubbles (Supplementary Fig. 8) increases to 220,168 (102,371 in the original graph) and the total sequence represented in the graph increases from 3,297,884,175 bases to 3,477,266,061 bases. To identify bubbles in the augmented graph representing variation that was previously not represented in the original graph, we created BED files with coordinates of bubbles present in both graphs using gfatools bubble. Then we used bedtools closest to compute the distance between each bubble in the augmented graph and their respective closest bubble in the original graph (Supplementary Fig. 10). We defined all bubbles in the augmented graph

whose distance to the closest bubble in the original graph is at least 1 kb as 'new' bubbles, representing novel SV sites.

To evaluate our augmented graph, we aligned ONT reads of human sample HG00513 (a sample not part of HPRC_mg) to HPRC_mg_44+966 using the command: minigraph -cx lr -t24 augmented-graph.gfa HG00513-reads.fa --vc 2 > alignments.gaf. For comparison, we aligned the same set of reads to the original graph (HPRC_mg) using the same command. We then computed alignment statistics using gaftools stat (Supplementary Table 3). We observed better alignment statistics when aligning reads to the augmented instead of the original graph. The number of aligned reads increased by 33,208, and the number of aligned bases by 152,454,715 bp.

### Preparing phased VCF panel

We developed a pipeline that can reconstruct the alleles of the samples in the graph using the graph and the assemblies for the samples. First, the bubbles in the graphs are identified using gaftools[66] (commit ID: feaf7f4). The function order_gfa tags the nodes of the graph to identify whether the nodes are bubble nodes (nodes inside a bubble) or scaffold nodes (nodes outside a bubble). The phased panel is created by aligning the HPRC assemblies back to the tagged HPRC graph using minigraph[21] (v.0.20-r559). The resulting alignments are processed to identify the alleles using the node paths between scaffold nodes. The allele information from the haplotype assemblies is then converted to a phased panel VCF. For the augmented graph, the same pipeline discussed above works where the pseudo-haplotypes are considered as assemblies and aligned to the augmented graph. On the basis of the tagging of the augmented graph, the alleles on the pseudo-haplotypes are identified and separate columns are created in the phased panel VCF corresponding to the alleles of the pseudo-haplotypes. This pipeline creates a multi-sample phased VCF, containing multiple alternative alleles for each record of the VCF.

The phased panel VCF is processed for its inclusion during Giggles[79] genotyping. Internal filter tags are set during the VCF creation step which allows for filtering of the bubbles for which allele information is not present for at least 80% of the samples. Additionally, bubbles are filtered out that do not have any alternative alleles in any of the haplotypes. From the phased panel VCF, a 'Giggles-ready' version is generated for which the reference alleles in the pseudo-haplotypes are masked with dots, because internally Giggles accounts for the allele frequency in the panels and the pseudo-haplotypes heavily bias the genotyping towards the reference used in the bcftools consensus step. The VCF records describe bubble structures in the graph. These bubbles often contain many nested and overlapping variant alleles, that is, a bubble does not necessarily represent a single variation event. To identify variant alleles represented in these bubbles, we applied the same bubble decomposition approach as previously described by the HPRC[12]. In short, the idea is to compare node traversals of the reference and alternative paths through bubbles to identify nested alleles. As in the previous HPRC study[12], we then annotate the bubbles in our VCF to encode nested variants, so that we can translate genotypes computed for bubbles to genotypes for the variants represented in the graph.

### Graph-aware genotyping

Graph-aware genotyping was conducted using Giggles[79] (v.1.0), a pangenome-based genome inference tool that leverages LRS data. Giggles serves as a long-read alternative to PanGenie's short-read-based approach[80]. It works with a Hidden Markov Model framework for which each variant position contains states corresponding to all possible genotypes at that position. The transition probabilities are based on the Li–Stephens model[81] and the emission probabilities are based on the alignment of reads around an interval window of the variant position. Forward–backward computation of the Hidden Markov Model gives the posterior likelihoods of each state at each variant position which can be used to compute the likelihood of genotypes across the genome.

Giggles requires a phased VCF panel, sorted graph alignments, the input long reads, tagged graph and the haplotype tagging of the reads. We use the command: giggles genotype --read-fasta <input reads> --sample <sample name> -o <output vcf> --rgfa <tagged gfa> --haplotag-tsv <haplotype tags> <phased panel VCF> <sorted alignments>. This outputs an unphased VCF with the genotypes for the given sample.

The VCFs were further filtered using bcftools[67] to create a high-quality genotype set. This was achieved by masking genotypes of samples having a genotyping quality of less than 10 and dropping variant positions for which more than 5% of genotypes are missing. The commands are: bcftools view --min-ac 1 -m2 -M2 <vcf> | bcftools +setGT ---t q -n. -i 'FMT/GQ<10' | bcftools +fill-tags ---t all | bcftools filter -O z -o <filtered-vcf> -i 'INFO/AC >= 1 && INFO/F_MISSING <= 0.05'.

We stratify the SVs as biallelic or multiallelic on the basis of the bubble these SVs originate from. If the bubble has more than two alleles defined in the phased VCF panel given to Giggles as an input, then the SV is defined to be multiallelic. Otherwise, if only one alternative allele is defined, the SV is defined to be biallelic.

### Mendelian inconsistency statistics

For the six complete family trios in our dataset, we calculated Mendelian inconsistency statistics (Supplementary Fig. 51). We provide the confusion matrices and various statistics for each family, genotyped against HPRC_mg and HPRC_mg_44+966, and also report for various variant types depending on whether the variant came from a biallelic or multiallelic bubble (Supplementary Tables 5–12). The definitions for each statistic can be found in Supplementary Table 4.

### Statistical phasing using high-coverage short-read data of 1kGP samples

Using SHAPEIT5 (ref. 82), we statistically phased the multi-sample VCF file outputted by Giggles using a recently constructed CHM13 haplotype reference panel[19]. This panel uses SNP and InDel calls generated from 1kGP short-read data[3,83] for CHM13. We first subsetted the unphased input VCF from Giggles with 167,291 SVs to the 908 unrelated samples present in the haplotype reference panel, set genotypes with a quality less than 10 to 'missing' and subsequently dropped all SV sites with allele count zero. We then used the SHAPEIT5 common variant phasing mode to incorporate the new SV alleles into the SNP and InDel haplotype scaffold, yielding a phased VCF with 164,571 SVs. Before the phasing we split multiallelic variants into biallelic variants using bcftools norm[67] with the option '-m -any'. After phasing, we joined multiallelic variants back into the original state using bcftools norm with the option '-m +any'.

Using plink[84], we assessed linkage disequilibrium to nearby SNPs in a window of 1 Mb. We first converted the VCF to plink input files and then used plink to calculate $r^2$ values of all SVs to nearby SNPs. For the linkage disequilibrium analysis, we further subdivided the CHM13 genome into GiaB 'difficult' regions and 'high-confidence' regions[85] (using the CHM13_notinalldifficultregions.bed.gz file; Extended Data Fig. 3 and Supplementary Figs. 26 and 52).

### Comparison with previous SV callsets

For the comparison with SV calls derived from high-quality genome assemblies from the HGSVC we used Truvari[78], using the CHM13 reference genome (Supplementary Note 3). To enable further analyses with previous callsets, we additionally lifted our SV calls to GRCh38 using liftOver[74] with the CHM13 to GRCh38 chain file. As extra baseline callsets (GRCh38-based), we included the callsets produced in ref. 3, ref. 22 and ref. 4, labelled as nygc, pangenie and phase3, respectively (Supplementary Fig. 20). The comparison of SVs was restricted to autosomes and chromosome X for deletions and insertions separately. We used sansa compvcf to compare the SV VCF files using default parameters. We altered the base and comparison VCFs to identify SVs that are distinct

for one or the other callset and to identify potential 1:many SV overlaps among the shared SVs. We used the presence of INSSEQ and the absence of IMPRECISE in the INFO field as criteria for sequence-resolved insertions in previous short-read SV callsets[3] (Fig. 1).

### Population differentiation

We used ADMIXTURE v.1.3.0 (https://github.com/NovembreLab/admixture) to compute admixture for $K = 5$. For performing SV-based principal component analysis we used EIGENSOFT 8.0.0 (https://github.com/DReichLab/EIG). Fst values were calculated using VCFtools for each continental population (against the remaining samples) and each population (against the remaining samples) per site and for 1-Mb windows.

### SV impact estimation on genomic features

We used Ensembl VEP with annotation from the CHM13 rapid release of Ensembl (107) to estimate the impact of the SVs on genomic features. Processing was performed using the command: vep --assembly T2T-CHM13v2.0 --regulatory --offline -i final-vcf.unphased.vcf.

We observe a large difference in the mean allele frequencies of SVs affecting coding (mean MAF 0.009) and non-coding (mean MAF 0.061) regions. This observation and the $P$ value of 0.0 reported by both $t$-test and Kolmogorov–Smirnov test thereby support the hypothesis that the two underlying distributions are different.

### Targeted genotyping of challenging loci

Locityper[86] (v.0.13.3) is a targeted whole-genome sequencing-based genotyper designed for challenging loci. Our initial set of target genes was preprocessed using locityper add -e 300k. Our analysis based on Locityper focused on 270 loci identified by GiaB as challenging, yet crucial for understanding genetic underpinnings of various diseases[26], along with 20 polymorphic *MUC* family genes and the *LPA* gene, which are of particular interest because of their association with various structural haplotypes and health conditions[87,88]. ONT datasets were preprocessed and genotyped with locityper preprocess --tech ONT and locityper genotype, respectively. A database of locus haplotypes was constructed on the basis of the GRCh38 and CHM13 reference genomes, as well as on the basis of the 44 diploid HPRC whole-genome assemblies. To evaluate genotyping accuracy, we aligned haplotypes to each other using the Wavefront alignment algorithm[89], and calculated sequence divergence as a ratio between edit distance and alignment size. Local haplotypes were extracted from the NYGC callset using bcftools consensus.

### SV annotation with SVAN

SV calls generated with our SAGA framework were processed with the SVAN tool (commit: 7b97325, v.1.3) (https://github.com/REPBIO-LAB/SVAN) to classify them in distinct classes on the basis of distinctive sequence features:

**Retrotranspositions.** Canonical retrotransposition events were identified by searching for poly(A) and poly(T) tails at the 5′ end 3′ ends of the deleted and inserted sequences. Poly(A/T) tracts were required to be at least 10 bp in size, have a minimum purity of 90% and be at a maximum of 50 bp from the insert end or beginning, for poly(A) and poly(T), respectively. The sequence corresponding to the TSD was detected and trimmed from either the 5′ or 3′ end of the L1 insert through the identification of exact matches with the genomic sequence at the integration position. To have all candidate retrotransposed inserts in forward orientation, the reverse complement sequence for every trimmed insert occurring in the minus strand was obtained.

Candidate 3′ partnered transductions were detected by searching for a second poly(A) stretch at the 3′ end of the trimmed sequences using the same criteria outlined above. Integrants with a secondary tail were annotated as '3′ transduction' candidates with the sequence in-between the poly(A) stretches corresponding to the transduced bit of DNA,

whereas those with a single tail were classified as 'solo' candidates. To trace transductions to their source loci, transduced sequences were aligned onto CHM13 using BWA-MEM[90]. To maximize sensitivity for particularly short transduction events, a minimum seed length (-k) of 8 bp and a minimum score (-T) of 0 were used. Alignment hits were filtered by requiring a minimum mapping quality of 10. Transduction candidates with less than 80% of the transduced sequences aligning on the reference were further filtered out.

The retrotransposition candidates were further trimmed by removing the poly(A) tails and transduced sequences. To identify processed pseudogenes, orphan transductions, 5′ partnered transductions and retroelement sequences, the resulting trimmed inserts were aligned onto CHM13 using minimap 2.1, as well as BWA-MEM 0.7.17-r1188, by using the parameters described above and with the alignment conducted onto a database of consensus L1, Alu and SVA repeats. The SVA repeats database contains separate sequences for each SVA component, including the Alu-like region, VNTR, SINE-R and exon 1 of the *MAST2* gene, which is characteristic of the SVA-F1 subfamily.

Alignment hits were filtered by requiring a minimum mapping quality of 10, and chained on the basis of complementarity to identify the minimum set of non-overlapping alignments that span the maximum percentage of the trimmed insert. On the basis of the alignment chains, retrotranspositions are classified as processed pseudogenes (more than or equal to 75% aligns on the reference over single or multiple Gencode 35 annotated exons[91]), orphan transductions (more than or equal to 75% aligns on the reference outside exons), 5′ transductions (more than or equal to 75% aligns both into the reference on its 5′ and into the consensus of a retrotransposon) and solo (more than or equal to 75% aligns into a single consensus retrotransposable element). Alignment chains are further analysed to decompose each MEI into its sequence components, including inversions at the 5′ end of L1 and processed pseudogene insertions or the set of individual repeats composing SVA inserts.

**Non-canonical MEIs.** Insertions and deletions without a poly(A) tail detected were aligned into a database of consensus L1, Alu and SVA repeats as described above to identify non-canonical MEIs. Alignment hits were filtered and chained and non-canonical MEI calls made if more than or equal to 75% of the sequence aligned into one or multiple repeat classes.

**Endogenous retroviruses.** Inserted and deleted sequences were aligned with BWA-MEM 0.7.17-r1188 into a database containing consensus ERV and LTR sequences. Alignment hits were filtered and chained as described above. SVs with hits spanning more than or equal to 75% of their sequence on the retrovirus database were classified as Solo-LTR and ERVK, on the basis of the presence of an LTR alone or LTR plus retroviral sequence, respectively.

**VNTRs.** Expansions and contractions of VNTR loci were annotated by processing inserted and deleted sequences with TRF (Tandem Repeats Finder) v.4.04 (ref. 92). TRF hits were processed and chained on the basis of complementarity to determine the minimum set of non-overlapping hits that span the maximum fraction of the sequence. On the basis of the hit chains, SVAN classified VNTRs as simple (more than or equal to 75% of the sequence corresponds to a single repeated motif) and complex (more than or equal to 75% corresponds to more than one repeated motif).

**Duplications.** Diverse classes of duplications, including tandem, inverted and complex duplication events, were annotated by realigning the inserted sequences onto the reference using minimap 2.1. Alignments were filtered to select only those located within a 2-kb window around the insertion breakpoint. These were further chained on the basis of complementarity to determine the minimum set of non-overlapping

hits spanning the maximum percentage of the sequence. On the basis of the alignment chains, duplications are classified as tandem (more than or equal to 75% of the sequence aligns at the insertion breakpoint in forward orientation), inverted (more than or equal to 75% of the sequence aligns at the insertion breakpoint in reverse orientation) and complex (more than or equal to 75% of the sequence aligns at the insertion breakpoint both in forward and reverse orientation).

**NUMTs.** Insertions with more than or equal to 75% of their sequence having one or multiple minimap 2.1 alignments on the mitochondrial reference are classified as NUMT.

### Transduction analysis

L1 and SVA transductions were clustered on the basis of the alignment position on the reference of their corresponding transduced sequences to identify their source regions. A buffer of 10 kb was applied to their start and end alignment positions before clustering. Source regions were further intersected with a database of full-length loci to determine the progenitor repeat. This database was generated by aggregating all the full-length L1s (more than 5.9 kb in size) and SVAs (more than 1 kb) in CHM13 detected as non-reference insertions in our SV callset. Progenitor repeats were further annotated by determining their insertion orientation and subfamily. For L1 events, subfamily assignment was performed through the identification of subfamily diagnostic nucleotide positions on their 3′ end. L1 progenitors bearing the diagnostic 'ACG' or 'ACA' triplet at the 5,929–5,931 position were classified as 'pre-Ta' and 'Ta', respectively. Ta elements were subclassified into 'Ta-0' or 'Ta-1' according to diagnostic bases at the 5,535 and 5,538 positions (Ta-0: G and C; Ta-1: T and G). Elements that did not show any of these diagnostic profiles could not be assigned to a particular category, and their subfamily status remained undetermined. For SVAs, the inserts were processed with RepeatMasker v.4.0.7 to determine the subfamily. If multiple RepeatMasker hits were obtained, the one with the highest Smith–Waterman score was selected as representative. To assess a bias in source elements to generate 5′ or 3′ transduction events, respectively, we conducted two-tailed binomial tests followed by controlling for the FDR according to Benjamini and Hochberg[93]. We included all source elements with at least five non-orphan transductions when conducting statistical testing for 5′ versus 3′ bias.

To investigate source element novelty, we compared active source L1s and SVAs identified in this study with source elements previously reported to be active on the basis of transduction traces seen in germline[22,94] as well as cancer genomes[95,96], and additionally compared our dataset with previously reported in vitro activity measurements[97,98].

### SV breakpoint junction analysis

The detection of homologous sequences and microhomologies flanking deletions and insertions was conducted on the primary callset, after removing calls less than 50 bp in length, using two approaches: Microhomology was quantified using the available homology output from SV calls generated by DELLY[72] (v.1.2.6). We systematically generated DELLY calls for each SV by building synthetic reads carrying the respective SV allele, mapping those synthetic reads with minimap2 (ref. 64) (v.2.2.26) onto the genomic reference of the SV with 4-kb padding and calling the respective SV with DELLY. The synthetic reads consist of a 2-kb reference sequence upstream of the SV start, the inserted sequence (only for insertions and modified for larger insertions) and a 2-kb reference sequence downstream of the SV end coordinate. For insertions longer than 100 bp, we used at most the first and last 50 bp of the inserted sequence to avoid aberrant mapping of the insert, which was especially observed in the case of long insertions. Owing to the use of truncated inserts for insertions, only homologies with a length of 50 bp were considered, and larger values were set to 50 bp. The second approach aimed to capture longer stretches of homology using BLAST[99]-based detection of homologous stretches of DNA. For this, various pairs of search windows around both breakpoints were defined. Search windows were either symmetric with a padding of 50 bp, 100 bp, 200 bp, 400 bp, 1 kb, 2 kb, 5 kb, 10 kb, 50 kb and 100 kb, or asymmetric with a window size of 'SV length', which is shifted regularly along the breakpoint (0 bp upstream/'SV length' bp downstream; 1/6 'SV length' upstream/5/6 'SV length' downstream; …; 'SV length' upstream/0 bp downstream). From the predefined windows, only those leading to no overlap between windows were used. Potential homologies were detected using blastn 2.12.0 with -perc_identity 80 and -word_size 5, for which the sequences inside the search windows were passed using the -subject and -query parameters. To allow for equivalent analyses between deletions and insertions, insertions were implanted into the CHM13 reference genome and the search windows were defined on the modified reference sequence. BLAST results were filtered to ensure flanking homology stretches show the same directionality, span the respective SV breakpoints and contain the respective breakpoint at the relative same position inside the homology segment.

For annotation of repeat-mediated SVs, the overlap of the homologous sequences with RepeatMasker (v.4.1.2) annotations was used. For deletions, overlap of the homologous regions with the elements of the RepeatMasker track of the CHM13 reference[12,22,63] (obtained from https://github.com/marbl/CHM13) was calculated with bedtools[73] (v.2.31.1) intersect. Deletions were classified as repeat-mediated if at least 85% of the bases of the homologous regions on both sides intersect a RepeatMasker annotation of the same class and at least one homologous region spans 85% of the RepeatMasker element. For insertions, the homologous sequences outputted by blastn were annotated with RepeatMasker and insertions for which both homologous flanks show a reciprocal overlap of at least 85% with a RepeatMasker annotation of the same class were deemed repeat-mediated. To analyse SD-mediated deletions, all deletions were intersected with the SD track of CHM13. Hits spanning more than 85% of the homologous region and spanning more than 200 bases or 50% of the SD were defined as putatively SD-mediated.

### VNTR genotyping using vamos

We performed VNTR genotyping on the long-read data from our resource using vamos (v.2.1.3)[7]. We used the VNTR motif annotations provided by the authors of vamos on the CMH13 reference[100] to genotype VNTRs on all the samples of our dataset. We performed quality control on the VNTR calls by comparing the calls produced from these ONT reads with the VNTR calls produced from recently generated multi-platform whole-genome assemblies[20], focusing on the 16 overlapping sampling between the two callsets (Supplementary Fig. 7). For each VNTR position described by the motif annotations, we compared the number of repeat units that were present in the VNTR allele called using vamos. As the VNTRs called on our dataset are read-based and the HGSVC dataset[20] is genome assembly-based, the VNTR alleles of our dataset are unphased whereas the HGSVC VNTRs are phased. For allelic comparison, we paired the ONT 1kGP-based alleles with the HGSVC alleles having the smallest difference in repeat unit counts. We removed the VNTR alleles for which the HGSVC assemblies showed the VNTR to be close to the reference VNTR (when the VNTR allele from the assemblies had a base pair length between 90% and 110% of the reference VNTR allele). We compared the rest of the VNTR alleles to assess the concordance between calls on the basis of our resource and the HGSVC assembly-based calls. We also conducted a comparison for mismatched samples, verifying that random VNTR matching does not produce similar results.

### Deletion recurrence analysis

To detect potential NAHR-mediated deletion recurrence, we analysed deletions shorter than 5 kb in length exhibiting a flanking sequence homology of 200–9,000 bp. The lower bound of 200 bp

was introduced to account for the minimal processing length of flanking repeat sequences previously thought to be necessary for NAHR to occur[52,101]. The upper bound of 9,000 bp was chosen to systematically assess the role of the most abundant repeat elements in the human genome, including small- to intermediate-sized SDs as well as mobile elements, which we consider to be ideal candidate sites for SV recurrence owing to their pervasive presence as repeats in the genome providing substrates for NAHR. Additionally, to focus the analysis on potentially recurrent events, we limited our analysis to deletions with allele frequency of 40–60%. Furthermore, we used Hardy–Weinberg equilibrium and Mendelian consistency statistics to exclude events with potential genotyping errors. Last, we filtered out deletions for which phasing information was unavailable. Using the above-mentioned criteria, we retained 42 deletions for our analysis. To find evidence of recurrence, we used an approach similar to the approach we previously devised to detect recurrent inversions in the genome[36]. We screened for SNPs with at least 10% allele frequency lying within a 20-kb window centred around the deletion in search for positions for which we observe SNP alleles in haplotypes both with and without the deletion–which we denote recurrence-indicating SNPs. Additionally, we used centroid hierarchical clustering to cluster the SNP haplotypes in a 100-kb window centred around the deletion, in an effort to assess whether the haplotypes with and without the deletion appear together in similar clusters, a phenomenon that indicates that the event recurred in humans. For all 42 shortlisted potentially recurrent candidates, we manually inspected these visualizations of SNP haplotypes as well as read alignments from carriers of deletion and reference alleles, respectively, to confirm that genotyping was accurate, that the clustering signal was not driven by recombination and that recurrence-indicating SNPs were present on both sides of the deletion. We selected six candidates with the strongest evidence for SV recurrence upon inspection and included the visualizations as Supplementary Figs. 38–43 and as Extended Data Fig. 8.

### Reporting summary

Further information on research design is available in the Nature Portfolio Reporting Summary linked to this article.

## Data availability

Our open data resource is fully available to the community for download at the IGSR. Data access FTP link: https://ftp.1000genomes.ebi.ac.uk/vol1/ftp/data_collections/1KG_ONT_VIENNA/. The data have additionally been made available at the Amazon Web Services (AWS) cloud computing platform at s3://1kg-ont-vienna and at the European Nucleotide Archive (ENA) under accession PRJEB89727. These repositories contain the graph and linear reference genomes, alignments, input SV callsets, the augmented genome graph, genotyped and phased SVs, as well as auxiliary data used for evaluating SV callset accuracy. Raw and basecalled sequencing data are also available via the OpnMe initiative of Boehringer Ingelheim: https://opnme.com/genomiclens. Further sequencing data are available from the European Genome-Phenome Archive under accession EGAS00001008170.

## Code availability

General information about this project is available at https://github.com/1kg-ont-vienna/sv-analysis. A pipeline to create a multi-sample SVarp VCF from single-sample SVarp calls (used as one of the callsets going into graph augmentation) and the graph augmentation pipeline is available at https://github.com/eblerjana/long-read-1kg (commit: 44a1752). The Giggles genotyping algorithm is available at https://github.com/samarendra-pani/giggles (commit: 5226884). The pipelines and scripts available at https://github.com/marschall-lab/project-ont-1kg (commit: 32d896d) are: haplo-tagging of aligned

reads, phasing experiments with WhatsHap, SV genotyping with Giggles on HPRC_mg and HPRC_mg_44_966, SV discovery with SVarp, chimpanzee ancestral allele annotation, VNTR genotyping with vamos and some pipelines for quality assessment. The repository also hosts analysis scripts for the genotypes and the SVAN annotations. Curation work for VNTR-associated diseases has also been provided. The sequence-to-graph alignment sorting and processing of GAF files was done using gaftools. Gaftools is available at https://github.com/marschall-lab/gaftools (commit: feaf7f4). The Structural Variants ANnotator (SVAN) tool is available at https://github.com/REPBIO-LAB/SVAN (commit: 7b97325, v.1.3). DELLY pangenome filtering mode is available at https://github.com/dellytools/delly (commit: 4984ff2) and sansa for SV comparison at https://github.com/dellytools/sansa (commit: 198be12). The pangenome-based SV discovery tool SVarp is available at https://github.com/asylvz/SVarp (commit: 0acba75). Scripts for finding and annotating homologous flanks at SV breakpoints are available at https://github.com/carstenhain/SV_homology (commit: 534b216). A pipeline to simulate ONT data on a genome with small inversions for benchmarking genome mappers and callers is available at https://github.com/celiatsapalou/Simulations_ONT_Data (commit: cf0513b). A pipeline to export high-mismatch regions, perform remapping and carry out calling inversions is available at https://github.com/celiatsapalou/Small_Inversions_Remap (commit: f9d8a44). The pipeline to genotype inversions using ONT reads is available at https://github.com/RMoreiraP/GeONTIpe (commit: 1b5db07). The pipeline for filtering rare disease-associated SVs using our resource is available at https://github.com/hugocarmaga/rare-disease-analysis (commit: add69e9). A pipeline producing structural variation VCF files from HiFi reads is available at https://github.com/hugocarmaga/variant-calling (commit: 5062a67). DNA methylome analysis scripts are available at https://github.com/santanaw/1kGP_mods (commit: 8c7ba71).

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

**Acknowledgements** We thank all participants of the 1000 Genomes Project, without whom this project would not have been possible. We thank M. Zody, X. Zhao and other members of the HGSVC for valuable feedback, P. Ebert for assistance with data management, and J. Charest and the Vienna BioCenter Core Facilities for assistance with long-read DNA sequencing. We thank P. Contzen for preparing DNA for long-read sequencing of rare disease samples. We thank L. Vissers and A. Hoischen for generously allowing us to use published patient data and providing oversight for their interpretation. Moreover, we acknowledge the EMBL IT services, the Centre for Information and Media Technology at Heinrich Heine University Düsseldorf, the IT services at the IMP, as well as the CRG Core Technologies Programme for providing resources for data processing and analysis. Finally, we thank members of the International Genome Sample Resource (IGSR) for assistance with providing the open data releases for this study. Funding for sequence data production was provided by the MARVL initiative, a collaboration between the IMP, BI X and Boehringer Ingelheim. Additional funding came from the following sources: National Institutes of Health (NIH) (to J.O.K., T.M., grant no. U24HG007497), the Ministry of Culture and Science of the State of North Rhine-Westphalia (to T.M., grant no. PROFILNRW-2020-107-A), the German Research Foundation (to T.M., grant no. 525152594) and the GraphGenomes project funded by the BMBF (to T.M., grant no. 031L0184A, and J.O.K., grant no. 031L0184C). W.H. received funding from the European Union's Horizon 2020 research and innovation programme under the Marie Skłodowska-Curie grant agreement no. 101150006. B.R.-M. was supported by a Bridging Excellence Fellowship provided by the Life Science Alliance. The project also received the support of a fellowship from the 'la Caixa' Foundation (ID 100010434), with fellowship code 'LCF/BQ/DI24/12070028' (to J.E.S.-F.). Additional funding came from the EU Horizon 2020 research and innovation programme under the Marie Skłodowska-Curie grant agreement no. 713673. We acknowledge support of the Spanish Ministry of Science and Innovation through the Centro de Excelencia Severo Ochoa (grant no. CEX2020-001049-S, MCIN/AEI /10.13039/501100011033), and the Generalitat de Catalunya through the CERCA programme. We received support with data management from the EMBL throughout this project. We thank the DFG Research Infrastructure West German Genome Center (grant no. 407493903) as part of the Next Generation Sequencing Competence Network (project 423957469) for the LRS support.

**Author contributions** M.A. and S. Schloissnig were responsible for data acquisition; P.H. and T.R. were responsible for base calling, read alignment and primary SV discovery versus linear references; S.P. performed read-based haplotype phasing, haplotype tagging of reads, SV genotyping, VNTR calling and chimpanzee ancestral allele annotation; T.R. performed statistical phasing of the SV genotypes; A.S. performed graph-based SV discovery with SVarp; J.E. was responsible for implementation and execution of the graph augmentation pipeline; S.P. and T.R. provided the final SV callset construction and SV callset benchmarking; S. Schloissnig performed the analysis of geographic stratification of SV alleles; C.H. performed breakpoint homology analysis; B.R.-M. and J.E.S.-F. performed SV class annotation and mobile element analysis; H.A. performed the deletion recurrence analysis; T.P. performed the complex locus genotyping; T.F., W.S.-G. and E.B. performed explorative analyses of DNA methylation; V.T., R.M.-P. and M.C. performed the inversion analysis; S.H. provided data management assistance; T.E.W. and F.J.P.-L. performed PacBio sequencing of patients with rare diseases; H.M., W.H., S. Sivalingam, D.W., C.G. and T.M. analysed the genomes of patients with rare diseases; S. Schloissnig, Z.D., J.N.J., N.P. and J.S. provided data production planning; T.R., T.M. and J.O.K. jointly supervised the project; H.A., C.H., S.P., B.R.-M., T.M., T.R., S. Schloissnig, V.T. and J.E.S.-F. prepared the main figures; H.A., J.E., C.H., J.O.K., T.M., S.P., T.P., B.R.-M., A.S., S. Schloissnig, V.T. and T.R. wrote the manuscript, with input from all authors.

**Funding** Open access funding provided by Heinrich-Heine-Universität Düsseldorf.

**Competing interests** Z.D., J.N.J. and N.P. are employees of Boehringer Ingelheim Pharma GmbH & Co. KG. J.S. is an employee of BI X GmbH. E.B. is a consultant for and shareholder of Oxford Nanopore Technologies. The other authors declare no competing interests.

**Additional information**
**Correspondence and requests for materials** should be addressed to Bernardo Rodriguez-Martin, Tobias Rausch, Tobias Marschall or Jan O. Korbel.

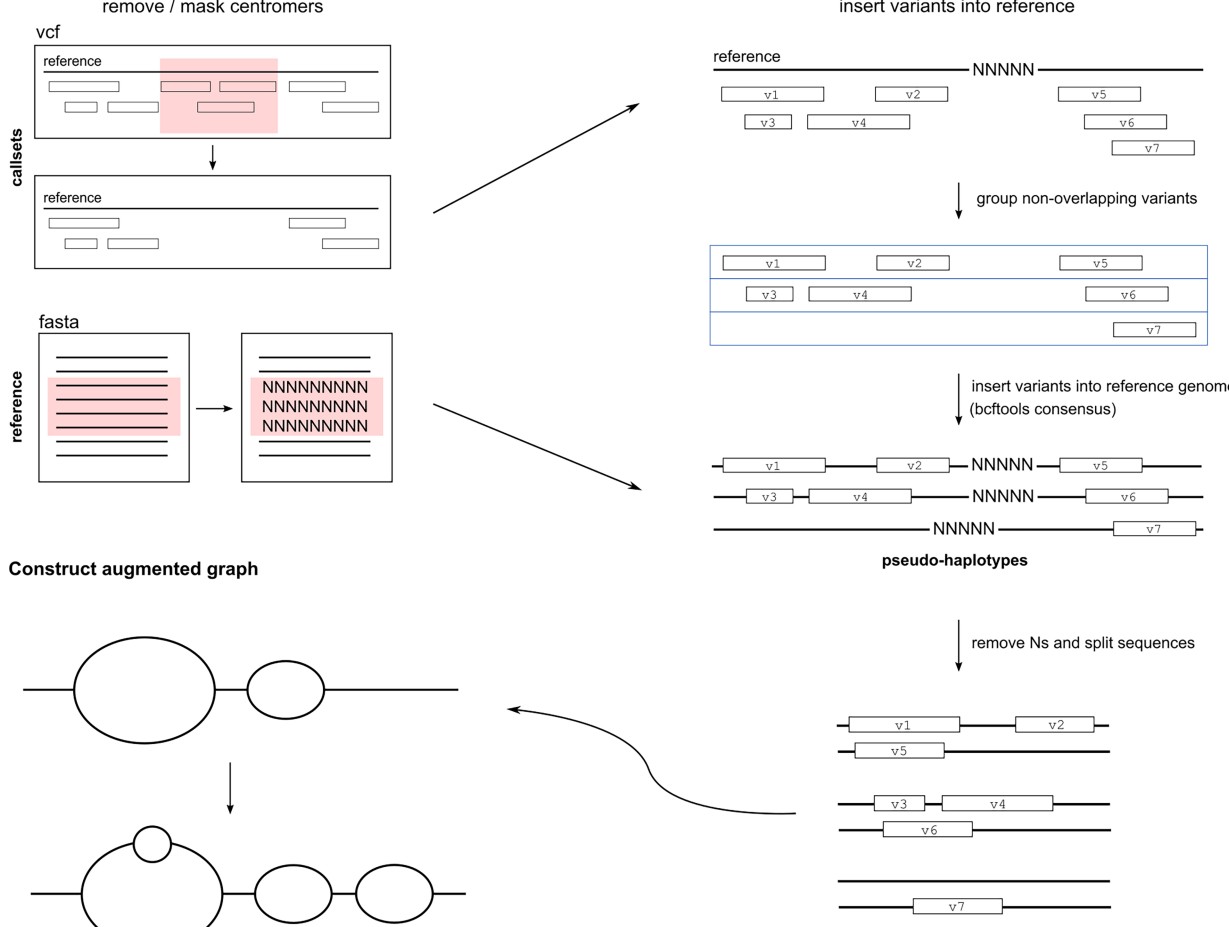

**Extended Data Fig. 1 | From pseudo haplotypes to generating an augmented graph.** Variant calls within centromere regions are removed and centromere regions are masked by 'Ns' in the reference genome. Then, sets of non-overlapping variants are grouped and inserted into the reference genome to obtain "pseudo-haplotypes". Finally, pseudo-haplotypes are added as new sequences, thereby augmenting the graph, using the minigraph tool.

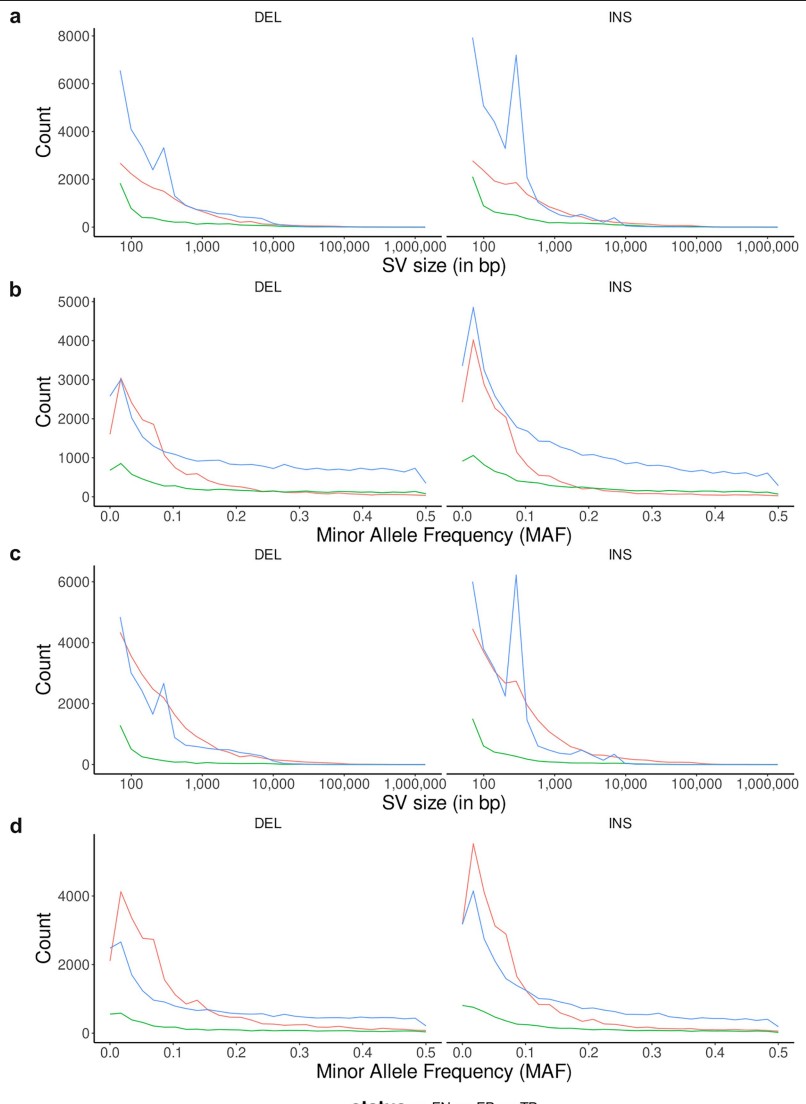

**Extended Data Fig. 2 | Evaluation of true positive (TP), false positive (FP) and false negative (FN) SV calls. a**) SV size compared to recently generated multi-platform whole genome assemblies[20] on 16 overlapping samples, for the Giggles genotyped callset, **b**) SV minor allele frequency (MAF) compared to these whole genome assemblies on 16 overlapping samples, for the Giggles genotyped call set, **c**) SV size compared to these whole genome assemblies on 16 overlapping samples for the final VCF, which was further filtered for high-quality genotypes emitted by Giggles (**Methods**) and **d**) SV MAF compared to these whole genome assemblies, on 16 overlapping samples for the final VCF that was further filtered for high-quality genotypes emitted by Giggles (**Methods**). DEL, deletion; INS, insertions.

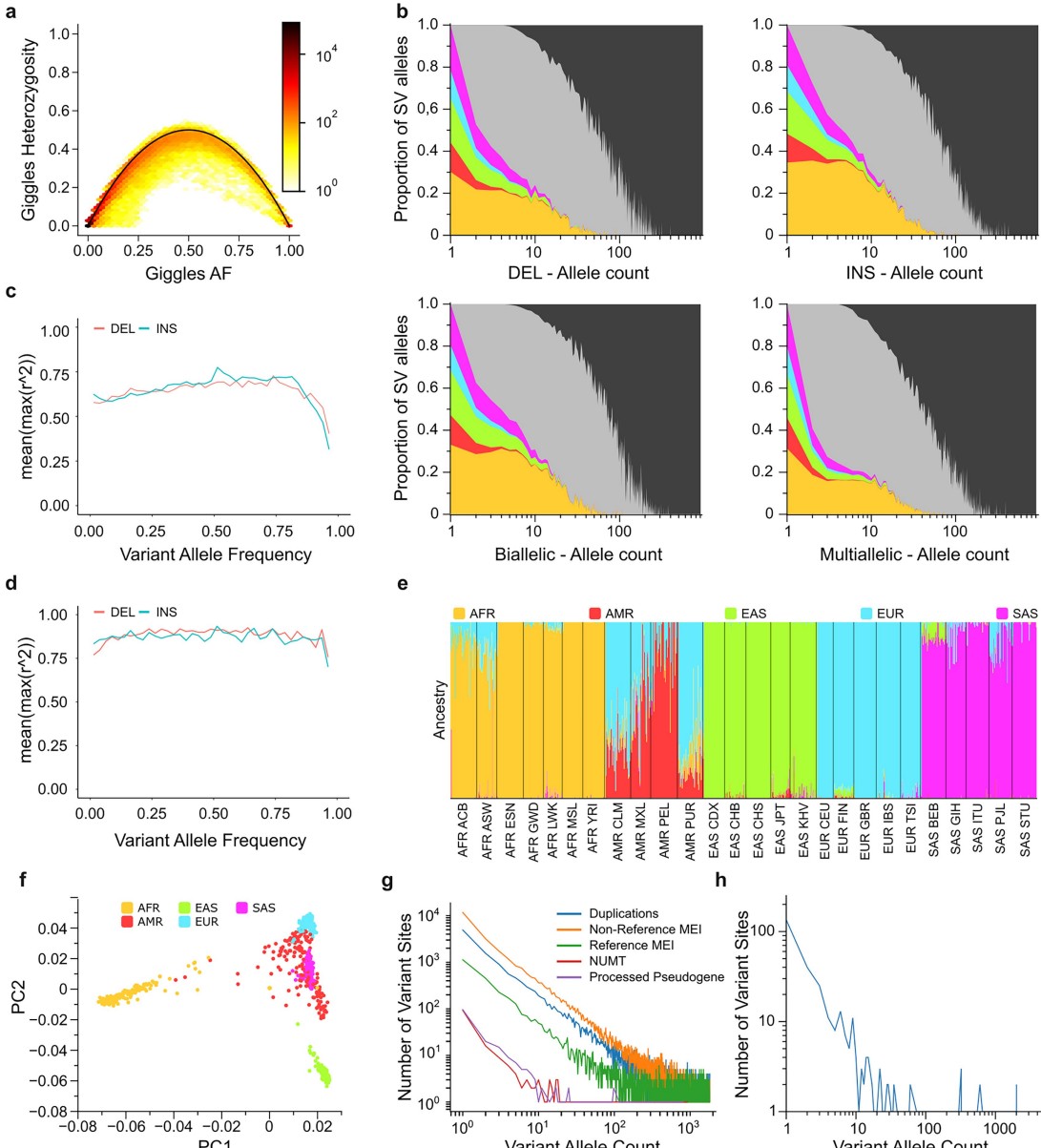

**Extended Data Fig. 3 | Quality assessment and population characteristics.**
**a**) Quality of the genotypes by Giggles on the HPRC_mg_44 + 966 graph after
filtering. Genotyping quality is shown here using a Hardy-Weinberg Equilibrium
(HWE) plot given with the allele frequency of the genotyped allele and the
percentage of samples heterozygous for that allele (using only the 908 unrelated
samples from our dataset). **b**) SV allele sharing across continental populations.
Grey: shared by at least two (and less then all) continental groups. Black: shared
by all continental groups. Deletions (top left), insertions (top right), all biallelic
SVs (bottom left), all multiallelic (SVs). **c**) Linkage disequilibrium (LD) of all SVs
(MAF > = 1%) with nearby single nucleotide polymorphisms (SNPs). **d**) As c)
with SVs restricted to Genome in a Bottle high-confident regions of the
CHM13 genome (2.3 Gbp, 74.2%). **e**) SV-based admixing spectrum using five

reference populations. **f**) Principal component analysis using all SVs. **g**) Relation
between Variant Allele Count and the Number of Variant Sites with that allele
count in the logarithmic space for the SV genotypes on the HPRC_mg_44 + 966
graph, annotated by SVAN. Duplications (DUP), Mobile element insertions
and deletions (MEI (non-reference) and MEI (reference), respectively), Nuclear
mitochondrial DNA integration (NUMT), processed pseudogene integration
(PSD). **h**) Relationship between the Inversion Allele Count (AC) and the Number
of Variant Sites with that allele count shown in log-space for the GeONTIpe
based inversion genotypes. The majority of inversions are rare, with most
exhibiting an AC < 10. A small subset of inversions is observed more frequently
across populations, with 37 inversions exceeding an AC of 1,000, potentially
corresponding to reference genome inversions.

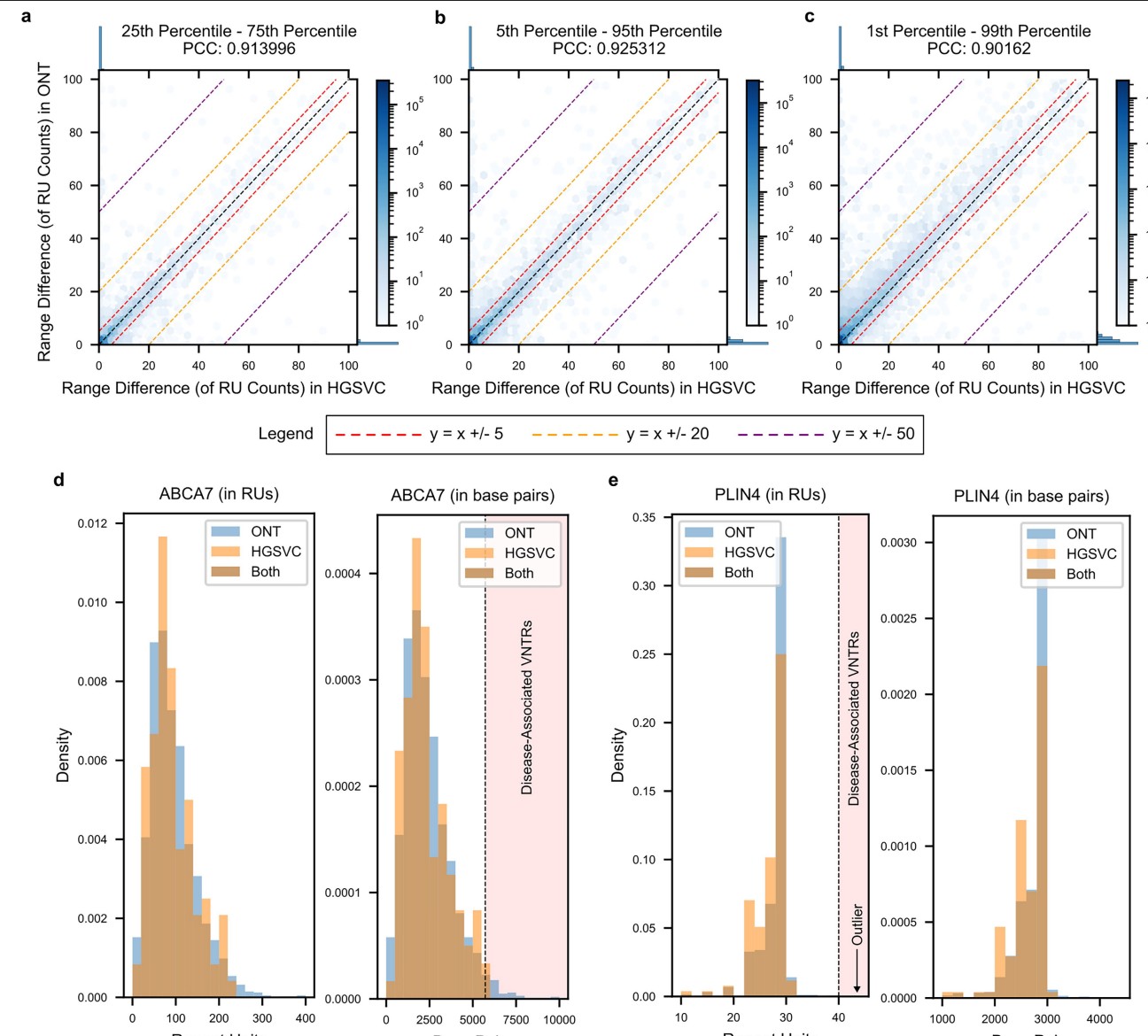

**Extended Data Fig. 4 | VNTR genotyping using vamos. a-c)** Density plots comparing the range difference of repeat unit (RU) counts for different percentile ranges for VNTRs genotyped from our resource ("ONT") and from multi-platform whole genome assemblies[20] ("HGSVC"), using vamos (plotted range is restricted to data points with x < 100 and y < 100 for visualisation purposes). Plots show guide lines for y = x +/− c with c = 5, 20, 50 for visualizing ranges as shown in the legend below (**a-c**). Higher c values indicate more extreme cases where one dataset reports higher RU ranges compared to the other (Note S5 and Table S31). **d-e)** Distribution of the base pair lengths and the count of RUs in the VNTR alleles genotyped by vamos on the ONT data and on

the HGSVC assemblies. We depict the distribution of two VNTR loci found in the genes *ABCA7* (chr19:1,012,105-1,014,401) in (**d**) and *PLIN4* (chr19:4,494,323-4,497,243) in (**e**), which have been associated with late-onset human disease[102,103]. For the *ABCA7* VNTR locus, alleles of a length greater than 5,720 bp (denoted through a dashed vertical line in (**d**)) are associated with late-onset disease, whereas for the *PLIN4* VNTR locus, alleles with repeat count of 40 (denoted as a dashed vertical line in (**e**)) are disease-associated. We identify a 43 RU count VNTR allele for the *PLIN4* locus in sample NA20127 (outlier denoted with an arrow), with this RU count confirmed using manual inspection (Fig. S62).

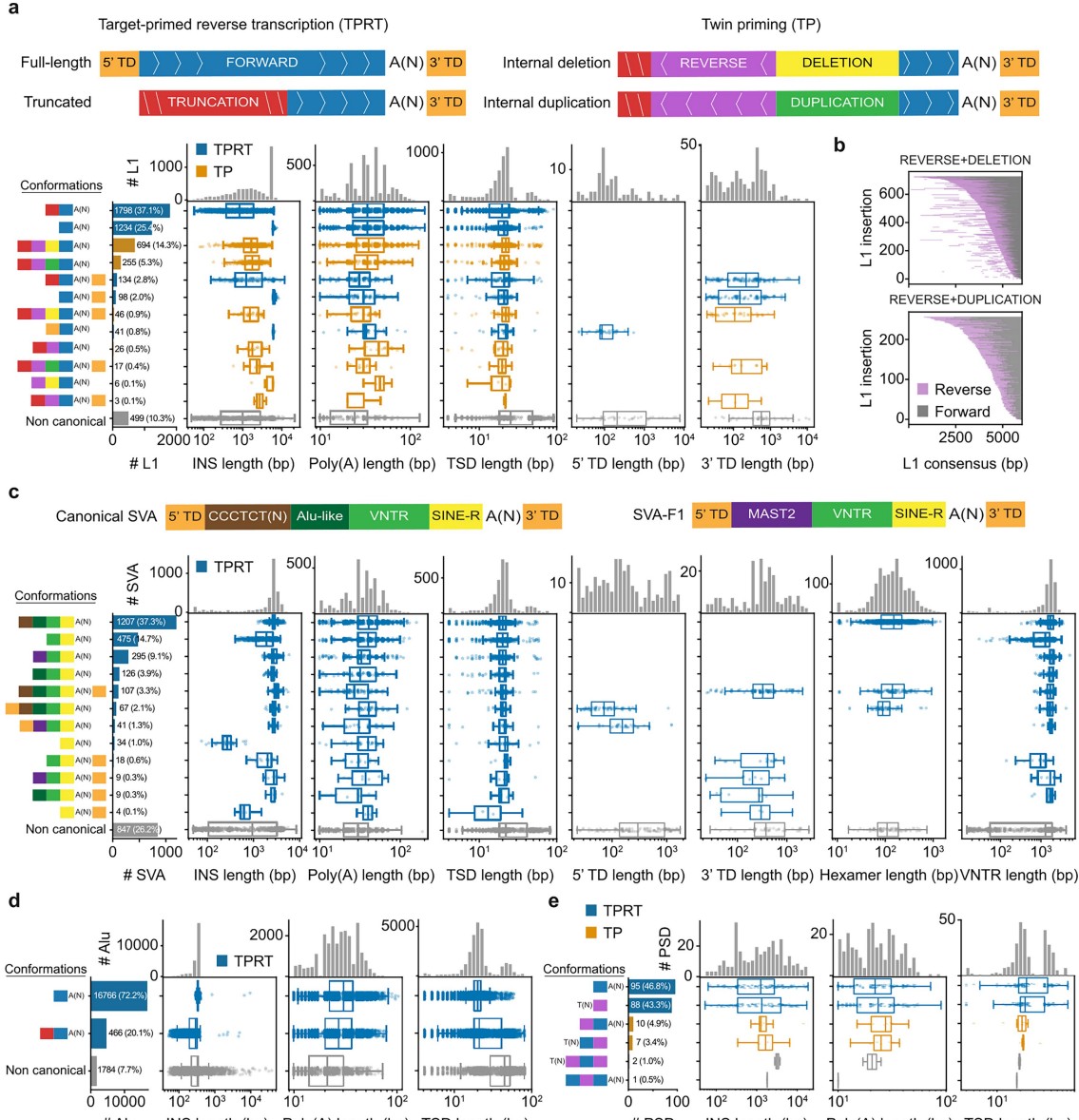

**Extended Data Fig. 5 | Sequence features for polymorphic MEIs annotated using SVAN. a)** At the top we depict a schematic representation of all possible sequence features for canonical L1 insertion conformations, shown as colored boxes. Features include poly-A tails (A(n)) and transductions (TD). Conformations are grouped based on their likely mechanism of origin: target-primed reverse transcription (TPRT) and twin priming (TP). At the bottom left, frequencies of each canonical L1 insertion conformation, where each conformation is defined by a unique combination of the sequence features shown in the schematic. Insertions with configurations inconsistent with TPRT or TP—such as those lacking poly-A tails or containing multiple internal breakpoints—are categorized as non-canonical. At the bottom right, for each L1 insertion conformation, box plots show length distributions of the full insertions and their individual sequence features. Box plots and data points are colored according to the inferred insertion mechanism. **b)** Stacked dot plots showing alignments of twin priming insertions containing deletions (top) and duplications (bottom) at internal inversion breakpoints. Alignments are colored by orientation, with magenta indicating the inverted L1 sequence. **c)** Schematic representation of sequence features observed in SVA insertions, along with frequencies of distinct SVA insertion conformations and corresponding length distributions of individual SVA features, shown using the same conventions as for L1 insertions. **d, e)** Insertion conformations (following the L1 sequence feature colour codes) and length distributions for Alu and processed pseudogene (PSD) insertions.

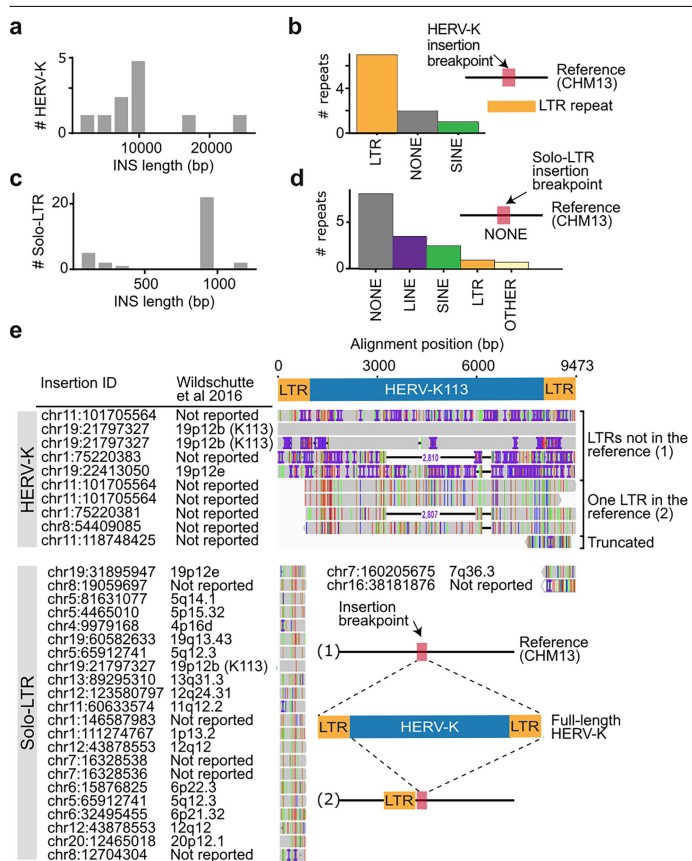

**Extended Data Fig. 6 | Polymorphic Human Endogenous Retrovirus Type K (HERV-K) insertions annotated using SVAN. a, c)** Length distributions of HERV-K and solo long terminal repeat (solo-LTR) insertions. **b, d)** Number of instance specific repetitive DNA element classes overlap the breakpoints for HERV-K and solo-LTR insertions. **e)** Alignments of HERV-K (top) and solo-LTR (bottom) insertions to the HERV-K113 provirus reference, visualized using the Integrative Genomics Viewer (IGV). Insertion coordinates relative to the CHM13 reference genome, and cytoband identifiers from previously reported[39] insertions. A schematic in the bottom right illustrates the two possible configurations for a full-length HERV-K insertion, with or without an LTR-flanking repeat present in the reference genome.

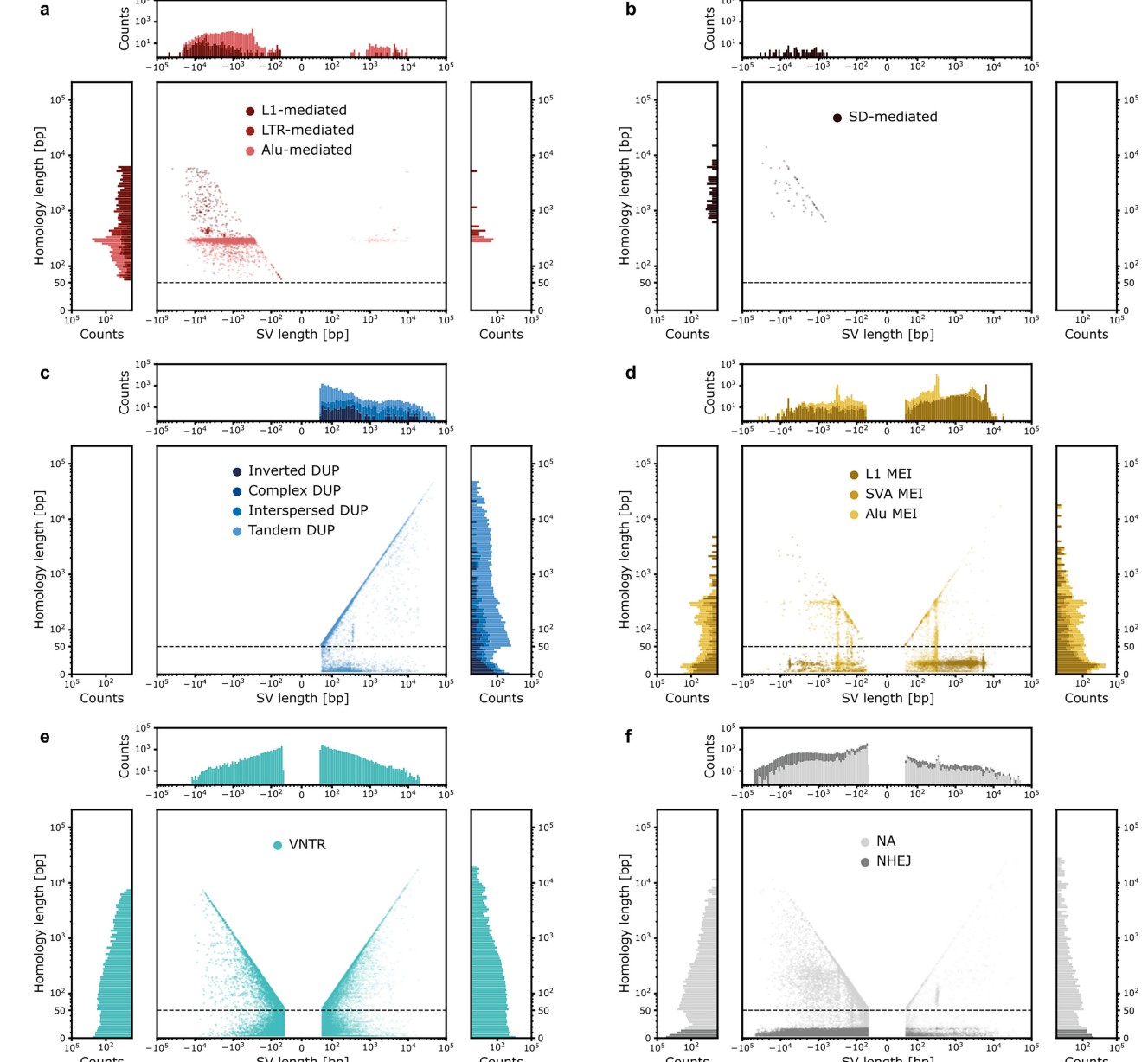

**Extended Data Fig. 7 | SV breakpoint homology and microhomology landscape separated by SV annotation.** For all SVs, homology and microhomology was determined. SVs were annotated using the SVAN pipeline as well as by leveraging flanking repeat elements and/or homologous sequence stretches. SVs were grouped into **a**) repeat-mediated SVs, **b**) segmental duplication (SD)-mediated SVs, **c**) duplications (DUP), **d**) mobile elements (MEI), **e**) VNTRs, **f**) not-classified (NA) or NHEJ-mediated SVs. The central scatter plot shows SV length versus (micro)homology length, for each group. Marginal histograms show the distribution of SV length (top) and homology length for deletions (left) and insertions (right). The axes are linear from 0 to 50 bp and log-scale afterwards, which is denoted by a dashed line. To highlight the distribution of rare SV classes, the stacking order in the marginal histograms proceeds from rare (bottom) to common classes (top). Colors correspond to those used in Fig. 5.

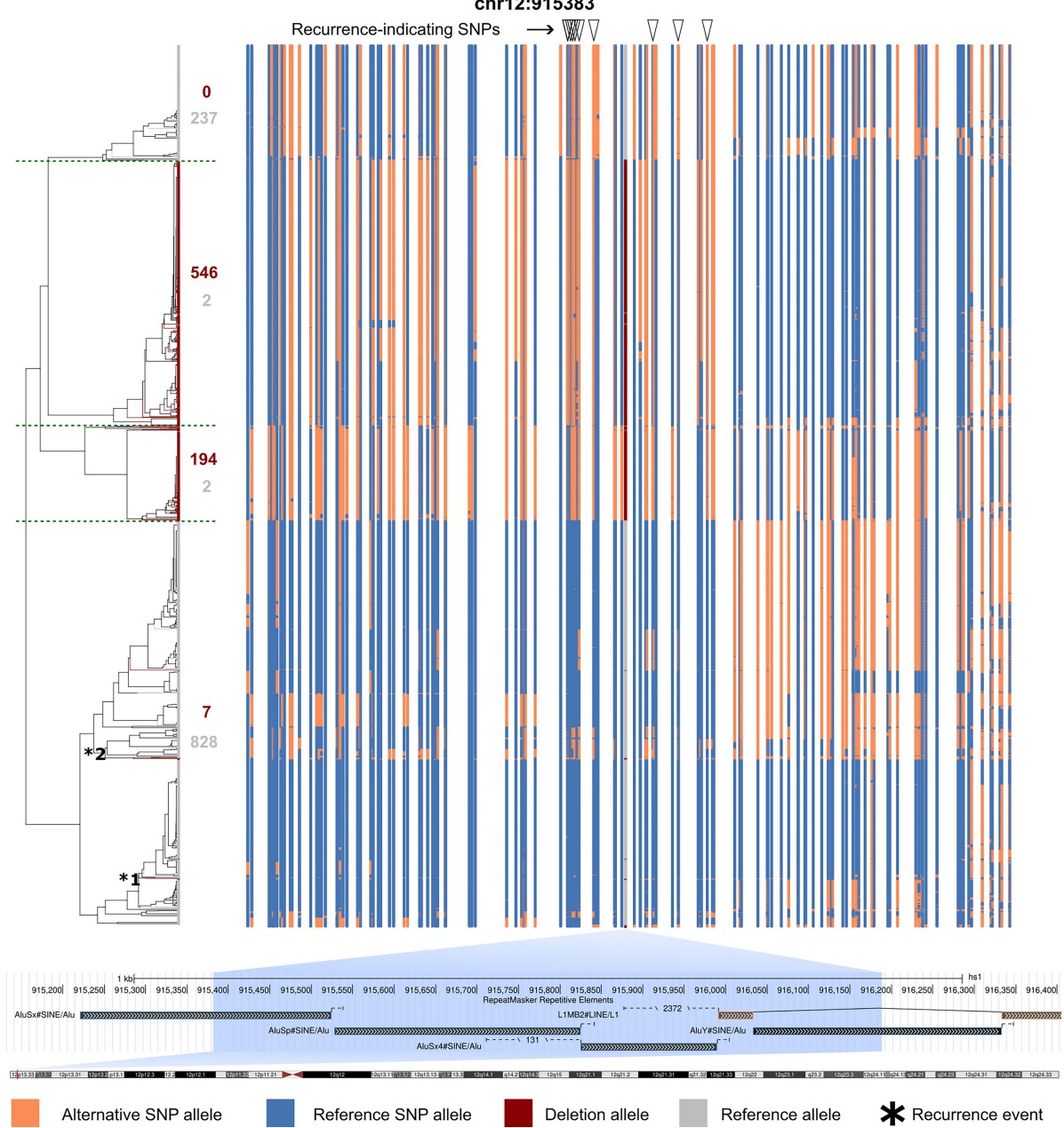

**Extended Data Fig. 8 | Inferred recurrent deletion at 12p13.3.** An inferred recurrent 806 bp deletion at 12p13.3 mediated by an AluSx-AluY pair. The figure shows the variation of haplotypes in a 100 kb window centered around the deletion and the relationship between haplotypes with (red) and without the deletion (grey). Dendrograms of haplotypes are plotted using a centroid hierarchical clustering method. Green dashed lines represent the separation of four haplotype groups shown in Fig. 5g. In each haplotype, reference and alternative alleles are shown in blue and orange, respectively. SNPs within 20 kb around the deletion showing evidence of deletion recurrence are marked by triangles at the top. Two predicted independent occurrences of the deletion event are marked as *1 and *2. The deletion genotypes of the samples involved in these events have been verified by manual inspection of the aligned sequencing reads (Fig. S59).

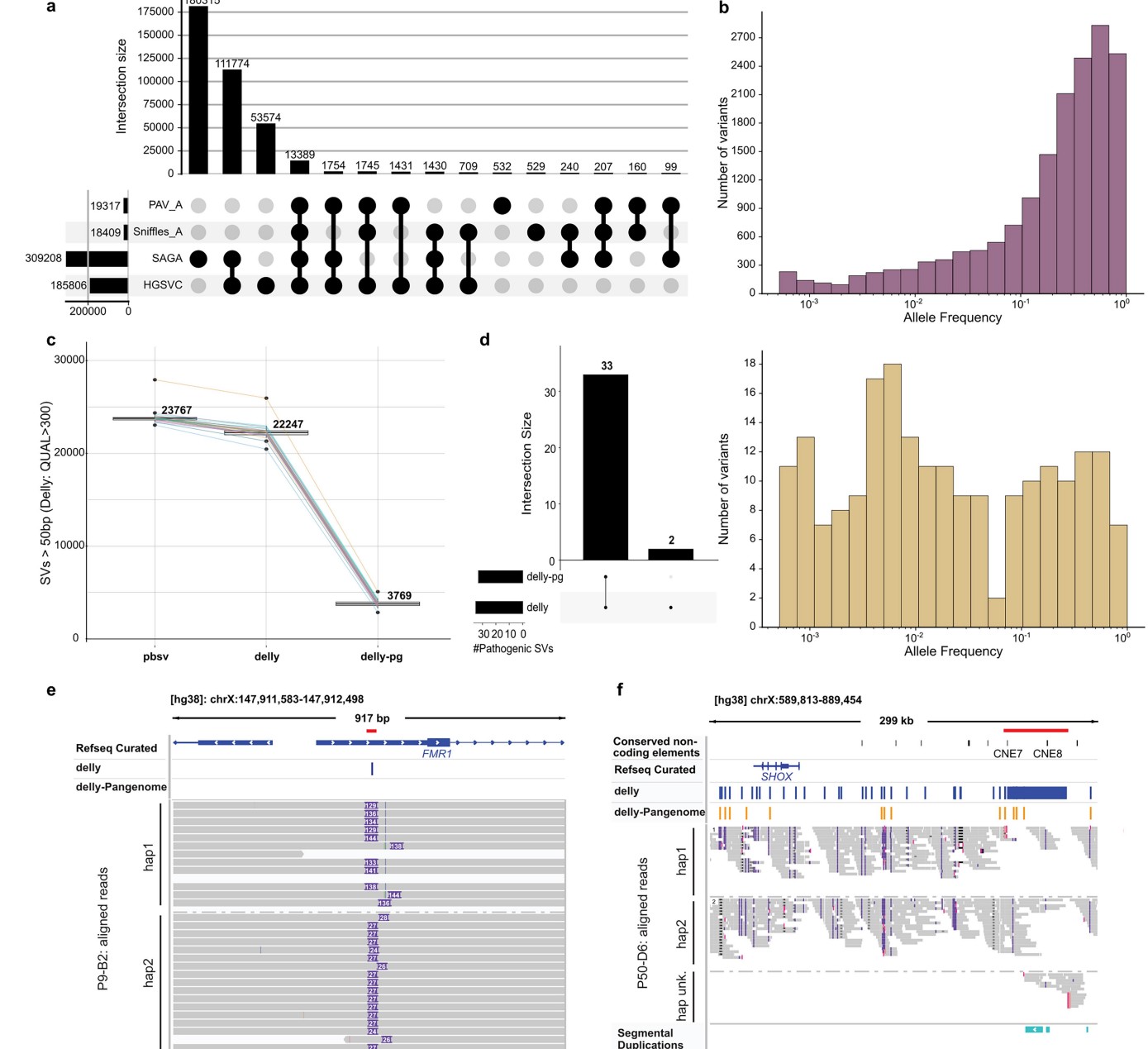

**Extended Data Fig. 9 | Patient Genome Analysis. a**) Comparison between SV callsets from 'rare disease patient A' generated by PAV and Sniffles, the phased VCF panel of HPRC_mg_44 + 966 and the HGSVC assembly-based SV callset[20], showing 160 SVs exclusive to the patient genome. **b**) Allele frequency distributions (log-scale) shown for SV alleles from our study matching those from rare disease patient A, for SVs found both in SAGA and HGSVC (top) and SVs found only in SAGA (bottom). **c**) Comparison of the number of SVs reported (1) by the pbsv caller (Note S9), (2) by DELLY using default settings, and (3) by DELLY when graph-based filtering is utilised. The median number of SVs detected in 31 rare disease patient genomes are indicated alongside the data points. The comparably high SV count in one patient sample (P1-D11; light orange) is likely attributable to population ancestry. **d**) An upset plot indicating

the number of pathogenic SVs found by DELLY, along with the number of pathogenic SVs retained in graph-based filtering mode ('delly-pg'). **e-f**) Integrative Genomics Viewer (IGV) views of the 2 validated pathogenic SVs filtered in pangenome mode. **e**) A ~140 bp insertion in an STR in *FMR1* called by DELLY (second row), but not retained in the DELLY-pangenome mode (third row). The length of this multiallelic STR varies in the population, with insert sizes beyond ~450 bp driving the fragile X syndrome[104]. **f**) A ~47 kbp deletion encompassing two regulatory conserved non-coding elements (CNEs) of *SHOX* is called by DELLY (second row), but not retained in DELLY-pangenome (third row). Variants in the *SHOX* CNEs exhibit recurrence and incomplete penetrance, consistent with the occasional presence of this SV in the general population (Note S9).

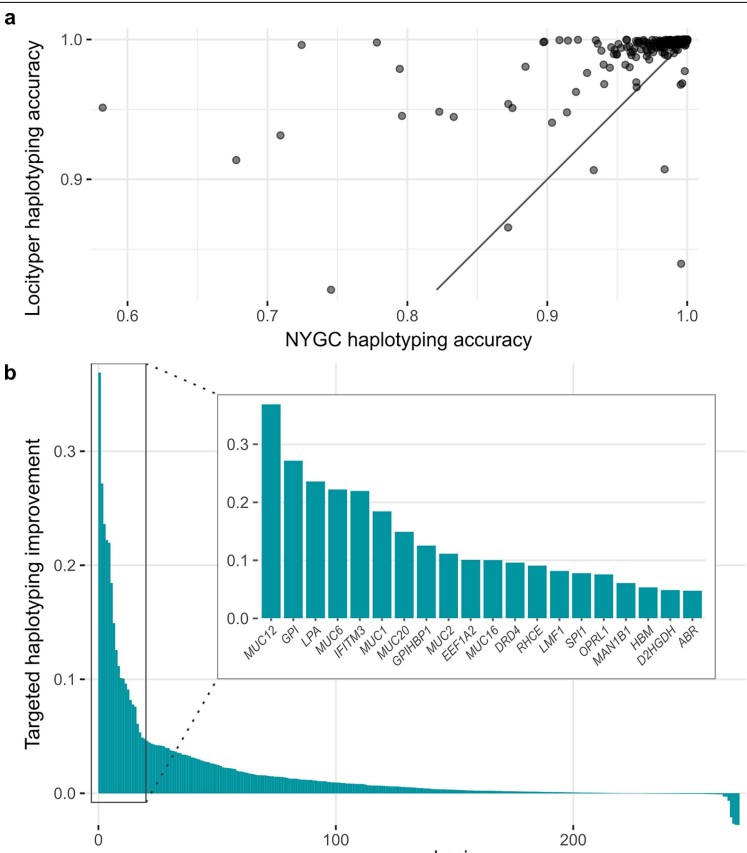

**Extended Data Fig. 10 | Targeted haplotyping accuracy based on Locityper.** Haplotyping accuracy, here explored in complex loci of the genome across 270 medically relevant loci, is calculated as sequence similarity between two predicted locus haplotypes and actual locus haplotypes, extracted from the whole genome assemblies for 1 sample from the HPRC and 8 samples from a recent multi-platform whole genome assembly study by the Human Genome Structural Variation Consortium (HGSVC)[20]. **a)** Comparison of haplotyping accuracy for high-coverage short-read[3] and intermediate-coverage ONT based haplotypes, inferred using Locityper. **b)** Improvement in haplotyping accuracy (Locityper accuracy on ONT data minus accuracy on short-read data) across 270 loci. The inset shows 20 genes with the highest improvement in haplotyping accuracy.

Tobias Rausch
Tobias Marschall
Jan O Korbel

# Reporting Summary

## Statistics

For all statistical analyses, confirm that the following items are present in the figure legend, table legend, main text, or Methods section.

| n/a | Confirmed | |
|---|---|---|
| ☐ | ☒ | The exact sample size (*n*) for each experimental group/condition, given as a discrete number and unit of measurement |
| ☐ | ☒ | A statement on whether measurements were taken from distinct samples or whether the same sample was measured repeatedly |
| ☐ | ☒ | The statistical test(s) used AND whether they are one- or two-sided *Only common tests should be described solely by name; describe more complex techniques in the Methods section.* |
| ☐ | ☒ | A description of all covariates tested |
| ☐ | ☒ | A description of any assumptions or corrections, such as tests of normality and adjustment for multiple comparisons |
| ☐ | ☒ | A full description of the statistical parameters including central tendency (e.g. means) or other basic estimates (e.g. regression coefficient) AND variation (e.g. standard deviation) or associated estimates of uncertainty (e.g. confidence intervals) |
| ☐ | ☒ | For null hypothesis testing, the test statistic (e.g. *F*, *t*, *r*) with confidence intervals, effect sizes, degrees of freedom and *P* value noted *Give P values as exact values whenever suitable.* |
| ☒ | ☐ | For Bayesian analysis, information on the choice of priors and Markov chain Monte Carlo settings |
| ☒ | ☐ | For hierarchical and complex designs, identification of the appropriate level for tests and full reporting of outcomes |
| ☐ | ☒ | Estimates of effect sizes (e.g. Cohen's *d*, Pearson's *r*), indicating how they were calculated |

*Our web collection on statistics for biologists contains articles on many of the points above.*

## Software and code

Policy information about availability of computer code

| Data collection | Sequencing was performed on a PromethION 48 using MinKNOW for data acquisition. |
|---|---|
| Data analysis | ADMIXTURE 1.3.0<br>bcalm2 2.2.3<br>bcftools 1.18<br>bedtools 2.31.1<br>BLAST 2.12.0<br>bwa-mem 0.7.17-r1188<br>delly 1.1.7<br>delly 1.2.6<br>EIGENSOFT 8.0.0<br>gaftools (commit 919bbecf51602161db7ba6e859cc5c26a7413c83)<br>gfatools<br>Giggles 1.0<br>Guppy 6.2.1<br>IGV 2.17.1<br>liftOver<br>lighter 1.1.2<br>Locityper 0.13.3<br>minigraph 0.20-r559<br>minimap2 2.2.26 |

nanopack (commit 4059a0a)
NGMLR 0.2.7
PAV 2.3.4
Porechop 0.2.4
pysamstats 1.1.2
samtools 1.18
sansa 0.2.1
SHAPEIT5 5.1.1
sniffles 2.0.7
SURVIVOR 1.0.7
SVAN
SVarp (commit 0acba75ebdfdd292d57e1bd133d852f6371ab677)
TRF 4.04
truvari 4.1.0
vamos 2.1.3
vcftools 0.1.16
vep 111
whatshap 2.0
yak 0.1

For manuscripts utilizing custom algorithms or software that are central to the research but not yet described in published literature, software must be made available to editors and reviewers. We strongly encourage code deposition in a community repository (e.g. GitHub). See the Nature Portfolio guidelines for submitting code & software for further information.

## Data

Policy information about availability of data

All manuscripts must include a data availability statement. This statement should provide the following information, where applicable:
- Accession codes, unique identifiers, or web links for publicly available datasets
- A description of any restrictions on data availability
- For clinical datasets or third party data, please ensure that the statement adheres to our policy

Our open data resource is fully available to the community for download at the International Genome Sample Resource (IGSR), at the following IGSR file transfer protocol (FTP) repository: https://ftp.1000genomes.ebi.ac.uk/vol1/ftp/data_collections/1KG_ONT_VIENNA/. This repository contains the graph and linear reference genomes, alignments, input SV callsets, the augmented genome graph, genotyped and phased structural variants as well as auxiliary data used for evaluating SV callset accuracy. For archiving purposes only, we will also mirror the complete dataset in the European Nucleotide Archive before the paper is accepted for publication.

## Research involving human participants, their data, or biological material

Policy information about studies with human participants or human data. See also policy information about sex, gender (identity/presentation), and sexual orientation and race, ethnicity and racism.

| Reporting on sex and gender | We report sex of the donor as provided in the metadata available from the Coriell Institute. |
|---|---|
| Reporting on race, ethnicity, or other socially relevant groupings | We report the ancestry of the donor as provided in the metadata available from the Coriell Institute and adhere to the "Guidelines when Referring to Populations" available at: https://catalog.coriell.org/1/NHGRI/About/Guidelines-for-Referring-to-Populations. |
| Population characteristics | Age, gender, family relationship and self-identified geographic origin were available from the Coriell Institute. Previous 1000 Genomes Project studies provided short-read genotyping information. |
| Recruitment | Performed as part of the 1000 Genomes Project. |
| Ethics oversight | Samples were sourced from the 1000 Genomes Project sample collection, which allows unrestricted public data access, data sharing and reuse |

Note that full information on the approval of the study protocol must also be provided in the manuscript.

## Field-specific reporting

Please select the one below that is the best fit for your research. If you are not sure, read the appropriate sections before making your selection.

☒ Life sciences　　☐ Behavioural & social sciences　　☐ Ecological, evolutionary & environmental sciences

For a reference copy of the document with all sections, see nature.com/documents/nr-reporting-summary-flat.pdf

# Life sciences study design

All studies must disclose on these points even when the disclosure is negative.

| | |
|---|---|
| Sample size | Decided with respect to available funding. |
| Data exclusions | We excluded 8 samples due to cross-contamination during sequencing. |
| Replication | Experiments were computational, therefore replication is not applicable. For reproducibility all of the datasets and codes/workflows are publicaly available. |
| Randomization | The samples were not allocated into different experimental groups. |
| Blinding | Comparison between experimental groups was not the purpose of this study, therefore blinding was not necessary. |

# Behavioural & social sciences study design

All studies must disclose on these points even when the disclosure is negative.

| | |
|---|---|
| Study description | *Briefly describe the study type including whether data are quantitative, qualitative, or mixed-methods (e.g. qualitative cross-sectional, quantitative experimental, mixed-methods case study).* |
| Research sample | *State the research sample (e.g. Harvard university undergraduates, villagers in rural India) and provide relevant demographic information (e.g. age, sex) and indicate whether the sample is representative. Provide a rationale for the study sample chosen. For studies involving existing datasets, please describe the dataset and source.* |
| Sampling strategy | *Describe the sampling procedure (e.g. random, snowball, stratified, convenience). Describe the statistical methods that were used to predetermine sample size OR if no sample-size calculation was performed, describe how sample sizes were chosen and provide a rationale for why these sample sizes are sufficient. For qualitative data, please indicate whether data saturation was considered, and what criteria were used to decide that no further sampling was needed.* |
| Data collection | *Provide details about the data collection procedure, including the instruments or devices used to record the data (e.g. pen and paper, computer, eye tracker, video or audio equipment) whether anyone was present besides the participant(s) and the researcher, and whether the researcher was blind to experimental condition and/or the study hypothesis during data collection.* |
| Timing | *Indicate the start and stop dates of data collection. If there is a gap between collection periods, state the dates for each sample cohort.* |
| Data exclusions | *If no data were excluded from the analyses, state so OR if data were excluded, provide the exact number of exclusions and the rationale behind them, indicating whether exclusion criteria were pre-established.* |
| Non-participation | *State how many participants dropped out/declined participation and the reason(s) given OR provide response rate OR state that no participants dropped out/declined participation.* |
| Randomization | *If participants were not allocated into experimental groups, state so OR describe how participants were allocated to groups, and if allocation was not random, describe how covariates were controlled.* |

# Ecological, evolutionary & environmental sciences study design

All studies must disclose on these points even when the disclosure is negative.

| | |
|---|---|
| Study description | *Briefly describe the study. For quantitative data include treatment factors and interactions, design structure (e.g. factorial, nested, hierarchical), nature and number of experimental units and replicates.* |
| Research sample | *Describe the research sample (e.g. a group of tagged Passer domesticus, all Stenocereus thurberi within Organ Pipe Cactus National Monument), and provide a rationale for the sample choice. When relevant, describe the organism taxa, source, sex, age range and any manipulations. State what population the sample is meant to represent when applicable. For studies involving existing datasets, describe the data and its source.* |
| Sampling strategy | *Note the sampling procedure. Describe the statistical methods that were used to predetermine sample size OR if no sample-size calculation was performed, describe how sample sizes were chosen and provide a rationale for why these sample sizes are sufficient.* |
| Data collection | *Describe the data collection procedure, including who recorded the data and how.* |
| Timing and spatial scale | *Indicate the start and stop dates of data collection, noting the frequency and periodicity of sampling and providing a rationale for these choices. If there is a gap between collection periods, state the dates for each sample cohort. Specify the spatial scale from which the data are taken* |

| | |
|---|---|
| Data exclusions | *If no data were excluded from the analyses, state so OR if data were excluded, describe the exclusions and the rationale behind them, indicating whether exclusion criteria were pre-established.* |
| Reproducibility | *Describe the measures taken to verify the reproducibility of experimental findings. For each experiment, note whether any attempts to repeat the experiment failed OR state that all attempts to repeat the experiment were successful.* |
| Randomization | *Describe how samples/organisms/participants were allocated into groups. If allocation was not random, describe how covariates were controlled. If this is not relevant to your study, explain why.* |
| Blinding | *Describe the extent of blinding used during data acquisition and analysis. If blinding was not possible, describe why OR explain why blinding was not relevant to your study.* |

Did the study involve field work? ☐ Yes ☐ No

## Field work, collection and transport

| | |
|---|---|
| Field conditions | *Describe the study conditions for field work, providing relevant parameters (e.g. temperature, rainfall).* |
| Location | *State the location of the sampling or experiment, providing relevant parameters (e.g. latitude and longitude, elevation, water depth).* |
| Access & import/export | *Describe the efforts you have made to access habitats and to collect and import/export your samples in a responsible manner and in compliance with local, national and international laws, noting any permits that were obtained (give the name of the issuing authority, the date of issue, and any identifying information).* |
| Disturbance | *Describe any disturbance caused by the study and how it was minimized.* |

# Reporting for specific materials, systems and methods

We require information from authors about some types of materials, experimental systems and methods used in many studies. Here, indicate whether each material, system or method listed is relevant to your study. If you are not sure if a list item applies to your research, read the appropriate section before selecting a response.

### Materials & experimental systems

| n/a | Involved in the study |
|---|---|
| ☒ ☐ | Antibodies |
| ☒ ☐ | Eukaryotic cell lines |
| ☒ ☐ | Palaeontology and archaeology |
| ☒ ☐ | Animals and other organisms |
| ☒ ☐ | Clinical data |
| ☒ ☐ | Dual use research of concern |
| ☒ ☐ | Plants |

### Methods

| n/a | Involved in the study |
|---|---|
| ☒ ☐ | ChIP-seq |
| ☒ ☐ | Flow cytometry |
| ☒ ☐ | MRI-based neuroimaging |

## Antibodies

| | |
|---|---|
| Antibodies used | *Describe all antibodies used in the study; as applicable, provide supplier name, catalog number, clone name, and lot number.* |
| Validation | *Describe the validation of each primary antibody for the species and application, noting any validation statements on the manufacturer's website, relevant citations, antibody profiles in online databases, or data provided in the manuscript.* |

## Eukaryotic cell lines

Policy information about cell lines and Sex and Gender in Research

| | |
|---|---|
| Cell line source(s) | *State the source of each cell line used and the sex of all primary cell lines and cells derived from human participants or vertebrate models.* |
| Authentication | *Describe the authentication procedures for each cell line used OR declare that none of the cell lines used were authenticated.* |
| Mycoplasma contamination | *Confirm that all cell lines tested negative for mycoplasma contamination OR describe the results of the testing for mycoplasma contamination OR declare that the cell lines were not tested for mycoplasma contamination.* |
| Commonly misidentified lines (See ICLAC register) | *Name any commonly misidentified cell lines used in the study and provide a rationale for their use.* |

# Palaeontology and Archaeology

Specimen provenance

*Provide provenance information for specimens and describe permits that were obtained for the work (including the name of the issuing authority, the date of issue, and any identifying information). Permits should encompass collection and, where applicable, export.*

Specimen deposition

*Indicate where the specimens have been deposited to permit free access by other researchers.*

Dating methods

*If new dates are provided, describe how they were obtained (e.g. collection, storage, sample pretreatment and measurement), where they were obtained (i.e. lab name), the calibration program and the protocol for quality assurance OR state that no new dates are provided.*

☐ Tick this box to confirm that the raw and calibrated dates are available in the paper or in Supplementary Information.

Ethics oversight

*Identify the organization(s) that approved or provided guidance on the study protocol, OR state that no ethical approval or guidance was required and explain why not.*

Note that full information on the approval of the study protocol must also be provided in the manuscript.

# Animals and other research organisms

Policy information about studies involving animals; ARRIVE guidelines recommended for reporting animal research, and Sex and Gender in Research

Laboratory animals

*For laboratory animals, report species, strain and age OR state that the study did not involve laboratory animals.*

Wild animals

*Provide details on animals observed in or captured in the field; report species and age where possible. Describe how animals were caught and transported and what happened to captive animals after the study (if killed, explain why and describe method; if released, say where and when) OR state that the study did not involve wild animals.*

Reporting on sex

*Indicate if findings apply to only one sex; describe whether sex was considered in study design, methods used for assigning sex. Provide data disaggregated for sex where this information has been collected in the source data as appropriate; provide overall numbers in this Reporting Summary. Please state if this information has not been collected. Report sex-based analyses where performed, justify reasons for lack of sex-based analysis.*

Field-collected samples

*For laboratory work with field-collected samples, describe all relevant parameters such as housing, maintenance, temperature, photoperiod and end-of-experiment protocol OR state that the study did not involve samples collected from the field.*

Ethics oversight

*Identify the organization(s) that approved or provided guidance on the study protocol, OR state that no ethical approval or guidance was required and explain why not.*

Note that full information on the approval of the study protocol must also be provided in the manuscript.

# Clinical data

Policy information about clinical studies

All manuscripts should comply with the ICMJE guidelines for publication of clinical research and a completed CONSORT checklist must be included with all submissions.

Clinical trial registration

*Provide the trial registration number from ClinicalTrials.gov or an equivalent agency.*

Study protocol

*Note where the full trial protocol can be accessed OR if not available, explain why.*

Data collection

*Describe the settings and locales of data collection, noting the time periods of recruitment and data collection.*

Outcomes

*Describe how you pre-defined primary and secondary outcome measures and how you assessed these measures.*

# Dual use research of concern

Policy information about dual use research of concern

## Hazards

Could the accidental, deliberate or reckless misuse of agents or technologies generated in the work, or the application of information presented in the manuscript, pose a threat to:

| No | Yes | |
|---|---|---|
| ☐ | ☐ | Public health |
| ☐ | ☐ | National security |
| ☐ | ☐ | Crops and/or livestock |
| ☐ | ☐ | Ecosystems |
| ☐ | ☐ | Any other significant area |

## Experiments of concern

Does the work involve any of these experiments of concern:

| No | Yes | |
|---|---|---|
| ☐ | ☐ | Demonstrate how to render a vaccine ineffective |
| ☐ | ☐ | Confer resistance to therapeutically useful antibiotics or antiviral agents |
| ☐ | ☐ | Enhance the virulence of a pathogen or render a nonpathogen virulent |
| ☐ | ☐ | Increase transmissibility of a pathogen |
| ☐ | ☐ | Alter the host range of a pathogen |
| ☐ | ☐ | Enable evasion of diagnostic/detection modalities |
| ☐ | ☐ | Enable the weaponization of a biological agent or toxin |
| ☐ | ☐ | Any other potentially harmful combination of experiments and agents |

# Plants

Seed stocks
*Report on the source of all seed stocks or other plant material used. If applicable, state the seed stock centre and catalogue number. If plant specimens were collected from the field, describe the collection location, date and sampling procedures.*

Novel plant genotypes
*Describe the methods by which all novel plant genotypes were produced. This includes those generated by transgenic approaches, gene editing, chemical/radiation-based mutagenesis and hybridization. For transgenic lines, describe the transformation method, the number of independent lines analyzed and the generation upon which experiments were performed. For gene-edited lines, describe the editor used, the endogenous sequence targeted for editing, the targeting guide RNA sequence (if applicable) and how the editor was applied.*

Authentication
*Describe any authentication procedures for each seed stock used or novel genotype generated. Describe any experiments used to assess the effect of a mutation and, where applicable, how potential secondary effects (e.g. second site T-DNA insertions, mosiacism, off-target gene editing) were examined.*

# ChIP-seq

## Data deposition

☐ Confirm that both raw and final processed data have been deposited in a public database such as GEO.

☐ Confirm that you have deposited or provided access to graph files (e.g. BED files) for the called peaks.

Data access links
*May remain private before publication.*
*For "Initial submission" or "Revised version" documents, provide reviewer access links. For your "Final submission" document, provide a link to the deposited data.*

Files in database submission
*Provide a list of all files available in the database submission.*

Genome browser session
(e.g. UCSC)
*Provide a link to an anonymized genome browser session for "Initial submission" and "Revised version" documents only, to enable peer review. Write "no longer applicable" for "Final submission" documents.*

## Methodology

Replicates
*Describe the experimental replicates, specifying number, type and replicate agreement.*

Sequencing depth
*Describe the sequencing depth for each experiment, providing the total number of reads, uniquely mapped reads, length of reads and whether they were paired- or single-end.*

Antibodies
*Describe the antibodies used for the ChIP-seq experiments; as applicable, provide supplier name, catalog number, clone name, and lot number.*

Peak calling parameters
*Specify the command line program and parameters used for read mapping and peak calling, including the ChIP, control and index files used.*

| Data quality | *Describe the methods used to ensure data quality in full detail, including how many peaks are at FDR 5% and above 5-fold enrichment.* |
|---|---|
| Software | *Describe the software used to collect and analyze the ChIP-seq data. For custom code that has been deposited into a community repository, provide accession details.* |

## Flow Cytometry

### Plots

Confirm that:

☐ The axis labels state the marker and fluorochrome used (e.g. CD4-FITC).

☐ The axis scales are clearly visible. Include numbers along axes only for bottom left plot of group (a 'group' is an analysis of identical markers).

☐ All plots are contour plots with outliers or pseudocolor plots.

☐ A numerical value for number of cells or percentage (with statistics) is provided.

### Methodology

| Sample preparation | *Describe the sample preparation, detailing the biological source of the cells and any tissue processing steps used.* |
|---|---|
| Instrument | *Identify the instrument used for data collection, specifying make and model number.* |
| Software | *Describe the software used to collect and analyze the flow cytometry data. For custom code that has been deposited into a community repository, provide accession details.* |
| Cell population abundance | *Describe the abundance of the relevant cell populations within post-sort fractions, providing details on the purity of the samples and how it was determined.* |
| Gating strategy | *Describe the gating strategy used for all relevant experiments, specifying the preliminary FSC/SSC gates of the starting cell population, indicating where boundaries between "positive" and "negative" staining cell populations are defined.* |

☐ Tick this box to confirm that a figure exemplifying the gating strategy is provided in the Supplementary Information.

## Magnetic resonance imaging

### Experimental design

| Design type | *Indicate task or resting state; event-related or block design.* |
|---|---|
| Design specifications | *Specify the number of blocks, trials or experimental units per session and/or subject, and specify the length of each trial or block (if trials are blocked) and interval between trials.* |
| Behavioral performance measures | *State number and/or type of variables recorded (e.g. correct button press, response time) and what statistics were used to establish that the subjects were performing the task as expected (e.g. mean, range, and/or standard deviation across subjects).* |

### Acquisition

| Imaging type(s) | *Specify: functional, structural, diffusion, perfusion.* |
|---|---|
| Field strength | *Specify in Tesla* |
| Sequence & imaging parameters | *Specify the pulse sequence type (gradient echo, spin echo, etc.), imaging type (EPI, spiral, etc.), field of view, matrix size, slice thickness, orientation and TE/TR/flip angle.* |
| Area of acquisition | *State whether a whole brain scan was used OR define the area of acquisition, describing how the region was determined.* |

Diffusion MRI    ☐ Used    ☐ Not used

### Preprocessing

| Preprocessing software | *Provide detail on software version and revision number and on specific parameters (model/functions, brain extraction, segmentation, smoothing kernel size, etc.).* |
|---|---|
| Normalization | *If data were normalized/standardized, describe the approach(es): specify linear or non-linear and define image types used for transformation OR indicate that data were not normalized and explain rationale for lack of normalization.* |

| | |
|---|---|
| Normalization template | *Describe the template used for normalization/transformation, specifying subject space or group standardized space (e.g. original Talairach, MNI305, ICBM152) OR indicate that the data were not normalized.* |
| Noise and artifact removal | *Describe your procedure(s) for artifact and structured noise removal, specifying motion parameters, tissue signals and physiological signals (heart rate, respiration).* |
| Volume censoring | *Define your software and/or method and criteria for volume censoring, and state the extent of such censoring.* |

## Statistical modeling & inference

| | |
|---|---|
| Model type and settings | *Specify type (mass univariate, multivariate, RSA, predictive, etc.) and describe essential details of the model at the first and second levels (e.g. fixed, random or mixed effects; drift or auto-correlation).* |
| Effect(s) tested | *Define precise effect in terms of the task or stimulus conditions instead of psychological concepts and indicate whether ANOVA or factorial designs were used.* |

Specify type of analysis: ☐ Whole brain   ☐ ROI-based   ☐ Both

| | |
|---|---|
| Statistic type for inference<br><br>(See Eklund et al. 2016) | *Specify voxel-wise or cluster-wise and report all relevant parameters for cluster-wise methods.* |
| Correction | *Describe the type of correction and how it is obtained for multiple comparisons (e.g. FWE, FDR, permutation or Monte Carlo).* |

## Models & analysis

| n/a | Involved in the study |
|---|---|
| ☐ | ☐ Functional and/or effective connectivity |
| ☐ | ☐ Graph analysis |
| ☐ | ☐ Multivariate modeling or predictive analysis |

| | |
|---|---|
| Functional and/or effective connectivity | *Report the measures of dependence used and the model details (e.g. Pearson correlation, partial correlation, mutual information).* |
| Graph analysis | *Report the dependent variable and connectivity measure, specifying weighted graph or binarized graph, subject- or group-level, and the global and/or node summaries used (e.g. clustering coefficient, efficiency, etc.).* |
| Multivariate modeling and predictive analysis | *Specify independent variables, features extraction and dimension reduction, model, training and evaluation metrics.* |

