## [Peer Review File · Nature]

Structural Variation in 1,019 Diverse Humans based on Long-Read Sequencing

Corresponding Author: Dr Tobias Marschall

Version 1:

Reviewer comments:

Referee #1

(Remarks to the Author)

This manuscript reports on medium coverage nanopore sequencing from ~1000 1KG samples. Using these data the authors identify ~170K structural variants (SVs) and report on several of the properties of these SVs. The efforts outlined dovetail with several other long read sequencing and genome assembly efforts, however the major contribution of this work in my opinion is the large number of individuals. Nevertheless, the sequencing depth and quality reported in this resource are of substantially lower quality than other efforts highlighting the tradeoff made towards sequencing more individuals. I think this work thus has potential, but the authors need to make more of a case for their approach and why it is useful in the current era of complete genome assembly, which is substantially better at resolving SVs, particularly in some specific regions of the genome. While the resource generated is commendable for its scope, stronger motivations are needed to justify the low coverage and short read lengths. The manuscript would be also substantially improved if the authors directly demonstrated/highlighted the utility of these data and made specific efforts to highlight novel biology or analyses that these data empower. To be sure, I think this resource is well constructed and important, but more up-front discussion of the limitations of the dataset are needed.

Specific comments:

This manuscript is arriving at an interesting time in that long read assemblies are indeed enabling the discovery and genotyping of previously intractable SVs. However, the largest gains are being made from haplotype-resolved, complete genome assemblies, some of which the authors of this manuscript are involved with (e.g. HPRC Nature 2023, Ebert et al, Science 2021). The authors make the argument that such assembly efforts are “hampered by small sample sizes and the lack of long read sequencing panels of normal individuals from diverse panels.” However, there are problems with this argument. First, these other studies *are* made up of diverse, normal individuals. Secondly, while the sample sizes are indeed smaller, the high quality of the resultant assemblies appears to more than make up for this – particularly when compared to the approach of this manuscript in which both the coverage and N50 read lengths are not particularly high. For instance, while this study identifies 167K SVs in ~1000 samples, Ebert et al identified 107K SVs in just 32 haplotype resolved genome assemblies (31-fold fewer individuals). Furthermore, Gustafson et al (10.1101/2024.03.05.24303792) performed a very similar analysis sequencing 10-fold fewer (n=100) 1000 genomes samples. However, presumably due to the higher sequencing depth (>2X higher) and read length (>2X longer) they were able to identify a nearly as many SVs (~150K). Indeed, Gustafson figure 3D is essentially identical to 2a in this manuscript. Thus, the authors need to make the case for why their approach is preferred / complementary and discuss the tradeoffs.

Comparisons to short read based discovery are relevant but need only be mentioned in passing – it is well understood that such methods are underpowered. Similarly, SNP chip-based validation seems underpowered (see below).

Related to above, a deep comparison / contrast is needed showing the utility of ONT resequencing as function of coverage, read length N50, and sample size (see below comment).

The resource quality assessment needs to compare this approach to assembly-based SV detection. Both approaches are being pushed extensively (as noted in the discussion) and so there is a strong interest in this.

The FDR seems on the high side (8-11% for deletions and insertions). Comparing how this would be augmented with

different coverage / read lengths would be beneficial.

“Unlike prior population-scale surveys of 1kGP samples based on short read sequencing^{3,4,30}, our resource captures SVs along their whole size spectrum, which includes 50 to several hundred basepair size SVs that are poorly captured^{6,7} by Illumina reads.” The statement “along their whole size spectrum” is not correct. This statement should be modified to clarify that this approach indeed can capture additional signal, but is low powered in many ways as well (e.g. large deletions, SDs, inversions, etc).

There is a large peak in 1.5kb insertions found in short reads by not the long reads (Fig 2b pink peak). What is this and why is it not captured by the long read data?

Is the Fst statistic in Fig 2g only reported for biallelic SVs?

The analysis of larger events being rarer irrespective of being dups or dels should be more rigorously performed using the full allele frequency distribution (Fig S27) opposed to arbitrary bins.

The complex SV calling approach utilizing both CHM13 and the HPRC pangenome graph is well done, and a huge effort to do all the phasing. However, ultimately the deletions vs insertions are only with respect to a single reference. The call set would be improved by adding information about the ancestral (e.g. chimp) allele for population genetics.

This approach (medium coverage, shorter read length nanopore) seems to work in some regions of the genome – “high confidence regions,” but less so in so-called “low confidence” regions... this should be discussed extensively. What are the difference between these regions and how are assembly approaches better at getting to the “low confidence regions.” Justification is needed for the utility of this low-coverage ONT approach, which is undoubtedly more cost effective but is poorly powered in “low confidence regions.” Discussion take into account the future directions envisioned by the authors -> e.g. would this approach even discover a disease causing SD mediated microdeletion?

In general, many of the insights about SV distribution and formation mechanisms do not strike me as particularly novel, and I think the manuscript could be substantially shortened. There are some exceptions to this, for instance the locus-specific transduction tendencies is quite interesting. However, the majority of the insights regarding SV formation, distribution, and mechanisms are not novel, and have even been reported in very similar fashion in previous works by these same authors (e.g. 5' vs 3' transductions figure is identical to that shown in Ebert 2021). Some care should be taken to highlight what is genuinely new.

The dataset should include extensive methylation information. Were there any interesting methylation results?

“Furthermore, analyses of SV breakpoint junctions suggest a continuum of homology-mediated rearrangement processes are integral to SV formation, and highlight evidence for SV recurrence involving repeat sequences.” I find this statement problematic. What does “continuum” mean? It seems to imply that there are many mechanisms at work – that is well known. At the very least the authors should simply say this – as it’s written it’s quite vague. Furthermore, the evidence for SV recurrence is very limited and focused on just a single example.

The analyses of recurrence needs to be extended and clarified, or alternatively downplayed in the abstract as it’s just a single example. I think more could be done by the authors to find additional examples if they wish to highlight these interesting cases.

“In the immediate future, the use of whole genome assembly in isolation is hence deemed less likely to significantly contribute to the advancement of precision medicine by facilitating the generation of comprehensive genetic variant data at a population-scale.” I don’t think this is true – however, regardless the authors need to justify this statement.

Much is made in the manuscript about data availability and this being an “open-access resource.” However, many of the links to code and pipelines do not work because the repositories are still private.

(Remarks on code availability)

I was able to look at some of the code, but much of it was in private github repositories and needs to be made public.

Referee #2

(Remarks to the Author)

In their manuscript “Long-read sequencing and structural variant characterization in 1,019 samples from the 1000 Genomes Project” Schloissnig, Pani et al. provide intermediate coverage resource of 1,019 long-read ONT genomes from 26 human populations from the 1000 Genomes Project.

In this tremendous effort the authors integrated linear and graph-based approaches for SV analysis via pangenome graph-augmentation and provide: Firstly, a very valuable resource of data and methods; secondly, insights into SV mechanisms/molecular origins and biology; thirdly, novel population SV patterns. The massive SV catalogue with basepair-resolved SV breakpoints represents a very significant increase for 1kGP samples (and in general), with particular insights into difficult SV types, such as VNTR-SVs, MEI transductions, and inversions.

Overall this is an intriguing study, and a well written manuscript with great value to the genomics community.

The hopefully constructive feedback points below may in part further improve the manuscript and resource, but in part may

even exceed beyond the scope of this work.

Major points of attention:

- Missing comparison to conventional long-read SV methods:

The augmented long-read graph is shown to outperform a) its non-augmented counterpart, and b) previous 1kGP studies. Regarding b), previous studies are based on short reads, so the performance improvement is encouraging but not too surprising. A fairer comparison, in my opinion, would be to compare the graph-based approach to a best-practices approach with conventional long-read sequencing. Are all variants that were 'fed' into the graph for augmentation (Step 2/3 of SAGA) later recovered in the SV calling? Does the graph-based approach 'out-of-the-box' surpass e.g. specialized (complex) SV, MEI, VNTR, callers?

- Understandably, the authors focus on the impressive successes of the model, but I would be equally interested to learn more about the approaches' limitations. For example, the authors note a lower sensitivity in large deletions, and the overall moderate size of SVs (Fig 2b, 3a,b) seems to support this trend. Also the SV-Mb sum per individual seems lower than expected from other studies. It makes perfect sense that SVs of 100's of kbp - especially the structurally more diverse ones - may not yet be captured by the graph, especially very rare events. I would value a little extra discussion of this (presumed?) limitation. I do absolutely not expect the authors to solve these problems, but I would appreciate it if they could probe the 'loose ends' of the graph's capabilities more, to help guide the field towards the next challenges; as these may be important for the suggested disease studies.

- In line with the previous point, a comparison for SV completeness of a subset of samples would be beneficial:

o On average 23969 SVs for AFR samples (19,297 for non-AFR) seems lower than previous studies suggested. Is it undercalling here or overcalling of previous work

o For a subset of samples orthogonal data (platinum genome pedigree, HG002) should be available for such comparison.

o Can the authors down sample coverage data, to estimate the effect of 5x, 10x, 15x, 20x coverage for some of the samples with higher coverage?

- As add on to paragraph page 6 (and Tables 4-6): With the availability of trio data in 1kGP have the authors systematically attempted de novo SV mutation rates per SV class? This may be important in the connect of long-read trios for rare disease studies. Or does use of cell line material hamper this capability. If so: how many of the SVs may be cell line artifacts, and is there an indication for somatic origin for certain SVs/SV classes? Or are other variant types e.g. VNTR-SVs somatically instable?

- In which form or ideally database will the data be available to the scientific community?

- SVAN is quite central to the downstream analysis, but the Github page is missing (<https://github.com/REPPIO/SVAN>) so it was not possible to evaluate.

Minor points of attention:

- Page 14: overall sum of SV numbers and sum of SV sizes per individual would be beneficial to add.

- Relation to similar/potentially overlapping work, as shown in recent preprint: Gustafson et al.

<https://www.medrxiv.org/content/10.1101/2024.03.05.24303792v1>

- Comparison to Beyter et al Nat Genet (2021): <https://www.nature.com/articles/s41588-021-00865-4>. The authors identify 22,636 SVs by applying multiple SV tools. Which variant types do/do not overlap?

- Previously implied strengths of ONT not yet utilized in this dataset and not yet captured in this manuscript:

o Short tandem repeat expansions as a separate variant class

o DNA modifications, such as DNA methylation. Do some SV classes lead to methylation alteration in cis?

- Previous work implied that SVs mediated by DNA-repair mechanisms, may lead to increased number of de novo SNVs near SV breakpoints. Have the authors explored this for respective SV classes?

- In this dataset an average of 93.6% of CHM13 are covered at least 5-fold. It may be interesting for the reader to understand which parts are consistently lacking?

- Did the authors compare their SV/CNV estimates to the recent paraphase work for 160 loci with paralogous genes, see e.g. Chen et al. <https://www.biorxiv.org/content/10.1101/2024.04.19.590294v2>

-

Figures:

- Fig2: Inset in panel g may be a separate panel

- Some supplementary figures/graphs lack labelling of axis e.g. [bp]

(Remarks on code availability)

I am not a bioinformatician, hence did not test the code provided. But recommend systemic checks by other reviewers/editors.

As recommended above:

-SVAN is quite central to the downstream analysis, but the Github page is missing (<https://github.com/REPPIO/SVAN>) so it was not possible to evaluate.

Referee #3

(Remarks to the Author)

This study conducts Oxford Nanopore Technologies (ONT) long-read structural variant (SV) calling and haplotyping on the 1000 Genomes Project (1kGP) sample collection. The 1kGP cohort is relatively large and represents geographically diverse populations. The analysis is impressive in scale and quality. I appreciate and respect the amount of work done here.

On the other hand, the advance in terms of concepts or biology is more difficult to see, at least with the way the manuscript is currently presented. Population-scale analyses of SVs with long-read sequencing have previously been published (refs 5, 20, 82). The study uncovers 167,291 SVs but does not say how many of these SVs are absent from existing databases or publications.

Long-read genomic analyses of mobile element insertions (MEIs) have been published before (e.g. PMID: 31853540, PMID: 33186547, PMID: 37823611) and their findings are recapitulated here, although these prior reports are not cited and should be. There are 28,358 non-reference MEIs mentioned, but no table (amongst 20 in the supplement) listing them? How many are completely new to this study? If the dataset is viewed more as a resource than as a breakthrough in terms of biology, in its current state it lacks sufficient information to be accessible to readers or future studies to realise its full value.

My main area of expertise is MEIs and I have focused on that part of the manuscript in my comments below.

Comments:

1) How many of the 167,291 SVs and 28,358 MEIs have been reported elsewhere? Here I mean in large-scale population datasets OR smaller datasets using long-reads specifically to look for SVs or MEIs. Please clarify as well whether 1 or more "spanning" reads were required to fully cover MEIs and their two breakpoints in order to be called.

2) Several studies (a selection noted above) have used long-read sequencing to examine MEIs, and their findings inform and largely agree with what is found here. That should be stated plainly. It is notable that these works studied L1 and/or SVA transductions, finding for example the differing 5'/3' ratios for L1s and SVAs. I would suggest referring to the earlier works when considering the presentation of this section. Similarly, retrotransposon associated inversions have been considered in detail elsewhere with long-read analysis (PMID: 36402840).

3) Of the source elements noted on page 12, how many have been reported before? From looking at Table S16, it looks to me like the vast majority have been, which is a good sign in terms of reproducibility but lacking in terms of continuity and annotation quality. This information should be noted in the text and in the table.

4) The 8q21.11 source L1 noted as only having 5' transduction events is one we showed does the same thing in the embryonic brain (PMID: 31230816, see Fig. 6) because there's a strong upstream promoter next to it (check the transductions are spliced please, if they are it is the same mechanism). It's very cool to see that this source element 5' transduces in the germline - that could be noted in the main text. In the same study we found the chr13q12.3 source L1 jumps readily in brain and carries 3' transductions. Also see PMID: 34772701 and other papers from the Devine lab taking the time to annotate source elements). Bottom line: a lot of work has gone into the source elements mentioned here by numerous other groups before now and those annotations should be exploited and used.

5) The authors really should cite these two earlier papers (PMID: 10066175, PMID: 7920631 - the latter is the first report of a germline L1 transduction, although they call it a "USC") from the Kazazian lab when talking about L1 transductions.

6) Along these lines, please check the references for the sentence "Consistent with prior studies^{5,9,73}, we find that the relative proportion of 3' or 5' transduction events differs considerably between both MEI classes" because reference 73 I doubt says anything about transductions.

7) The text says that 96.4% of MEIs show retrotransposition hallmarks. Please provide a table of these MEIs, their genomic coordinates, and the hallmarks (TSD sequence, polyA sequence, integration site EN motif). What is the median TSD size? The cutoff for a polyA to be called (10bp) is very short considering these tend to be much longer than 10bp on average. A 1bp TSD cutoff is also very short. If more stringent TSD sequence feature cutoffs are applied, is the figure still anywhere close to 96.4%? Along those lines, it is a concern that so many of the MEIs (1,115 events) are 3' truncated. This generally won't happen via TPRT ... please show these in the table. Providing this information will make the data more useful to others.

8) There are a handful of HERVK insertions reported. These are unexpected. Do they have the correct sized TSDs? Did the authors do a manual inspection of these in the IGV to check they look right? It seems (from Fig 3a) that the HERVKs aren't proviruses or LTRs ... are they just random bits of HERVK? Have they been RT'd by an L1? (TSD length informative here).

Geoff Faulkner (University of Queensland)

(Remarks on code availability)

Version 2:

Reviewer comments:

Referee #1

(Remarks to the Author)

The authors have performed a great many additional analyses and have reworked the manuscript extensively which I appreciate. While I think the manuscript is substantially strengthened I am still confused by the comparison to genome assemblies. These confusions are to some extent an extension of my original concerns regarding the utility of the dataset overall (R1.1). There does not seem to be a comprehensive analysis of sensitivity and specificity. FDRs are rather confusingly given for different MEI classes and then for "all deletions" (presumably also including the MEIs?) for two different size bins. The question is, what is the True positive rate? Or the false negative rate? This needs to be clearly broken down for different event types shown in Figure 3 (not just MEIs) and importantly considered as a function of allele frequency and size. If there is a singleton SV how likely am I to find it? To be sure, the approach here is very interesting and important, but it seems that given the number of SVs discovered in 1000 individuals vs the number discovered in 10-fold fewer individuals that this is coming at a huge cost to sensitivity. This does not invalidate the interesting results therein to be sure, but is critical to evaluate the utility of this approach for disease studies and highlighted by the manuscript in several places.

Related to the above, what does this mean: "reflecting altogether 37,834 SVs (43.7%) in our SAGA callset" does this mean that the SAGA callset has 43% of the calls that are detected in the assemblies?

(Remarks on code availability)

it is open and quite good! they did a lovely job!

Referee #2

(Remarks to the Author)

It was a pleasure to review the revised version of "Long-read sequencing and structural variant characterization in 1,019 samples from the 1000 Genomes Project" by Schloissnig, Pani et al.. The authors have not only addressed the majority of my previous suggestions, or provided logical argument why it is not possible to address few minor points now. They have also provided a lot of extra analyses, and have further improved their manuscript. There is few very minor remaining suggestions:

- Great the authors updated the SV-Mb sum (previous comment 2.2.). They may consider that their total number of SV per individual and the SV-Mb sum per individual is still significantly lower compared to higher coverage/quality datasets, such as platinum-pedigree (see e.g.: The Platinum Pedigree: A long-read benchmark for genetic variants | bioRxiv), and cite this accordingly in their discussion.
 - Previous comment 2.5: While the full public data sharing is wonderful, I was hoping for a very practical sharing of the SV dataset for non-bioinformaticians. E.g. a bed file with all SVs calls, or an aggregate SV database with allele frequencies within 1kG. this would allow many more scientists to use the SV catalogue to filter their (long-read) genome data to e.g. filter-out common SVs. A UCSC SV track could also be of great added value.
 - The citation of the newly included dataset by Höps et al. (ref 28) is not accurate. The MedRxiv preprint is available at AJHG, and the same citation in the supplemental note (ref 19) is incorrect.
- In summary I now fully support publication of te revised manuscript.

(Remarks on code availability)

Referee #3

(Remarks to the Author)

The authors have done a commendable job of addressing my concerns and I have no further criticisms to offer.

Geoff Faulkner (University of Queensland)

(Remarks on code availability)

Version 3:

Reviewer comments:

Referee #1

(Remarks to the Author)

I appreciate the detailed analyses of true positive, false positive, and false negatives. However, this is largely relegated to the supplement and thus gives the reader an incomplete view of the data. The updated main text should make reference to these numbers as requested. Currently FDR is mentioned, but not TPR. Sensitivity is mentioned, but not specificity. Simply mentioning these numbers in passing is essential.

(Remarks on code availability)

To the three referees:

We thank all three reviewers for their positive and constructive feedback, and are delighted to see that they have recognized the advances made in our study.

- In particular, **Reviewer #1** highlights the “large number of individuals” for which we provide long-read data, noting that the generated “resource is well-constructed and important”, and that the “SV calling approach utilizing both CHM13 and the HPRC pangenome graph is well done”. Reviewer #1 also acknowledged the insights on locus-specific transduction tendencies presented.
- **Reviewer #2** refers to our work as “an intriguing study”, and underscores the effort in achieving “integrated linear and graph-based approaches for SV analysis via pangenome graph-augmentation,” generating a “very valuable resource of data and methods.” Reviewer #2 also notes the significant increase in the number of “basepair-resolved SV breakpoints for 1kGP samples”, and findings into challenging SV types which includes “insights into SV mechanisms/molecular origins and biology”, and “novel population SV patterns”.
- **Reviewer #3** emphasises that the “analysis is impressive in scale and quality” and “appreciates and respects the amount of work done.”

We have revised our manuscript to comprehensively address the feedback provided. This includes:

- (1) Systematic analyses clarifying the sensitivity and specificity of our SV dataset;
- (2) A much more detailed analysis and extension of our resource through resource-wide genotyping of polymorphic variable number tandem repeats (VNTRs) using the vamos toolkit¹;
- (3) An expanded analyses of recurrent deletion events mediated by flanking mobile elements;
- (4) An analyses of our SV dataset following ancestral state polarisation using a chimpanzee telomere-to-telomere (T2T) assembly;
- (5) Substantial expansion of our analyses of mobile element insertions, including by assessing novelty of element polymorphisms with respect to prior data. Moreover, our revised manuscript reports on a regulatory element mediating 5` transduction bias somatically and in the germline for an active source L1. This mechanism involves the formation of SVs due to aberrant splicing and subsequent retrotransposition of a L1 juxtaposed to a strong transcriptional start site.
- (6) A novel vignette comprising the analysis of rare disease patient genomes, with an illustration of how our resource could facilitate the prioritisation of causal SVs in rare disease diagnostics.

In revising our manuscript, we have incorporated these new analyses while maintaining brevity wherever feasible. At the editor's discretion, we are open to further shortening of our manuscript.

Lastly, we stress that we recently experienced a great number of requests for data access and reuse. Preprints from De Coster *et al.*², Lansdon *et al.*³, Zheng *et al.*⁴, Noyvert *et al.*⁵, and Reeve *et al.*⁶ have leveraged our data to advance their scientific findings, while adhering to the embargo guidelines not currently allowing genome-wide analyses. Numerous other researchers have inquired about genome-wide data reuse, emphasising that they will rapidly integrate our data once our embargo for genome-wide analyses lifts. This underscores the utility and timeliness of our long-read data resource.

Point-by-point responses to each specific reviewer comment

Referee #1 - General Remarks

Referee expertise: genomics, SV

This manuscript reports on medium coverage nanopore sequencing from ~1000 1KG samples. Using these data the authors identify ~170K structural variants (SVs) and report on several of the properties of these SVs. The efforts outlined dovetail with several other long read sequencing and genome assembly efforts, however the major contribution of this work in my opinion is the large number of individuals. Nevertheless, the sequencing depth and quality reported in this resource are of substantially lower quality than other efforts highlighting the tradeoff made towards sequencing more individuals. I think this work thus has potential, but the authors need to make more of a case for their approach and why it is useful in the current era of complete genome assembly, which is substantially better at resolving SVs, particularly in some specific regions of the genome. While the resource generated is commendable for its scope, stronger motivations are needed to justify the low coverage and short read lengths. The manuscript would be also substantially improved if the authors directly demonstrated/highlighted the utility of these data and made specific efforts to highlight novel biology or analyses that these data empower. To be sure, I think this resource is well constructed and important, but more up-front discussion of the limitations of the dataset are needed.

Response: We appreciate the thoughtful and constructive feedback on our manuscript. A major strength of our study is indeed the large number of individuals sequenced, resulting in a publicly available open-access long-read population-scale data resource. To clarify, we categorise long-read genomic approaches into two primary types:

Approach 1: Sequencing relatively few samples at high coverage with multiple complementary platforms, yielding high-quality whole genome assemblies (e.g., Ebert *et al.*⁷, Liao *et al.*⁸, Logsdon *et al.*⁹). This approach is labour-intensive and very costly, and does not currently scale to 1000 human genomes or more. Yet, it excels in SV resolution, including in extremely hard sequence contexts such as centromeres and complex segmental duplications. We have observed that sample set sizes in studies using this approach have shown somewhat limited growth in recent years (i.e., 35 genomes in Ebert *et al.* (2021)⁷, 47 genomes in Liao *et al.* (2023)⁸, and 65 genomes in Logsdon *et al.* (2024)⁹) in line with the high costs and labor-intensive nature of the approach.

Approach 2: Sequencing larger cohorts on one long-read platform at intermediate coverage, yielding variant calls but typically no robust genome assemblies. This application-driven approach has previously been restricted to controlled-access datasets^{10,11}. While it does not achieve T2T-like assemblies, Approach 2 is significantly less laborious and more cost effective, offering valuable applications for population genetics and disease studies (see e.g. the American All-of-Us project, which pursues this approach too). We adopted Approach 2 to sequence over a thousand samples from the open-access 1kGP cohort, thereby openly coordinating sample overlap with various ongoing efforts like the Human Pangenome Reference Consortium (HPRC), the Human Genome Structural Variation Consortium (HGSVC), and a US-based study led by Danny Miller. This has enabled SV discovery across diverse haplotypes, complementing efforts targeting fewer genomes per population with multi-platform assemblies. The computational tools developed with our resource set a foundation for further scaling long-read analyses in cohorts similar to ours.

In our revised manuscript, we have clarified the distinctions into **Approaches 1** and **2**, acknowledging both strengths and limitations associated with sequencing depth and the use of a single sequencing platform. Besides expressing the motivation for this study more clearly, we now also provide a more nuanced assessment of gains for different SV classes. In particular:

- The ability to detect insertions in a sample is notably very comparable between **Approach 1** and **Approach 2**. Thereby, the cost-effectiveness of Approach 2 provides a great advantage for characterising mobile element insertions (MEIs) segregating in the global population down to low allele frequencies (see e.g. our response to **Comment #1.5** below).
- For VNTRs, our ONT dataset is in principle well-suited to characterise the different alleles based on the number of repeat copies. Our graph augmentation pipeline (SAGA), which is based on the minigraph toolkit¹², however incorporates only SVs (i.e., differences between alleles of 50 bp or larger), which leads to an underrepresentation of VNTR allelic diversity, as VNTR polymorphisms frequently involve repeat sizes <50 bp and as such allelic structures differing by less than 50 bp. We now acknowledge this limitation in our manuscript – and note that challenges in capturing multiallelic VNTRs within a graph were alluded to in the original minigraph paper¹² (see also response to **Comment #1.1**). Our Discussion section now highlights the importance of addressing multiallelic VNTRs, by encouraging the community to focus on designing improved algorithms for graph-based SV analyses in the future. In the meantime, we have expanded our manuscript with a specific analysis of VNTR diversity using a specialised VNTR genotyping¹ tool. We illustrate utility by typing extreme repeat unit polymorphisms at VNTRs implicated in late-onset diseases (see **Comment #1.1**).
- The value of our open-access resource is further demonstrated by insights gained from it: This includes our findings pertaining to mobile element related structural variation addressed in detail below in our responses to **Comments #3.5** and **#3.6**. In our revised manuscript, we now also showcase how candidate causal SVs can be filtered effectively in rare disease patients using our SAGA-based SV resource (see response to **Comment #1.11**). Moreover, we have expanded our analysis of recurrent SVs mediated by NAHR, which now covers a more systematic analysis rather than showcasing only a single case (response to **Comment #1.15**).

Referee #1 - Specific Comments

Reviewer #1.1: This manuscript is arriving at an interesting time in that long read assemblies are indeed enabling the discovery and genotyping of previously intractable SVs. However, the largest gains are being made from haplotype-resolved, complete genome assemblies, some of which the authors of this manuscript are involved with (e.g. HPRC Nature 2023, Ebert et al, Science 2021). The authors make the argument that such assembly efforts are “hampered by small sample sizes and the lack of long read sequencing panels of normal individuals from diverse panels.” However, there are problems with this argument. First, these other studies *are* made up of diverse, normal individuals. Secondly, while the sample sizes are indeed smaller, the high quality of the resultant assemblies appears to more than make up for this – particularly when compared to the approach of this manuscript in which both the coverage and N50 read lengths are not particularly high. For instance, while this study identifies 167K SVs in ~1000 samples, Ebert et al identified 107K SVs in just 32 haplotype resolved genome assemblies (31-fold fewer individuals). Furthermore, Gustafson et al (10.1101/2024.03.05.24303792) performed a very similar analysis sequencing 10-fold fewer (n=100) 1000 genomes samples. However, presumably due to the higher sequencing depth (>2X higher) and read length (>2X longer) they were able to identify a nearly as many SVs (~150K). Indeed, Gustafson figure 3D is essentially identical to 2a in this manuscript. Thus, the authors need to make the case for why their approach is preferred / complementary and discuss the tradeoffs.

Response: We appreciate the acknowledgment of the significance of long reads in identifying challenging SV classes. We have responded to parts of this reviewer comment above in our response to the reviewer #1 general comments. In brief, we emphasize that our study – unlike prior work such as Ebert *et al.*⁷, the HPRC⁸ or Gustafson *et al.* – utilises a large sample size, representing 26 human populations with ≥32 samples each. This comprehensive sampling ensures robust detection of low allele frequency and geographically stratified SVs in 1kGP samples^{13,14}.

We acknowledge that prior smaller-scale studies have also reported extensive SV catalogs. To address this reviewer comment, we have conducted additional analyses to assess to what extent our data resource is contributing to a dataset growth in low-frequency SV alleles. A heavy tail of rare alleles is expected in the human population, as the site-allele frequency spectrum typically behaves as a power law – i.e., a line in the log-log plot, as confirmed in **Fig. R1/2c** and **Fig. R2/S10**. More detailed analyses confirm the anticipated growth in key SV classes, such as full length MEI source elements, for which our resource increases previously published catalogs by 4-5 fold (see our detailed response to **Comment #3.5**).

These findings underscore the advantages of our sequencing strategy and the SAGA method compared to other studies (e.g., Gustafson *et al.*, Ebert *et al.*, and the HPRC). On the other hand, we acknowledge an under-representation of VNTR-based SV alleles when using SAGA, which is explained as follows: VNTRs are typically multiallelic whereby SV alleles often differ by very small periodicity changes at these sites. The graph-based representation geared towards SVs that we employ in SAGA shows limitations in this case. Specifically, with the use of minigraph for graph construction¹², a new allele is incorporated only if it is sufficiently different (typically ≥50 bp) to all alleles already represented in the graph. As Heng Li *et al.* (Genome Biology, 2020) previously noted, multiallelic VNTRs pose a particular challenge to graph construction toolkits, as these frequently differ by just a few basepairs between structural haplotypes, preventing an optimal graph representation. Consequently, SAGA, which in its present form uses minigraph, offers a

lower-bound estimate of the SV diversity at these sites. We envision that in the future, SAGA could use other graph construction and sequence-to-graph mapping tools to remove this limitation. This is presently hampered by the lack of scalable tools to map long reads to graphs incorporating all variation, including small variants (i.e., SNPs and Indels), such as graphs built by MinigraphCactus¹⁵ or PGGB¹⁶.

Our revised manuscript addresses these current pangenome graph-related limitations associated with multiallelic SV sites in the Results, Discussion and Methods. Irrespectively, we stress the advantages of SAGA for population-scale variant integration: SAGA enables graph-aware SV discovery, integration of variants with specific strengths for sites with moderate allelic diversity, and large-scale genotyping of SVs using long reads. Furthermore, as stated above - to better account for VNTRs, we have constructed an expanded VNTR dataset based on the *vamos* caller¹. This callset preserves the multiallelic structure of VNTRs and adds variation at sites undergoing VNTR expansion and contraction¹. This expanded resource supports two objectives: advancing the utility of our 1,019 ONT genome resource for VNTR-focused research and enhancing the ability of researchers to exclude “common SV structures” in disease studies.

A detailed analysis of the VNTR calls derived from the *vamos* dataset reveals 767 biallelic and 369,656 multiallelic VNTR sites – significantly surpasses the estimates of VNTR diversity obtained through graph augmentation, which identified 26,489 VNTRs originating from biallelic bubbles and 26,935 VNTRs originating from multiallelic bubbles. An in-depth comparison of repeat unit counts from the 1kGP samples against a recent multi-platform whole genome assembly study ($N=65$ samples)⁹ shows strong correlation in VNTR length inference (**Fig. R3/S38**) – showing that VNTR lengths are robustly estimated from our long read data resource. Furthermore, with its 1,019 genomes, our data resource allows exploration of more extreme repeat unit counts compared to prior long-read studies on smaller 1kGP sample sets, providing data pertaining to the spectrum of normal VNTR variation including for loci relevant to human disease. Our revised manuscript illustrates this for the *PLIN4* and *ABCA7* loci implicated in late-onset human disease (see **Note S5, Fig. R4/S40**).

In summary, our manuscript emphasises the advances enabled by SAGA in relation to whole genome assembly studies pursued for smaller sample sets – see, for example, our new vignette on rare disease candidate SV filtering that we refer to in our response to **Comment #1.11** enabled by SAGA, and main findings of our manuscript based on graph augmentation (**Figs. 2, 4 and 5**). We have added to this the ability to utilise our read data to explore VNTR diversity (**Fig. R5/S39**). Additionally, we have expanded the discussion of the tradeoffs inherent to SAGA. Irrespectively of this, we emphasise that our study – including the open source tools provided – sets a strong basis for future studies in large cohorts, where harmonised representations (specifically for biallelic SVs) now enabled by SAGA are likely to be crucial to study success^{13,14,17}.

Figure R1 (Corresponds to Figure 2c): SV allele count (x-axis) relative to the count of SV sites (y-axis), constructed by genotyping the original HPRC graph (denoted HPRC_mg, blue line) and the augmented graph (HPRC_mg_44+966, green line) with giggles, as part of the SAGA framework. Consistent with the much smaller panel size employed, the HPRC_mg graph under-represents low frequency SV alleles. The augmented graph distribution, however, includes low frequency alleles, consistent with a power-law distribution typical of the SV site-allele frequency spectrum in a thousand genomes.

Figure R2 (Corresponds to Figure S10): Cumulative growth of SV set with each sample added for the genotypes on the original HPRC graph (above) as well as the augmented graph as part of the SAGA framework (this resource; below).

Figure R3 (Corresponds to Figure S38): The Pearson's Correlation Coefficient (PCC) of comparing vamos-based VNTR calls generated from our long-read 1kGP resource to VNTRs called from multi-platform whole genomes assemblies⁹ (HGSVC3) using vamos (see **Note S5** 'Calling VNTRs using vamos'). We compared the count of repeat units in the VNTR alleles obtained from the 1kGP ONT reads to the alleles obtained from HGSVC3 assemblies. We filtered out the VNTR alleles where the HGSVC3 assemblies reported alleles close in length to the reference VNTR allele. We determined the alleles as close in length to the reference when the length of the HGSVC allele was within 90% to 110% of the reference allele.

Figure R4 (Corresponds to Figure S40): The figure shows the distribution of the base pair lengths and the count of repeat units (RUs) in the VNTR alleles genotyped by vamos on the ONT data of the 1019 samples of this study (referred as 'ONT' in the plot) and alleles genotyped by vamos¹ on the multi-platform-based whole genome assemblies from Logsdon *et al*, 2024⁹ (referred as 'HGSVC' in the plot). We show the distribution of two VNTR loci found in the genes ABCA7 (chr19:1012105-1014401) in (A) and PLIN4 (chr19:4494323-4497243) in (B), which have been associated with late-onset human disease^{18,19}. For the ABCA7 VNTR locus, alleles of a length greater than 5,720 bp (denoted through a vertical line in (A)) are associated with late-onset disease, whereas for the PLIN4 VNTR locus, alleles with repeat count of 40 (denoted as a vertical line in (B)) are disease-associated. We report a 43 repeat unit count VNTR allele for the PLIN4 locus found in sample NA20127 (outlier denoted with an arrow); this variant was confirmed using manually inspection (**Fig. S86**).

Figure R5 (Corresponds to Figure S39): Density plots comparing the range difference of repeat unit counts for different percentile ranges for the VNTRs called from our resource (“ONT”) and from HGSV3 multi-platform whole genome assemblies⁹ (“HGSV3”). The plotted range is restricted to data points where $x < 100$ and $y < 100$ (for visualisation purposes). They also show lines for $y = x \pm c$ where $c = 5, 20, 50$. Increased c values indicate more extreme cases where one dataset reports higher VNTR repeat unit (RU) ranges compared to the other (see **Note S5** ‘Analysing the VNTR diversity’ and **Table S31**). RU, repeat unit.

Reviewer #1.2: Comparisons to short read based discovery are relevant but need only be mentioned in passing – it is well understood that such methods are underpowered. Similarly, SNP chip-based validation seems underpowered (see below).

Response: We thank the reviewer for this statement and agree that while short read data from 1kGP samples have seen very wide adoption in the genomics field^{13,14,20}, these data indeed have inherent limitations for SV characterization. This limitation underscores our motivation to resequence 1kGP samples using ONT reads. In response to this reviewer comment, we have focused on reducing the emphasis on short-read sequencing data in the main text.

We acknowledge that SNP chip-based validation, used routinely as a validation tool by the 1kGP over the years^{13,21–23}, captures only a subset of SVs identified from long-reads. Our Methods section now clarifies this (although we stress that other validation approaches likewise have their limitations). Our manuscript complements the use of SNP chips with further means of validation, including augmenting the CHM13 reference with insertions coupled with mapping sequences from compacted de Bruijn graphs using short reads, Mendelian inconsistency testing, and genotype concordance analysis comparing to short reads. During paper revision, we have also performed a systematic comparison to state-of-the-art T2T-style multi-platform assemblies, as an additional

complementary validation approach (see our responses to **Comments #1.4** and **#2.1** below). We emphasise that all validation methods consistently confirm the high quality of biallelic SV calls generated from SAGA.

Reviewer #1.3: Related to above, a deep comparison / contrast is needed showing the utility of ONT resequencing as function of coverage, read length N50, and sample size (see below comment).

Response: We performed several analyses in response to this reviewer request, including:

- (1) Assessment of the effect of coverage on the number of discovered SV sites per sample after integration with the SV Analysis by Graph Augmentation (SAGA) framework.
- (2) Analysis of the effect of coverage on the SV genotype quality seen in each sample based on Giggles.
- (3) Examination of the effect of read length N50 on the number of discovered SV sites per sample after integration with the SAGA framework.
- (4) Analysis of the effect of read length N50 on the SV genotype quality seen in each sample based on Giggles.
- (5) Single-sample discovery of SVs with the long read version of DELLY, using samples from a variety of distinct coverage bins.
- (6) Single-sample discovery of SVs with DELLY, using samples with a variety of distinct N50 read lengths.

To perform these analyses, we first stratified our samples by coverage and N50 read lengths (**Table R1**) and evaluated SV genotype qualities using the Giggles tool. Stratification by coverage (**Fig. R6/S81**) or N50 (**Fig. R7/S82**) did not noticeably impact SV genotyping qualities. However, very low coverage levels did affect the number of SVs called: samples with $<10\times$ coverage showed a slight reduction, while SAGA consistently identified similar numbers of SVs across coverage levels $\geq 10\times$ (**Fig. R8/S83**). Since 91% of our samples have $\geq 10\times$ coverage in our resource (median $\sim 16.6\times$, **Table R1**), and since our analyses suggest saturation at this coverage level when using SAGA, we conclude that sequencing coverage is indeed adequate for the vast majority of samples. Notably, there is a considerably more pronounced coverage dependence when performing single sample SV discovery with DELLY alone (**Fig. R8/S83**). Additionally, we find that N50 read length has only a minimal impact on the number of SVs called (**Fig. R9/S84**). In the light of these beneficial characteristics of SAGA, we refrained from repeating the complete graph augmentation process with intermediate sample sizes with SAGA – which would come at considerable computational cost and in the light of the results obtained would be unlikely to reveal any additional unexpected (or interesting) behaviour.

In summary, these additional analyses demonstrate the robustness of graph augmentation, particularly in handling variations in experimental parameters such as coverage and N50 read length in our resource. In contrast, single-sample SV discovery methods, such as DELLY when run in a single-sample discovery mode (also seen for the SVision SV calling method; see also **Comments #1.4** and **#2.1**) are much more affected by coverage fluctuations. These findings highlight the necessity of employing SV callset integration strategies like SAGA when working with long read data resources at population-scale. The analyses described in this response as well as the relevant display items have been added to our revised Supplement.

Median Coverage Range	Number of Samples	N50 Range (kb)	Number of Samples
0x to 9x	92	0 to 10	59
10x to 19x	617	10 to 20	422
20x to 29x	234	20 to 30	387
30x to 39x	60	30 to 40	132
>= 40x	24	> 40	27

Table R1: Number of samples in N50 and sequencing coverage range bins.

Figure R6 (Corresponds to Figure S81): Genotyping quality assessment when stratifying samples by coverage. The plot demonstrates the genotype quality of the genotypes by Giggles on the HPRC_mg_44+966 graph after filtering across four sequencing coverage ranges (<math><10\times</math>;

Figure R7 (Corresponds to Figure S82): Genotyping quality assessment when stratifying samples by N50. The plots demonstrate the genotype quality of the genotypes by Giggles on the HPRC_mg_44+966 graph after filtering across four sample N50 ranges (<10 kb; \geq 10 kb and <20 kb; \geq 20 kb and <30 kb; \geq 30 kb and <40 kb). A) shows genotyping quality using a comparison of the SV allele frequency of an allele in the VCF panel (created using the HPRC assemblies and the pseudohaplotypes of the SAGA framework) with the allele frequency of the same allele genotyped by Giggles in the callset (using only the 908 unrelated samples from our callset). B) Genotyping quality using a Hardy-Weinberg Equilibrium (HWE) plot, with the SV allele frequency (AF) of the genotyped allele plotted versus the percentage of samples heterozygous for that allele (using only the 908 unrelated samples from our callset).

Figure R8 (Corresponds to Figure S83): Assessment of the effect of coverage on the number of SV sites per sample after integration with the SAGA framework. Differences in SV discovery between single-sample SV calling with DELLY (labelled 'single_sample_delly'), DELLY population-level SV calling (labelled 'delly'), and SV integration with SAGA followed by Giggles based genotyping ('callset') after stratifying samples into four coverage bins (<10 \times ; \geq 10 \times and <20 \times ; \geq 20 \times and <30 \times ; \geq 30 \times and <40 \times), as well as into samples of African (AFR) and non-African (non_AFR) ancestry. Note that SV numbers plotted on the y-axis correspond to the number of heterozygous (het) SV sites in the dataset plus twice the number of homozygous (hom) SV sites.

Figure R9 (Corresponds to Figure S84): Assessment of the effect of read N50 on the number of SV sites per sample after integration with the SAGA framework. Differences in SV discovery between single-sample SV calling with DELLY (labelled 'single_sample_delly'), DELLY population-level SV calling (labelled 'delly'), and SV integration with SAGA followed by Giggles based genotyping ('callset') after stratifying samples into five read N50 bins (<10 kb; ≥ 10 kb and <20 kb; ≥ 20 kb and <30 kb; ≥ 30 kb and <40 kb), and into samples of African (AFR) and non-African (non_AFR) ancestry. Note that SV numbers plotted on the y-axis correspond to the number of heterozygous (het) SV sites in the dataset plus twice the number of homozygous (hom) SV sites.

Reviewer #1.4: The resource quality assessment needs to compare this approach to assembly-based SV detection. Both approaches are being pushed extensively (as noted in the discussion) and so there is a strong interest in this.

Response: In response to this reviewer comment we have added a new section to the **Supplementary Material**, and amended the resource quality assessment section in the **Results**. These new sections provide a detailed comparison between our dataset and recently generated multi-platform whole genome assemblies (from Logsdon *et al.*)⁹. We also performed a VNTR-based comparison with the multi-platform whole genome assemblies from Logsdon *et al.*⁹ (see also **Comments #2.1, #1.5 and #1.11** in this regard).

Comparison to multi-platform whole-genome assemblies⁹ enables an in-depth assessment by SV class. This notably revealed FDRs from 0.85%–6.75% for different classes of biallelic mobile element insertions (**Fig. R10/S16, R11/S17, R12/S18**). For deletions, we estimate an FDR of 1.94% for biallelic mobile element deletions based on multi-platform assemblies (**Note S3**). When considering all deletion sites, which contain numerous multiallelic tandem repeat events, graph-based analysis yields FDR estimates dependent on the SV length: For deletions ≥ 250 bp, the estimated FDR is 10.2%, whereas for smaller deletions that predominantly consist of short tandem repeats (**Fig. R13/S19**) the estimated FDR is 37.7%. This finding supports the notion that polymorphic tandem repeats currently present an inherent challenge for SV interpretation, in particular in pan-genome graphs¹² (where the comparison process itself is likely to be affected) but also as highly multiallelic sites in population cohorts in general. This finding led us to pursue genotyping of percentile ranges of repeat units for all VNTRs in all 1,019 samples using the vamos caller (see response **Comment #1.1**).

Reviewer #1.5: The FDR seems on the high side (8-11% for deletions and insertions). Comparing how this would be augmented with different coverage / read lengths would be beneficial.

Response: As alluded to in our response just above (**Comment #1.4**), we find that the FDR is lower than 8-11% for several well-defined classes of biallelic SV. For example, when comparing MEI calls generated in our resource to multi-platform long read assembly based MEI calls⁹ we observe an FDR of 3.94% for canonical *Alu* insertions, 6.75% for canonical L1 insertions, as well as 0.85% FDR for canonical SVA element insertions (**Fig. R10/S16**). We note however that when assessing all MEIs including non-canonical events the estimated FDR is somewhat higher, namely 13.66% for *Alu* (canonical + non-canonical), 24.63% for L1 (canonical + non-canonical), and 45.31% for SVA (canonical + non-canonical). We stress that this is likely attributed to differences in the annotation of non-canonical MEIs between studies and calling algorithms, and thus, should not necessarily be interpreted as a reduction in the quality of these SV calls (**Fig. R11/S17**).

When compared to MEI calls derived from multi-platform long-read assemblies⁹, our low-cost SAGA base approach also shows remarkable sensitivity: 90.59% for *Alu* insertions, 85.22% for L1, and 84.29% for SVA, considering both canonical and non-canonical MEIs. These sensitivity estimates notably do not show a consistent trend with respect to coverage or N50 read length, and are thus broadly in agreement with respect to the observations made for genotyping (**Fig. R12/S18**, see response to **Comment #1.3**).

Figure R10 (Corresponds to Figure S16): Comparison of mobile element insertion calls of the SAGA framework compared to whole genome assemblies using canonical MEIs only. Using the MEI calls from multi-platform whole genome assemblies (with MEI calls based on PALMER2 and L1ME-AID9)⁹, we show the number of shared and unique MEIs by transposable element type for all canonical MEIs in the SAGA call set annotated with SVAN.

Figure R11 (Corresponds to Figure S17): Comparison of mobile element insertion calls of the SAGA framework compared to whole genome assemblies using non-canonical and canonical MEIs. Using the MEI calls from whole genome assemblies⁹, we show the number of shared and unique MEIs by transposable element type for all non-canonical and canonical MEIs in the SAGA call set annotated with SVAN. One possible explanation for the larger number of unique non-canonical MEIs in our dataset not detected by Logsdon *et al.*⁹ is that unlike in our study (see e.g. our response to **Comment #3.9**), Logsdon *et al.* did not separately characterise non-canonical MEI events in their study.

Figure R12 (Corresponds to Figure S18): Comparison of whole-genome based mobile element insertion calls stratified by N50 read length and coverage with respect to the calls made from our ONT 1kGP sample resource.

Figure R13 (Corresponds to Figure S19): Deletions called by the SAGA framework compared to whole genome assemblies from HGSVC for the 16 overlapping samples. Using the deletion calls from whole genome assemblies⁹, we show the false discovery rate and the fraction of deletions with tandem repeats identified using TRF²⁴ stratified by deletion size.

Reviewer #1.6: “Unlike prior population-scale surveys of 1kGP samples based on short read sequencing^{3,4,30}, our resource captures SVs along their whole size spectrum, which includes 50 to several hundred basepair size SVs that are poorly captured^{6,7} by Illumina reads.” The statement “along their whole size spectrum” is not correct. This statement should be modified to clarify that this approach indeed can capture additional signal, but is low powered in many ways as well (e.g., large deletions, SDs, inversions, etc).

Response: In response to this reviewer’s feedback, we revised our manuscript to clarify both the strengths and limitations of our approach, and we have rewritten this sentence (we do agree with the reviewer that “along the whole size spectrum” was not correct). Furthermore, we emphasise that our findings are compatible with the previously reported high sensitivity of short reads in the large deletion discovery based on read depth^{25–27}. Our revised Discussion section also clarifies that SDs, VNTRs and SD-embedded inversions are captured with reduced sensitivity using SAGA. Regarding inversions, we also emphasise that a relevant subclass – twin priming events resulting from aberrant retrotransposition within mobile elements^{28,29} – was omitted in prior multi-platform genome assembly based SV surveys of 1kGP samples (i.e., Ebert *et al.*⁷, Liao *et al.*⁸, and Logsdon *et al.*⁹), thus highlighting one additional strength of our study. This class of inversions is included in our updated resource, and is facilitated by algorithmic advancements such as the SVAN method. Additionally, as mentioned above, we have now expanded our resource by exploring VNTRs beyond the SAGA framework during manuscript revision, utilising the vamos tool¹ (see our response to **Comment #1.1**).

Reviewer #1.7: There is a large peak in 1.5kb insertions found in short reads by not the long reads (Fig 2b pink peak). What is this and why is it not captured by the long read data?

Response: A detailed investigation of the '1.5 kb insertion peak' identified in the short-read sequencing data, supported by consulting the primary authors of the Byrska-Bishop *et al.* study¹⁴,

reveals that this peak does not correspond to the true insertion length distribution. Instead, it underscores the limitations of SV calling inherent to short-read analysis. Byrska-Bishop *et al.* applied MELT for MEI discovery¹⁴, a tool that aligns aberrantly aligned short-read sequences against a predefined database of MEI reference sequences. As a result, the estimated sizes of insertions in Byrska-Bishop *et al.* were (largely) constrained by the size of these predefined reference sequences. Byrska-Bishop *et al.* identified peaks in the MEI size distribution at ~270-281 bp for Alu, ~1240 bp for SVA, and ~6016-6019 bp for L1, which closely correspond to the reference sizes used in MELT (Alu: 281 bp, SVA: 1316 bp, L1: 6019 bp)¹⁴. Leveraging our ONT resource, we are now able to precisely measure true MEI lengths, and we find that these are typically larger than predicted from short reads. This shortcoming of short read analysis can explain the (artifactual) ~1.5 kb peak reported earlier¹⁴.

We also note that these observations help underscore the primary motivation of our study, and the need for a long-read sequencing reanalysis of the 1kGP cohort, which we conducted for 1,019 genomes. During paper revision, we have included **Figure R14/S24** into our manuscript, which offers a detailed analysis of the (approximated) allelic lengths of MEI polymorphisms predicted from short reads¹⁴ compared to those determined from our ONT data resource using SAGA. Furthermore, in the caption for **Figure 2b**, we now explicitly clarify that our study provides precise MEI lengths, contrasting with the approximations inherent in short-read analysis. In addressing this reviewer comment, we have also made a minor revision to **Figure 2b** by extending the X-axis range beyond 6 kb, improving the visualisation of the L1 insertion peak. We further stress that our ONT 1kGP data resource provides the resolution necessary for detailed analysis of canonical and non-canonical MEIs, such as transductions – exemplified by the biological insights on the transduction process mediating the formation of a class of SVs in the germline (see e.g. our revised **Figure 4** in the main text).

Figure R14 (Corresponds to Figure S24): Comparison of the SV length of short-read derived MEI predictions and MEIs in the sequence-resolved SAGA call set.

Reviewer #1.8: Is the F_{st} statistic in Fig 2g only reported for biallelic SVs?

Response: The F_{st} statistic in **Figure 2g** is reported for both biallelic and multiallelic SVs. The revised figure caption clarifies that. We also included a new display item in our revised manuscript (**Fig. R15/S25**), which presents the F_{st} statistics separately for biallelic and multiallelic SVs identified with the SAGA framework, and updated **Table S13** to indicate whether the reported SV is bi- or multiallelic.

Figure R15 (Corresponds to Figure S25): Histogram of a) biallelic SV F_{st} values and b) multiallelic SV F_{st} values for all five continental populations.

Reviewer #1.9: The analysis of larger events being rarer irrespective of being dups or dels should be more rigorously performed using the full allele frequency distribution (Fig S27) opposed to arbitrary bins.

Response: In response to the reviewers comment we have included **Figure R16/S33**, which depicts the median SV size in relation to the minor allele count of insertions (depicted in light blue) to illustrate the increase in SV size for rare alleles using the full allele frequency spectrum.

Figure R16 (Corresponds to Figure S33): Median SV size by minor SV allele count. The x-axis value indicates the upper bound of the minor allele count bin. The first bin represents all SVs of allele count 1, the second bin all SVs of allele count greater than 1 and smaller or equal to 2 and so on for increasing allele counts (SVs discovered as deletions are depicted in red, and insertions discovered as insertions are depicted in blue).

Reviewer #1.10: The complex SV calling approach utilizing both CHM13 and the HPRC pangenome graph is well done, and a huge effort to do all the phasing. However, ultimately the deletions vs insertions are only with respect to a single reference. The call set would be improved by adding information about the ancestral (e.g., chimp) allele for population genetics.

Response: Our revised manuscript now incorporates ancestral SV allele information, as suggested. We utilised a long-read T2T chimpanzee assembly (GCF_028858775.2; submission date of Assembly: 8 Jan 2024)³⁰, which we aligned to the augmented graph using minigraph¹², identifying the aligned alleles within each bubble of the graph. We matched the ancestral allele of each bubble to our callset, based on the graph nodes that define both the SV allele and the ancestral allele. We find no noticeable bias related to SV size affecting the inference of ancestral alleles (see **Fig. R17/S75**), indicating that polarisation by ancestral state is largely unaffected by SV size. For most evaluated sites (71.16%), the reference allele corresponds to the ancestral allele, while in 16.37% of cases the SV represents the ancestral allele. Furthermore, 2.96% of SV sites have an unknown ancestral allele due to the inability of the aligner to map into the respective region, and 9.52% exhibit an ancestral allele not matching either the SV or reference allele, likely due to multiallelic variation at these loci. Determined ancestral allelic states are now reported in our release dataset.

We also explored the utility of ancestral state assignments. These resulted in 14,816 deletions being reclassified as insertions, and conversely, 7,189 insertions as deletions. **Fig. R18/S76** depicts the distribution of SV length, homology, and class after ancestral state based polarisation. We find that the number of mobile element deletions decreased by 55.3%, while the number of mobile element insertions increased by 5.6% during polarisation. This is consistent with the well-known retrotransposition process, which results in insertions rather than deletions. We note that members of the younger *AluY* family were most frequently reclassified from deletions to insertions, whereas members of the older *AluS* family more often retained their original SV type. This could reflect the occurrence of deletions over time accumulating at older insertion sites (and potentially, yet if at all to a much lesser extent, polymorphisms shared between humans and chimpanzees). We also find that a fairly large fraction of SVs classified as VNTR (7,981 SVs; 19.7% of all VNTRs) did not receive a definitive ancestral state assignment, underscoring

challenges in defining ancestral states at sites that are prone to multi-allelic variation and high mutation rates.

Interestingly, we observe differences in polarisation rates between insertions and deletions also in the case of homology-driven SVs involving flanking *Alu* elements. Most such events (87.5%) initially classified as insertions were reclassified as deletions. By comparison, only a very small proportion (3.8%) of such events originally called as deletions were reclassified as insertions. This is consistent with *Alu*-mediated homology-driven SV formation (i.e., transposable element-mediated rearrangement, TEMR) resulting largely in deletions³¹. Our revised manuscript now reports these additional insights.

Figure R17 (Corresponds to Figure S75): Length distribution of the SVs where the ancestral allele could be inferred using the chimpanzee genome, in comparison to the full SV sites list. Ancestral state polarisation is not noticeably affected by the size of the SV allele ('All') in our resource.

Figure R18 (Corresponds to Figure S76): Effect of callset polarisation with respect to a chimpanzee T2T genome assembly on SV class distribution and homology landscape. Our SV callset was polarised using ancestral allele assignments. This display item depicts our final callset (a), relative to the CHM13 genome) in comparison to our callset after ancestral state polarisation (b). Scatter plots of SV length versus homology length are shown for SVs coloured by the respective class – with deletions depicted with negative length, and insertions with positive length. Marginal plots show the size-binned fraction of SV classes perpendicular to both axes, depicting deletions and insertions at the left and right, respectively. Panel (a), shown here for visualisation purposes, is the same as in **Figure 5b**. Histograms showing SV length and homology distribution are depicted for our original SV callset (c), and our callset after polarisation (d).

Reviewer #1.11: This approach (medium coverage, shorter read length nanopore) seems to work in some regions of the genome – “high confidence regions,” but less so in so-called “low confidence” regions... this should be discussed extensively. What are the difference between these regions and how are assembly approaches better at getting to the “low confidence regions.” Justification is needed for the utility of this low-coverage ONT approach, which is undoubtedly more cost effective but is poorly powered in “low confidence regions.” Discussion take into account the future directions envisioned by the authors -> e.g. would this approach even discover a disease causing SD mediated microdeletion?

Response: For clarification, stratification of the genome into “difficult” and “high-confident” regions relied fully on the respective genomic masks released by the Genome in a Bottle (GiaB) consortium³² (i.e., the GiaB BED file CHM13_notinalldifficultregions.bed.gz). Our revised manuscript clarifies this. Using these masks, we see considerably increased sensitivity for MEI discovery in high-confident regions compared to difficult regions (92.26% [53.92% in the difficult regions] for Alus, 87.72% [44.72% in difficult regions] for L1s, and 86.23% [63.14% in difficult regions] for SVAs), when using MEI calls from multi-platform whole-genome assemblies⁹ as the ground truth. Yet, based on the same multi-platform whole-genome assemblies⁹, the genome-wide estimate for sensitivity across all genomic regions combined is 90.6% for Alus, 85.2% for L1s and 84.3% for SVAs in our data – showing our cost-effective approach is overall very sensitive.

We also noticed improved SNP-tagability of SVs in high-confident regions (**Fig. R19/S31, Fig. R20/S32**). Furthermore, a more than 12-fold enrichment of VNTR density in GiaB “difficult” regions (**Fig. R21/S70**) suggests that our graph augmentation approach based on minigraph is more likely to struggle in these regions – since VNTRs were previously reported to represent a particular challenge to graph-based tools such as minigraph¹². We have thus performed separate genotyping of all VNTR regions using vamos during revision stage, thereby improving how this variant class can now be explored by VNTR experts using our data resource (see **Comment #1.1**).

As also discussed in detail in our response to **Comment #1.1**, we justify the utility of our approach by emphasising its goals: to enable graph-aware SV discovery, allow for seamless variant integration, and support population-wide SV genotyping based on long-reads. This enables combining the strengths of high-quality assemblies in a smaller subset of samples with low-cost intermediate coverage sequencing of 1,019 samples as done here – to allow constructing a unified graph containing SVs discovered from both approaches.

Furthermore, while SAGA is designed to collect and integrate rare and low-frequency SV alleles, we emphasise that it is not intended to “replace” diagnostic methods targeting rare diseases, such as those linked to SD-mediated microdeletions. In cases of rare diseases, national health systems (e.g., those in the UK and Germany) have begun reimbursing deep-coverage genome sequencing, and the field is gradually adopting long reads for such diagnostic applications^{33,34}. Thus, we do not anticipate a future need for intermediate-coverage sequencing for rare disease diagnostics. However, we emphasise that our resource addresses an important need for the rare disease community. It allows improved filtering “normal” SV alleles found in 1kGP populations, and as such may significantly enhance rare disease diagnostics. To illustrate this, our revised **Discussion** section reports on how the SVs catalogued in our resource enhance the filtering of normal population variants in four rare disease patient genomes. In brief, initial filtering using multi-platform whole-genome assemblies⁹ left an average of 386 candidate SVs per patient genome. Incorporating our SAGA-based SV resource enabled the reduction of candidate SVs by more than half (54.7%-56.4%), resulting in only 159-187 candidate SVs per patient genome

remaining (**Fig. R22/S62**). This improvement is mainly due to the ability to filter out low allele frequency SV alleles using our resource (**Fig. R23/S63, Fig. R24/S64**).

We also assessed the rate by which validated rare disease-causing SV alleles are filtered out using our SAGA based SV resource, by analysing 31 genomes of rare disease patients previously sequenced with PacBio technology³⁵ harbouring known causal SV alleles (described in the revised **Discussion**). Out of 35 previously validated rare disease-causing SV alleles identified by DELLY in these genomes, only two were filtered based on our resource. One of these two represents a multiallelic short tandem repeat expansion, whereas the other represents a 49 kb deletion near *SHOX* with incomplete disease penetrance^{36,37} – consistent with the occasional presence of SVs at these genomic sites in normal individuals.

These new analyses therefore show effective filtering of SVs from rare disease patients based on our data resource, while causal variants are sensitively retained – illustrating how our study could serve as a basis for the future construction of an open-access long-read panel of normals from diverse ancestries. We hope these additional analyses and the justification given in this response letter clarify the reasoning for the approach taken in our study, its remaining shortcomings – and its strengths.

Figure R19 (Corresponds to Figure S31): Linkage disequilibrium (LD) of SVs (MAF \geq 1%) with nearby single nucleotide polymorphisms (SNPs). The left panel shows all SVs whereas the right panel shows the subset of SVs in Genome in a Bottle high-confident regions of the CHM13 genome (2.3 Gbp, 74.2%).

Figure R20 (Corresponds to Figure S32): Linkage disequilibrium (LD) of SVs (MAF $\geq 1\%$) with nearby single nucleotide polymorphisms (SNPs) as a function of the distance of the SV to the SNP. The left panel shows all SVs whereas the right panel shows the subset of SVs in Genome in a Bottle high-confident regions of the CHM13 genome (2.3 Gbp, 74.2%).

Figure R21 (Corresponds to Figure S70): VNTR density by GiaB difficult and high-confidence regions. Using the vamous efficient motif set for short (STR) and variable number of tandem repeat (VNTR) annotation, we computed the autosomal density of these motifs for high-confident and difficult genomic regions (based on GiaB genomic masks).

Figure R22 (Corresponds to Figure S62): Comparison between SV callsets from four rare disease patients (Patient A-D in panel a-d, respectively) generated by PAV and Sniffles, the phased VCF panel of HRC_mg_44+966 given as input for genotyping SAGA-based SV resource from our study and the Logsdon *et al.* (HGSVC3) SV callset⁹ generated from multi-platform whole genome high quality assemblies. After filtering intersection, we found a) 160, b) 159, c) 187 and d) 180 SVs exclusive to each rare disease patient genome, respectively.

Figure R23 (Corresponds to Figure S63): Allele frequency distributions (log-scale) shown for SV alleles from our study matching those from rare disease patients A-D. These variants were successfully filtered using our SAGA-based SV resource but were missed in a recent study involving multi-platform whole-genome assembly⁹. Consistent with current multi-platform whole genome assembly efforts targeting smaller sample sizes⁹, As shown here, these SV alleles are mostly rare in the population. SAGA's VCF filters out mostly rare SVs.

Figure R24 (Corresponds to Figure S64): Allele frequency distributions (log-scale) shown for SV alleles from our study matching those from rare disease patients A-D, here shown for SVs filtered out both by SAGA and the HGVC multi-platform assembly based dataset⁹. The majority of these sites represent common SVs.

Reviewer #1.12: In general, many of the insights about SV distribution and formation mechanisms do not strike me as particularly novel, and I think the manuscript could be substantially shortened. There are some exceptions to this, for instance the locus-specific transduction tendencies is quite interesting. However, the majority of the insights regarding SV formation, distribution, and mechanisms are not novel, and have even been reported in very similar fashion in previous works by these same authors (e.g. 5' vs 3' transductions figure is identical to that shown in Ebert 2021). Some care should be taken to highlight what is genuinely new.

Response: We have revised the paper and have included additional key references where adequate to allow us to emphasise novel findings, while minimising emphasis on confirmatory findings. We have further expanded our analysis of MEIs, including MEI transductions, and we have ensured that primary display items in this context do not duplicate visualisations shown in Ebert *et al.* 2021⁷. In particular, we have updated **Figure 4** (included below as **Fig. R25**) by removing panels a and b and including additional panels to highlight novel findings pertaining to the germline 5'-transduction activity of the 8q21.11 source element (see also responds to **Comments #3.5** and **#3.6** below). In revising our manuscript, we have incorporated new analyses

while maintaining brevity wherever feasible. At the editor's discretion, we are open to further shortening our manuscript before publication.

Figure R25 (Corresponds to Figure 4): Polymorphic landscape of L1 and SVA transductions. a-b) Contribution relative to the total number of transductions (5', 3' and orphan) for the 20 most active L1 and SVA loci. Source elements are annotated by their presence/absence on the reference, genomic orientation and subfamily designation. Source element novelty, hot activity profiles, *in vitro* activity^{38,39}, and transduction traces in human populations^{7,40} and cancer genomes^{41,42} are shown as heatmaps. While the source L1 at 8q21.11 was not previously reported in large scale studies, a somatic transduction from this element was reported in the human brain⁴³. Transduction 5' and 3'-bias: * $P_{adj} < 0.05$; ** $P_{adj} < 0.005$. **c)** Circos plot displaying the integration positions for the 22 instances of 5' transductions mediated by the 8q21.11 element. **d)** Alignment of inserts containing 5' transductions at the L1 sequence, including a single somatic transduction event reported by Evrony *et al.*⁴³, at 8q21.11. Inserts coloured according to whether they align in forward (black) or reverse (blue) orientation. Splicing event between the full-length L1 and an upstream exon from the long non-coding RNA, *ENSG00000253784*, leading to the formation of 5' transductions, highlighted in yellow. **e)** Zoom into the highlighted region showing that the 5' transductions are generated due to transcription initiation at a strong promoter located upstream, and subsequent canonical splicing between the first and second exon of *ENSG00000253784*, in addition to a second acceptor splice site residing within the L1 body. Transcription initiation is supported by the presence of an annotated transcription start site (hg_93584.1) and total CAGE read counts⁴⁴.

Reviewer #1.13: The dataset should include extensive methylation information. Were there any interesting methylation results?

Response: The 1kGP sample resource is based on EBV-transformed lymphoblastoid cell lines, and the transformation process as such, and EBV-transformation in particular, have been shown to affect the DNA methylation profiles of genes⁴⁵. The use of primary tissue samples, which were unfortunately not available to us for this cohort, would therefore be highly preferred for such an analysis. We previously examined the effects of SVs on nearby methylation signals in primary brain cancer tissues, and while some large complex DNA rearrangements were associated with methylation differences the overall effect of SVs on DNA methylation appeared to be small in this primary tissue⁴⁵.

Nonetheless, to address this reviewer comment, we explored DNA methylation signals in a subset of 66 samples from our 1kGP ONT resource. We employed the latest version of Dorado (v0.7.2; <https://github.com/nanoporetech/dorado>), focusing on the previously generated haplotype calls from LociTyper⁴⁶ for 294 clinically important genes. This analysis revealed well known imprinted genes such as *HG19* and *NLRP2*⁴⁷, which we notably find to rank first and forth for the amount of haplotype-specific methylation (HDM) observed. We also determine strong HDM signals at the *HLA* locus (see also our response to **Comment #2.11**). These results suggest the principle utility of data from our long-read resource for DNA methylation analyses. They also mirror other reports that have described patterns of haplotype-specific methylation at imprinted loci and genes in EBV-transformed cells, rather than reporting strong HDM signals around SVs⁴⁹. A comprehensive genome-wide DNA methylation analysis across all 1,019 samples could yield insights into common haplotype structures linked to methylation in such EBV-transformed cell lines. While we see potential utility of our resource for such future research, such enormous undertaking would be clearly outside of the scope of this current study.

Reviewer #1.14: “Furthermore, analyses of SV breakpoint junctions suggest a continuum of homology-mediated rearrangement processes are integral to SV formation, and highlight evidence for SV recurrence involving repeat sequences.” I find this statement problematic. What does “continuum” mean? It seems to imply that there are many mechanisms at work – that is well known. At the very least the authors should simply say this – as it’s written it’s quite vague. Furthermore, the evidence for SV recurrence is very limited and focused on just a single example.

Response: Our revised manuscript clarifies that the analysis of SV-flanking homology lengths does not show any clear evidence of inflection points (**Fig. 5bc**). This absence suggests a lack of thresholds in breakpoint homology lengths, which might otherwise indicate dominance by a specific minimal efficient processing segment (MEPS), as proposed in models of non-allelic homologous recombination (NAHR)⁵⁰ (**Fig. 5bc**). Our data therefore suggests that homology-mediated SV formation processes operate across a spectrum, rather than reflecting distinctive peaks as would be expected from a MEPS for homologous recombination^{51,52}. While the predominance of SVs involving repeats at their flanks underscores that the repeat architecture plays an important role in SV formation^{7,13,29,50,53–55}, our data therefore support a model in which various homology-mediated mechanisms (including NAHR and others) contribute to SV formation in the human germline.

Additionally, we highlight that our analysis of deletions and duplications, polarized by ancestral state during the paper revision process (refer to **Fig. R17/S75** in our response to **Comment #1.10**), aligns with our earlier observation of the absence of distinct inflection points, thereby supporting our conclusion regarding the lack of evidence for a MEPS. Lastly, with respect to the

comment on SV recurrence we emphasise that our revised manuscript includes an expanded analysis of recurring SVs (see response immediately below).

Reviewer #1.15: The analyses of recurrence needs to be extended and clarified, or alternatively downplayed in the abstract as it's just a single example. I think more could be done by the authors to find additional examples if they wish to highlight these interesting cases.

Response: In response to this reviewer comment, we have expanded our analysis of recurrent SV formation mediated by flanking sequence homologies. We broadened our search parameters to include deletions with flanking homology lengths ranging from 200 to 9,000 bp, in contrast to the previous focus on lengths between 300 and 400 bp. The upper bound of 9,000 bp was chosen to systematically assess the role of the most abundant repeat elements in the human genome, including small to intermediate sized SDs as well as mobile elements.

Apart from changing the range of flanking homology length considered, we maintain the same selection criteria as in our original analysis and now identify a set of 42 candidate recurrent deletions for further investigation. Out of these, we selected six loci with the most robust evidence for recurrence (see the new **Fig. S54-S59**). All these six correspond to transposable element-mediated rearrangement³¹ (TEMR) SVs, likely caused by homology-directed repair (HDR) or NAHR utilising the flanking retrotransposon sequence. One case involves L2/LINE sequence, while for the remaining five Alu elements flank the recurrent SV. These expanded analyses suggest that recurrent TEMRs are actively contributing to SV polymorphisms within human populations.

We have revised the abstract of our manuscript. Furthermore, in the **Methods** section we have clarified remaining challenges associated with identifying cases of SV recurrence – including the need for population-scale SV haplotyping resolution across a wide variety of haplotype backgrounds, precise mapping of SV breakpoints particularly in repeat-rich genomic areas, integration with phased SNP data, as well as population genetic methods to estimate recombination rates in structurally variable regions.

Reviewer #1.16: “In the immediate future, the use of whole genome assembly in isolation is hence deemed less likely to significantly contribute to the advancement of precision medicine by facilitating the generation of comprehensive genetic variant data at a population-scale. “ I don't think this is true – however, regardless the authors need to justify this statement.

Response: In the revised manuscript, we emphasize the complementary strengths of multi-platform whole-genome assembly and the cost-efficient intermediate-coverage approach from our study, and thus offer a more robust justification (see e.g. response to **Comment #1.1**).

Reviewer #1.17: Much is made in the manuscript about data availability and this being an “open-access resource.” However, many of the links to code and pipelines do not work because the repositories are still private.

Response: We have ensured that all the code, pipelines and data listed under “Data Availability” and “Code Availability” are now fully accessible.

Referee #1 - Remarks on Code Availability

Reviewer #1.18: I was able to look at some of the code, but much of it was in private github repositories and needs to be made public.

Response: As stated above (**Comment #1.17**), all the code, pipelines and data listed under “Data Availability” and “Code Availability” are now fully accessible.

Referee #2 - General Remarks

Referee expertise: long-read sequencing

In their manuscript “Long-read sequencing and structural variant characterization in 1,019 samples from the 1000 Genomes Project” Schloissnig, Pani et al. provide intermediate coverage resource of 1,019 long-read ONT genomes from 26 human populations from the 1000 Genomes Project.

In this tremendous effort the authors integrated linear and graph-based approaches for SV analysis via pangenome graph-augmentation and provide: Firstly, a very valuable resource of data and methods; secondly, insights into SV mechanisms/molecular origins and biology; thirdly, novel population SV patterns. The massive SV catalogue with basepair-resolved SV breakpoints represents a very significant increase for 1kGP samples (and in general), with particular insights into difficult SV types, such as VNTR-SVs, MEI transductions, and inversions.

Overall this is an intriguing study, and a well written manuscript with great value to the genomics community.

The hopefully constructive feedback points below may in part further improve the manuscript and resource, but in part may even exceed beyond the scope of this work.

Response: We much appreciate the positive feedback on the quality of our study and the utility of the resource we provide. Below, we address each point raised by this reviewer.

Referee #2 - Major Points of Attention

Reviewer #2.1: Missing comparison to conventional long-read SV methods:

The augmented long-read graph is shown to outperform a) its non-augmented counterpart, and b) previous 1kGP studies. Regarding b), previous studies are based on short reads, so the performance improvement is encouraging but not too surprising. A fairer comparison, in my opinion, would be to compare the graph-based approach to a best-practices approach with conventional long-read sequencing. Are all variants that were ‘fed’ into the graph for augmentation (Step 2/3 of SAGA) later recovered in the SV calling? Does the graph-based approach ‘out-of-the-box’ surpass e.g. specialised (complex) SV, MEI, VNTR, callers?

Response: To address this reviewer comment, we conducted additional analyses comparing our SAGA framework based on graph-augmentation to several conventional long-read SV callers and best-practice approaches – including the long read version of DELLY (calling deletions and insertions), SVision (calling complex SVs), PALMER2/L1ME-AID (specialised tools for MEI calling and annotation, respectively) coupled with multi-platform whole genome assembly, and vamos (for calling VNTRs). Below, we present the key results from these analyses, which are (mainly) highlighted in our revised paper supplement. Notably, the SVision analysis is discussed herein but is excluded from the revised manuscript due to the very low number of SV calls generated per sample from this tool.

- (1) We find that single-sample SV discovery, exemplified here via the use of DELLY and SVision is considerably more susceptible to coverage dependent allele loss than using our SAGA framework, underscoring how the data integration enabled by SAGA makes SV calling more resilient to coverage fluctuations (see **Fig. R8/S83** shown in our response to **Comment #1.3** above, as well as **Table R2** below). For SVision in particular, only a very limited number of calls are being made at the coverage levels of this study: For samples

with coverage between 14x and 18x, less than 20 calls per sample are made (when removing calls in satellite DNA, as suggested by the SVision authors). Furthermore, there is still marked growth in the SVision callset even between 30x and 60x coverage, indicating that within our study SVision based callsets are fairly strongly affected by coverage. Irrespective of this, we do see evidence for an increased overlap for SVision calls with HGSVC3 calls compared to SAGA calls, suggesting advantages for the use of multi-platform assemblies in recovering this small subset of complex SVs. We therefore emphasise the expected advantages of multi-platform assemblies to defining complex SVs in our revised Discussion.

- (2) When compared to MEIs called with state-of-the-art software applied to “best practice” multi-platform whole-genome assemblies analysed by PALMER2/L1ME-AID,⁹ we estimate a sensitivity of 90.6% for Alu insertions, 85.2% for L1s and 84.3% for SVAs – suggesting that ability to detect MEIs in a sample in our study is largely comparable to current multi-platform whole-genome assemblies (see **Fig. R10/S16, R11/S17, R12/S18**, and our response to **Comment #1.5**). Therefore, the larger cohort from our study coupled with the cost-effectiveness of our approach provide a clear advantage for characterising MEIs segregating in the global population down to low allele frequencies when compared to multi-platform whole genome assembly studies pursued in much fewer samples – allowing our study to provide important insights into the biology of MEIs (see e.g. our revised **Fig. 4**).
- (3) For VNTRs, the ONT data is in principle well-suited to characterise the different alleles based on the number of repeat copies. Our graph augmentation pipeline, however is designed to incorporate only SVs (i.e., differences between alleles of 50 bp or larger), which leads to an underrepresentation of VNTRs allelic diversity, as VNTR polymorphisms frequently involve repeat sizes <50 bp. We now acknowledge this limitation in the revised manuscript, and have added a specific analysis of VNTR diversity using the vamos caller¹ – providing a population resource of VNTR diversity to encourage future use of our ONT genomic resource by VNTR specialists (see also **Fig. R5/S39** and response to **Comment #1.1**).

Moreover, we address the question whether variants that were ‘fed’ into the graph for augmentation (Step 2/3 of SAGA) are part of our genotyped SV release. SAGA introduces newly discovered SVs into the graph by using pseudohaplotypes and the minigraph toolkit¹². This process does not yield a one-to-one mapping of SVs that were fed into the graph and the SVs that are retrieved after genotyping and phasing. To answer the reviewer’s question, we therefore compared alignments of pseudohaplotypes and HPRC assemblies to the resulting graph and in this way identified SVs exclusive to the pseudohaplotypes. Of the 189,142 pseudohaplotype-exclusive SV alleles given to Giggles as input, we successfully genotyped and phased (using Shapelt5) 83,965 alleles, which corresponds to a 44.4% recovery of novel SV alleles (**Fig. R26**). The drop in SV alleles can be attributed to two factors: (1) The genotyping algorithm relies on sequence-to-graph alignments which we pursue using minigraph. As the graph becomes more complex due to added pseudohaplotypes, minigraph’s minimizer-based approach struggles with alignments spanning complete bubbles (see also our response to **Comment #1.1**). (2) Newly discovered SV alleles can contribute to the complexity of already existing bubbles, which makes genotyping more challenging as Giggles now may need to distinguish between more alleles, for instance at highly multi-allelic VNTR sites. Therefore, an increased number of SV alleles do not pass the final filtering criteria after genotyping, and this is especially pronounced for multi-allelic and complex SVs (i.e., SVs that are neither clean insertions nor deletions).

In summary, the SAGA pipeline is comparatively stringent when adding new alleles to the graph and only retains those that can be reliably genotyped. We believe that this is indeed currently the correct approach in order to provide a high-quality resource to the community. With improved tools (specifically for sequence-to-graph alignment), we anticipate that in the future we will be able to recover an increased number of alleles from our data set. We have added **Note S2** describing these observations to the Supplementary Material. Furthermore, to allow us to more systematically capture multi-allelic VNTR complexity, we additionally genotyped VNTR sites across our resource using *vamos* (see response to **Comment #1.1**).

Figure R26: This Figure shows the number of SV sites exclusively present in the pseudohaplotypes of the panel given to Giggles for genotyping (labelled as 'in_panel') and the SV sites successfully genotyped by Giggles and subsequently phased by Shapelt5 (labelled as 'in_callset').

Sample	Coverage	#SAGA All	#HGSVC3 All	#SVision CSVs	#SVision overlap SAGA All	#SVision overlap HGSVC3 All
HG00731	14.46	18,818	22,175	17	4	13
NA19331	14.55	23,724	29,155	15	6	11
NA19347	16.22	23,621	28,859	16	4	12
NA12878	16.75	18,610	-	18	5	-
NA19238	18.87	23,694	29,551	32	16	28
HG02554	19.88	23,219	28,238	19	8	17
NA19129	22.56	23,736	28,885	32	16	27
HG00513	26.97	19,029	23,015	39	14	32
HG02953	34.43	23,685	29,009	58	32	50
HG00268	61.16	18,240	21,783	110	48	91
Average	23.6	21,638	26,741	35.6	15.3	31.2

Table R2: Comparing the SAGA framework to conventional long-read SV callers – conventional complex SV calling with SVision. We ran 10 samples using SVision with respect to T2T reference genome and compared the complex SV predictions (CSVs) against our SAGA based release callset and HGSVC3 callset using bedtools reciprocal overlap. We filtered SVision calls without the “PASS” filter as well as those overlapping with centromeres/satellites. Some of the SVision calls contain multiple CSVs at the same loci, and to make the comparison fair, we counted them as a single variant. Notes: a) multiple CSVs in the same loci are counted as 1 CSV b) variants intersecting satellites are removed from the SVision call-set c) used bedtools with any reciprocal overlap d) added 1 kb to each side of the variant breakpoint for SAGA and HGSVC3 variants for comparison.

Reviewer #2.2: Understandably, the authors focus on the impressive successes of the model, but I would be equally interested to learn more about the approaches’ limitations. For example, the authors note a lower sensitivity in large deletions, and the overall moderate size of SVs (Fig 2b, 3a,b) seems to support this trend. Also the SV-Mb sum per individual seems lower than expected from other studies. It makes perfect sense that SVs of 100’s of kb - especially the structurally more diverse ones – may not yet be captured by the graph, especially very rare events. I would value a little extra discussion of this (presumed?) limitation. I do absolutely not expect the authors to solve these problems, but I would appreciate it if they could probe the ‘loose ends’ of the graph’s capabilities more, to help guide the field towards the next challenges; as these may be important for the suggested disease studies.

Response: We do appreciate the reviewers’ interest in discussing the limitations of our study more extensively. To address this, we performed an array of additional analyses that enable a much more thorough evaluation of the current limitations of SAGA, including revealing current limitations to graph-based analyses as a whole. Relevant details are in our responses to **Comments #1.1, #1.2, #1.11, and #2.1** and not necessarily repeated here to ensure brevity. This includes a focus on the challenges associated with capturing variation in repeat-rich regions, especially multi-allelic VNTRs, using our graph based approach. Large deletions are more likely to intersect repeat-rich regions, which pose a particular challenge for SV interpretation in pangenome graphs¹². At the same time, short reads are optimally suited to identify large deletions via depth-of-coverage

analysis even in relation to long reads as noted previously in the literature²⁵⁻²⁷. Our revised manuscript now reports on these aspects including the existing limitations of SAGA at different places in the text including the Discussion. Our Discussion section also states that SAGA is expected to have reduced discovery power for extremely rare SVs, such as with an allele count of 1, while at the same time introducing challenges in ensuring sequence consensus quality for rare alleles in the graph.

Lastly, we note that during revision we realised that we previously reported a too small SV lengths sum per sample, as we inadvertently omitted the "COMPLEX SV" category in the cumulative length calculations. Our revised cumulative SV lengths (SV-Mb sum per individual) range from 4.78 to 9.88 Mb of sequence variation per genome (median: 7.35 Mb) – which is a marked increase from the originally reported values (range: 3.44-7.45 Mb; median: 5.48 Mb). We thank the reviewer for alerting us to this issue.

Reviewer #2.3: In line with the previous point, a comparison for SV completeness of a subset of samples would be beneficial:

- On average 23969 SVs for AFR samples (19,297 for non-AFR) seems lower than previous studies suggested. Is it undercalling here or overcalling of previous work
- For a subset of samples orthogonal data (platinum genome pedigree, HG002) should be available for such comparison.
- Can the authors down sample coverage data, to estimate the effect of 5x, 10x, 15x, 20x coverage for some of the samples with higher coverage?

Response: We have addressed this comment through several additional analyses. In particular, we conducted systematic comparisons of our callset to recently published orthogonal data, that is, multi-platform genome assemblies generated for an overlapping sample subset⁹. These analyses are described in detail as part of our responses to **Comments #1.4, #1.11, #2.1, and #2.2** above.

In brief, the ability to detect biallelic SVs using SAGA, exemplified by mobile element deletions and insertions, shows performance comparable to a recent study utilising highest quality multi-platform whole-genome assemblies⁹ (see **Comment #1.4**). For VNTRs, the ONT data is in principle well-suited to characterise the different alleles based on the number of repeat copies. However, our graph augmentation pipeline incorporates only SVs (i.e., differences between alleles of 50 bp or larger), which leads to an underrepresentation of VNTR diversity, as VNTR polymorphisms frequently involve repeat sizes <50 bp. We now acknowledge this limitation, and emphasise the need for the community to develop improved tools for graph based SV analyses in the future. As alluded to above, we have added a systematic analysis of VNTRs with a specialised genotyping tool (vamos; **Comment #1.1**).

Additionally, we have carried out comprehensive analyses of the effect of ONT sequencing coverage on our callset, which can be found in our response to **Comments #1.3** above. In brief, we first stratified our samples by coverage, into four coverage bins: <10×; ≥10× and <20×; ≥20× and <30×; ≥30× and <40×. We next evaluated SV genotype qualities using the Giggles tool. Stratification by coverage (**Fig. R6/S81**, above) did not noticeably impact SV genotyping qualities. However, coverage did somewhat affect the number of SVs called: samples with <10× coverage showed a slight reduction, while SAGA consistently identified similar numbers of SVs across coverage levels ≥10× (**Fig. R8/S83**, above). Since 91% of our samples have ≥10× ONT sequencing coverage in our resource (median ~16.6×), and since our analyses suggest saturation at this coverage level, we conclude that sequencing coverage is appropriate in most samples. By comparison, we observe a considerably more pronounced coverage dependence when performing

single sample SV discovery with the DELLY caller in its “long-read mode”. This underscores the effect of SV integration using SAGA, which makes our callset more robust to coverage fluctuations (**Fig. R8/S83**, above).

Reviewer #2.4: As add on to paragraph page 6 (and Tables 4-6): With the availability of trio data in 1kGP have the authors systematically attempted *de novo* SV mutation rates per SV class? This may be important in the connect of long-read trios for rare disease studies. Or does use of cell line material hamper this capability. If so: how many of the SVs may be cell line artifacts, and is there an indication for somatic origin for certain SVs/SV classes? Or are other variant types e.g. VNTR-SVs somatically instable?

Response: Unfortunately, the use of cell lines makes it infeasible to reliably distinguish *de novo* germline variants from post-zygotic somatic mosaicism present in tissues, as well as from “*in vitro*” cell line artefacts. We note that parallel to our effort, there is a recent comprehensive study⁵⁶ on SV mutation rates using a genome assembly approach that spans multiple generations in a family pedigree⁵⁶. In that study, Porubsky *et al.* utilise DNA from a primary tissue (blood). As a key take home message, their study describes that germline and post-zygotic somatic mosaicisms are both transmitted to the offspring⁵⁶. This underscores the importance of separating out germline variants from post-zygotic somatic mosaicism as well as cell line artefacts ° which necessitates access to primary donor tissue samples that were unavailable to our study.

Therefore, the analysis of germline *de novo* SVs is outside the scope of this study. Nevertheless, we emphasise that our revised manuscript now outlines the principal utility of our SAGA-based SV resource for the rare disease community by showing how it enables prioritising candidate disease-causing *de novo* SVs (see our response to **Comment #1.11**).

Reviewer #2.5: In which form or ideally database will the data be available to the scientific community?

Response: The data are openly available through different platforms to facilitate broad community access and computational utility. These include the International Genome Sample Resource (IGSR) database, the main repository for 1kGP studies. To meet the needs of researchers working with large datasets, we have made the data freely accessible also via Amazon Web Services (AWS; see `s3://1kg-ont-vienna`). This will allow users to run large-scale computational analyses directly within the AWS ecosystem, ensuring efficient processing without the overhead of managing this large dataset locally. Additionally, we are in the process of sharing the data through the European Nucleotide Archive (ENA) to further maximize accessibility and reach within the scientific community.

Reviewer #2.6: SVAN is quite central to the downstream analysis, but the Github page is missing (<https://github.com/REPPIO/SVAN>) so it was not possible to evaluate.

Response: We apologise for this and have ensured that all the code, pipelines and data listed under “Data Availability” and “Code Availability” are now fully accessible – including for the SVAN framework, which is available at <https://github.com/REPPIO-LAB/SVAN>.

Referee #2 - Minor Points of Attention

Minor point of attention – Reviewer Comment #2.7: Page 14: overall sum of SV numbers and sum of SV sizes per individual would be beneficial to add.

Response: We agree, and have consolidated the respective data into the new **Figure R27/S61** (also shown below). This new figure illustrates the number and cumulative length of all SVs, as well as of relevant SV subsets. As stated above in our response to **Comment #2.2**, while generating this figure, we noticed that the "COMPLEX SV" category had previously been omitted from the cumulative SV length reported. This oversight has been corrected in the revised manuscript.

Figure R27 (Corresponds to Figure S61): SV count (a) and cumulative length (b) per sample for all SVs and different SV subsets. From the callset, different SV subsets were extracted and the number of SV as

well as the cumulative SV length per sample was determined. HET: heterozygous SV, HOM: homozygous SV, DEL: deletion, INS: insertion.

Minor point of attention – Reviewer Comment #2.8: Relation to similar/potentially overlapping work, as shown in recent preprint: Gustafson *et al.*
<https://www.medrxiv.org/content/10.1101/2024.03.05.24303792v1>

Response: Our study and the work by Gustafson *et al.*, which involved sequencing a smaller cohort at greater depth, were initiated separately – yet, we have coordinated our initial open-access data releases with Gustafson *et al.*, in line with the guiding principles of the 1000 Genomes Project with regards to data sharing. It is our understanding that our sequencing project was launched prior to theirs, and as a result, our project is currently significantly more advanced in terms of data generation, algorithmic development and SV characterisation. While Gustafson *et al.* have sequenced approximately 100 genomes thus far, our study encompasses 1,019 genomes. Moreover, Gustafson *et al.* have to our understanding not yet released SV genotypes for their 100-sample cohort. Given the less mature nature of their SV dataset, we have refrained from pursuing comparative analyses to Gustafson *et al.* and instead performed comprehensive comparisons to current, state-of-the-art multi-platform whole genome assemblies⁹. These comparisons support the high quality of our 1kGP ONT data based SV data resource (see further above, including our response to **Comment #1.5**).

We emphasise that like Gustafson *et al.*, we share the ultimate goal of resequencing the entire 1kGP cohort using long-read technology and are committed to collaborating on future endeavors. To this end, we have recently exchanged letters of support affirming this shared commitment and have cited Gustafson *et al.* in the forward-looking section of our Discussion.

Minor point of attention – Reviewer Comment #2.9: Comparison to Beyter *et al* Nat Genet (2021): <https://www.nature.com/articles/s41588-021-00865-4>. The authors identify 22,636 SVs by applying multiple SV tools. Which variant types do/do not overlap?

Response: While such a comparison would undoubtedly have been highly valuable, these data are unfortunately not publicly accessible. Beyter *et al.* state that access to these data is restricted—permitted only within the premises of deCODE genetics in Iceland and regulated by Icelandic law, which prohibits data reuse outside Iceland. These access restrictions would preclude any meaningful analysis or reproducibility of results. In terms of overlapping variant types: Beyter *et al.*, in their study, detected a median of 22,636 SVs per individual, focusing their analysis on deletions and multiallelic tandem repeat structures in relation to phenotypes of Icelandic donors registered in the (restricted) deCODE genetics data resource.

Minor point of attention – Reviewer Comment #2.10: Previously implied strengths of ONT not yet utilised in this dataset and not yet captured in this manuscript:

- Short tandem repeat expansions as a separate variant class

Response: Short tandem repeats (STRs) are commonly defined as genetic variants composed of repeat units ranging from 2 to 5 bp¹, with some classifications extending this definition to also include repeat units of 6 and 7 bp (https://en.wikipedia.org/wiki/STR_analysis). STRs primarily encompass genetic variants that fall within the size range of short insertions and deletions

(InDels). These variants are beyond the scope of this study, which focuses on SVs defined as sequence polymorphisms 50 bp or longer^{7,13,14,20,57,58}. We have revised the Discussion section to highlight the value of long-read sequencing in future comprehensive studies of STRs. We note further that during paper revision we expanded our analysis of VNTRs by employing the *vamos* toolkit (see e.g. our response to **Comments #1.1**).

Minor point of attention – Reviewer Comment #2.11: DNA modifications, such as DNA methylation. Do some SV classes lead to methylation alteration in cis?

Response: We thank the reviewer for this question. We emphasize that the 1kGP sample resource is based on EBV-transformed lymphoblastoid cell lines, and that the transformation process as such (and EBV transformation in particular) can affect the DNA methylation profiles of genes⁴⁵. More suitable for such analysis would therefore be the use of primary tissue samples, which were not available for our sample set unfortunately. We note that we did previously examine the effects of SVs on nearby methylation signals in primary brain cancer tissues, and while some large complex DNA rearrangements were associated with methylation differences the overall effect of SVs on DNA methylation appeared to be small in this primary tissue⁴⁵.

To address this reviewer comment, we conducted an analysis in a representative subset of 66 samples from our 1kGP ONT resource using the latest version of Dorado (v0.7.2; <https://github.com/nanoporetech/dorado>), resulting in modification likelihoods at base pair resolution for 5mC genome wide. Next we made use of the Haplotype calls from LociTyper⁴⁶ to link individual reads to haplotypes at 294 clinically important genes. Across all CpG sites within these genes we observe a median of 80% methylation (min 73%; max 87%) across all samples (**Fig. R28/S66A**). With the reads containing methylation calls linked to haplotypes we performed within sample haplotype-specific differential methylation tests, which has the benefit of controlling any sample level confounding factors by providing internal sample level control. To model haplotype-specific differential methylation patterns (HDM) we applied a Mann-Whitney test between the distribution of modification ratios for haplotype 1 and haplotype 2 at every gene for each of the samples independently (**Methods**). Every sample had at least one gene with significant HDM and most genes had at least one significant sample, suggesting that this test is sensitive towards local haplotype-specific imbalances in DNA methylation. We found 41 genes with significant HDM in 10 or more samples (**Fig. R28/S66B**) including well known imprinted genes such as *HG19* and *NLRP2*⁴⁷, which rank first and forth for the amount of HDM observed respectively. Encouragingly we also see high levels of HDM at the *HLA* locus and in particular at and nearby the *HLA-DRB5* locus (**Fig. R28/S66C**). *HLA-DRB5* has been reported by several previous studies on HDM, with HDM at this locus suggested to contribute to the risk for conditions such as rheumatoid arthritis⁴⁸. We note, however, that the gene content differs by *HLA-DR* haplogroup and only haplogroup DR2 carries the *HLA-DRB5* gene, warranting a haplogroup-aware analysis in future research.

These analyses suggest the principle utility of data from our long-read resource for methylation analyses and mirror other reports describing patterns of haplotype-specific methylation at imprinted loci and genes in EBV-transformed cells, rather than reporting strong signals at SVs⁴⁹. A comprehensive genome-wide DNA methylation analysis across all 1,019 samples could yield insights into common haplotype structures linked to methylation in EBV-transformed cell lines. While this emphasises the potential utility of our resource for future research, such enormous undertaking would be clearly outside of the scope of this current study.

Figure R28 (Corresponds to Figure S66): Investigation of haplotype-specific DNA methylation signals at 294 clinically important genes, called from ONT long-reads in a representative subset of the 1kGP samples. **A)** The percentage of all CpG sites across the 294 genes that are methylated per sample. **B)** Barplot showing the number of samples per gene for genes with 10 or more samples showing significant haplotype-specific differential methylation signal. **C)** Manhattan plot showing the $-\log_{10}(p)$ from a haplotype-specific differential methylation test for the 41 genes that have 10 or more significant samples.

Minor point of attention – Reviewer Comment #2.12: Previous work implied that SVs mediated by DNA-repair mechanisms, may lead to increased number of *de novo* SNVs near SV breakpoints. Have the authors explored this for respective SV classes?

Response: We appreciate the reviewer’s comment on the potential for increased *de novo* SNVs near SV breakpoints due to DNA-repair mechanisms (i.e., ‘joint mutagenesis’), which this reviewer categorised as a ‘minor comment’. We also note that alternative hypotheses exist, such as the notion that integrating an SV into a haplotype could lead to a sustained increase in local mutability at the affected loci (i.e., ‘regionally increased mutability’)^{29,59}. Addressing these hypotheses thoroughly would require re-basecalling the entire ONT dataset across 1,019 samples for precise *de novo* SNV discovery, followed by population-scale graph-based variant analysis with consideration of SNPs, and ultimately, detailed phylogenetic analyses to distinguish between the hypothesis of joint mutation and regionally increased mutability. Given the resource-oriented focus of our study and the comprehensive analyses already presented including new data provided during revision—as summarised on the first page of this response letter—we regard this analysis to be beyond the scope of this study.

Minor point of attention – Reviewer Comment #2.13: In this dataset an average of 93.6% of CHM13 are covered at least 5-fold. It may be interesting for the reader to understand which parts are consistently lacking?

Response: We agree that identifying regions with consistently low coverage is important for providing deeper context to our data interpretation. To address this reviewer comment, we analysed regions in the CHM13 assembly that show coverage below 5-fold throughout our dataset (now presented in the manuscript as **Fig. R29/S79**; Figure also pasted below). These low-coverage regions correspond to some of the most complex genomic areas, notably the alpha satellite Higher Order Repeat (HOR) arrays within the centromeres (regions that we masked (filtered out) in our analysis), satellite-dense regions adjacent to HOR arrays on chromosomes 1, 9, and 16, and the acrocentric arms of chromosomes 13, 14, 15, 21, 22, and Y—collectively covering approximately 58% of these low-coverage areas. A fraction of the remaining low-coverage regions are associated with segmental duplications (8.4%) and other areas annotated by repeatmasker (18.7%).

Figure R29 (Corresponds to Figure S79): Number of samples contributing at least 5x coverage by genomic region (mean across 100 kb bins). Primarily, these low-coverage areas correspond to the most complex regions of the genome, mainly peri/centromeric regions such as: (i) the alpha satellite (α Sat) Higher Order Repeat (HOR) array (highlighted in red), (ii) the acrocentric arms of chromosomes 13, 14, 15, 21, 22, and Y, and (iii) the satellite-rich regions downstream of the HOR arrays on chromosomes 1, 9, and 16. These three regions jointly account for 58.2% of these lower coverage regions. Most of the remaining low-coverage regions are associated with segmental duplications (8.4%) and other areas annotated by repeatmasker (18.7%).

Minor point of attention – Reviewer Comment #2.14: Did the authors compare their SV/CNV estimates to the recent paraphase work for 160 loci with paralogous genes, see e.g. Chen et al. <https://www.biorxiv.org/content/10.1101/2024.04.19.590294v2>

Response: Our understanding is that paraphase is a bioinformatics tool optimised for Pacific Biosciences (HiFi) sequencing data, which has been developed to resolve highly similar genes by phasing all haplotypes within a gene family. The study describing Paraphase⁶⁰ was published as a preprint while our study was under review. Unfortunately, so far no Paraphase-based callsets have been made available for 1kGP samples. Chen *et al.* restricted their analysis to loci previously regarded as challenging to genotype. For these, we noticed that SNV and SV genomic positions are not systematically disclosed in the Chen *et al.* preprint, and it appears that Chen *et al.* have restricted access to their callset to mitigate the risk of donor re-identification – thereby hindering meaningful and reproducible analyses. While the lack of positional data on variants in their study prevents a direct comparison, we note that our conclusions are generally in agreement: long-read sequencing can genotype these challenging to genotype regions, revealing substantial genetic variation affecting paralogous sequences. While not within the scope of this current work, we expect that future application of Paraphase to 1kGP samples sequenced with PacBio HiFi technology will allow for a more comprehensive algorithmic comparison.

Minor point of attention – Reviewer Comment #2.15: Figures:

- Fig2: Inset in panel g may be a separate panel
- Some supplementary figures/graphs lack labelling of axis e.g. [bp]

Response: We thank the reviewer for suggesting improvements to our figures. The inset from **Fig. 2g** is now available as a separate panel and we introduced axis labeling in various supplementary figures.

Referee #2 - Remarks on Code Availability

Reviewer #2.16: I am not a bioinformatician, hence did not test the code provided. But recommend systematic checks by other reviewers/editors.

As recommended above: SVAN is quite central to the downstream analysis, but the Github page is missing (<https://github.com/REPPIO/SVAN>) so it was not possible to evaluate.

Response: We have ensured that all the code, pipelines and data listed under “Data Availability” and “Code Availability” are now fully accessible – including SVAN, which is available at: <https://github.com/REPPIO-LAB/SVAN>

Referee #3 - General Remarks

Referee expertise: TEs

This study conducts Oxford Nanopore Technologies (ONT) long-read structural variant (SV) calling and haplotyping on the 1000 Genomes Project (1kGP) sample collection. The 1kGP cohort is relatively large and represents geographically diverse populations. The analysis is impressive in scale and quality. I appreciate and respect the amount of work done here.

Response: We thank the reviewer for their thorough evaluation of our manuscript and for emphasising the scale and quality of our analysis. We appreciate the recognition of the extensive work conducted, and below responded to all specific reviewer comments in detail.

Referee #3 - Specific Comments

Reviewer #3.1: On the other hand, the advance in terms of concepts or biology is more difficult to see, at least with the way the manuscript is currently presented. Population-scale analyses of SVs with long-read sequencing have previously been published (refs 5, 20, 82). The study uncovers 167,291 SVs but does not say how many of these SVs are absent from existing databases or publications.

Response: We appreciate the feedback regarding the importance of comparing our findings to prior studies (e.g., “ref. 5,” “ref. 20,” and “ref. 82” from our original manuscript). To facilitate a meaningful analysis, we compared our SV calls to publicly available datasets from long-read studies, including Liao *et al.* (2023)⁸ (“ref. 20”), Ebert *et al.* (2021)⁷ (“ref. 5”), and Logsdon *et al.* (2024)⁹ (a recent preprint). Additionally, we analyzed our data against the Genome Aggregation Database (GnomAD), which spans data from 730,947 exome sequences and 76,215 whole-genome sequences from unrelated individuals in its most recent release (<https://gnomad.broadinstitute.org/>). However, we were unable to include comparisons with Beyter *et al.* (“ref. 82”) due to their stringent access restrictions, which limit data usage to the premises of deCODE genetics under Icelandic law, as detailed in **Minor Comment #2.9**.

- Comparison of the original Human Pangenome Reference Consortium (HPRC) graph-based SV callset⁸ with our expanded resource highlights the anticipated growth in low-frequency alleles in our dataset. This is evident in the relationship between SV allele counts and SV site counts, as shown in **Fig. 2c**, following genotyping of both the original HPRC graph and the augmented graph using Giggles.
- Cohort sizes of long-read based multi-platform assembly studies targeting overlapping 1kGP samples have shown a relatively slow (yet steady) growth in recent years – from 35 genomes in Ebert *et al.* (2021)⁷, and 47 genomes in Liao *et al.* (2023)⁸, to 65 genomes in Logsdon *et al.* (2024)⁹ (which includes higher-coverage upgrades of several samples from Ebert *et al.*, 2021). In our revisions, we focused on comparing our SV callset to Logsdon *et al.* (2024) yielding insights into specificity and sensitivity. This revealed FDRs from 0.85%–6.7% for different classes of biallelic insertions (see **Comment #1.4**). For biallelic deletions, we estimate an FDR of 1.94% by comparison. When considering all deletion sites, which contain numerous multiallelic tandem repeat events, graph-based analysis yields FDR estimates dependent on the SV length: For deletions ≥ 250 bp, the estimated

FDR is 10.2%, whereas for smaller deletions that predominantly consist of short tandem repeats (**Fig. R13/S19**) the estimated FDR is 37.7%. This supports the notion that polymorphic tandem repeats present an inherent challenge for SV interpretation, in particular in pan-genome graphs¹² (where the comparison process itself is likely to be affected) but also as highly multiallelic sites in general. This observation therefore led us to pursue comprehensive genotyping of percentile ranges of repeat units for all VNTRs in all 1,019 samples using *vamos* (see **Comment #1.1**) – thereby increasing the utility of our data resource to VNTR experts.

- Finally, we anticipate that our resource will show utility for research on the phenotypic implications of SVs, including in underrepresented ancestries for which our data provide a catalog of normal variation. In this regard, a comparison with gnomAD, a repository of variation with population allele frequencies, reveals that a large proportion (50.86%) of the insertions in our resource are novel. These include 8,077 Alu (34.75%), 2,586 L1 (52.89%), and 1,269 SVA elements (47.12%) not reported in gnomAD. For deletions, 14.5% are novel, reflecting the strengths of short reads in deletion detection^{13,14,20}.

These new analyses have now been incorporated into the manuscript. We also made every effort to clarify advances in terms of concepts and biology, which includes substantial revisions to the Discussion. As alluded to on page 1 of this response letter, we further expanded several analyses presented in this manuscript and included new analyses – including by expanding the analysis of recurrent deletion events driven by NAHR involving mobile element sequences at the flanks, the more detailed analysis of VNTRs, and additional systematic analyses of MEIs (alluded to further below).

Reviewer #3.2: Long-read genomic analyses of mobile element insertions (MEIs) have been published before (e.g. PMID: 31853540, PMID: 33186547, PMID: 37823611) and their findings are recapitulated here, although these prior reports are not cited and should be. There are 28,358 non-reference MEIs mentioned, but no table (amongst 20 in the supplement) listing them? How many are completely new to this study? If the dataset is viewed more as a resource than as a breakthrough in terms of biology, in its current state it lacks sufficient information to be accessible to readers or future studies to realise its full value.

Response: We thank the reviewer for their suggestions for improving the manuscript, and for enhancing the accessibility and utility of our MEI dataset as a resource. We acknowledge the importance of prior studies that have used long-reads to analyse MEIs, and now include several references^{7,29,61,62} when describing canonical MEI events in the main text. To enhance the accessibility of our dataset as a resource, we have included a supplementary table (**Table S18**) listing all identified MEIs, including information on their genomic coordinates, classification, retrotransposition hallmarks, and conformation. Furthermore, as described in much detail in our responses to **Comments #1.4** and **#3.1**, we have compared our SV resource, including the MEI, with gnomAD to assess novelty, as well as with a recent study using genome assemblies⁸ to evaluate both FDR and sensitivity.

Reviewer #3.3: My main area of expertise is MEIs and I have focused on that part of the manuscript in my comments below.

- How many of the 167,291 SVs and 28,358 MEIs have been reported elsewhere? Here I mean in large-scale population datasets OR smaller datasets using long-reads specifically to look for SVs or MEIs. Please clarify as well whether 1 or more "spanning" reads were required to fully cover MEIs and their two breakpoints in order to be called.

Response: We appreciate the reviewer's inquiry regarding the overlap between the SVs and MEIs identified in our study and those reported in previous datasets, as well as the clarification request regarding the criteria for calling MEIs. As stated above, during paper revision we conducted comprehensive comparisons of our SV resource, including the MEIs, with gnomAD to assess novelty, as well as with a recent study using multi-platform whole genome assemblies⁸ to verify that both FDR and sensitivity of our callset are acceptable. We have also performed more extensive novelty assessments revealing that a considerable fraction – 8,077 Alu (34.75%), 2,586 L1 (52.89%), and 1,269 SVA elements (47.12%) – are novel in relation to numerous prior studies (see our response to **Comment #3.1**).

Furthermore, as explained in the response to **Comment #3.9**, SVAN classifies the MEI into canonical and non-canonical based on the presence/absence of the hallmarks of retrotransposition, including target site duplications and poly(A) tails. We applied all three SV calling approaches (DELLY, Sniffles, SVarp) with standard parameter settings, implying that at least 2 "spanning" reads are required to fully cover MEIs and their two breakpoints in order to be called. We emphasise that using high-quality human assemblies⁸, we estimate an FDR of 3.9% for canonical *Alu* insertions, 6.7% for canonical L1 insertions, as well as 0.85% FDR for canonical SVA element insertions, thus demonstrating very high SV callset quality (see details in our response to **Comment #1.4**).

Reviewer #3.4: Several studies (a selection noted above) have used long-read sequencing to examine MEIs, and their findings inform and largely agree with what is found here. That should be stated plainly. It is notable that these works studied L1 and/or SVA transductions, finding for example the differing 5'/3' ratios for L1s and SVAs. I would suggest referring to the earlier works when considering the presentation of this section. Similarly, retrotransposon associated inversions have been considered in detail elsewhere with long-read analysis (PMID: 36402840).

Response: We thank the reviewer for this comment, and have revised our manuscript to explicitly reference and discuss relevant prior studies exploring MEI transductions using long-reads. In particular, we cited^{7,29,61,62} when describing canonical MEI events, including those that are 5'-inverted, in the main text. Furthermore, as described in the response to the **Comment #3.5** below, we performed a comprehensive comparison between the source elements reported in our resource and prior studies.

Reviewer #3.5: Of the source elements noted on page 12, how many have been reported before? From looking at Table S16, it looks to me like the vast majority have been, which is a good sign in terms of reproducibility but lacking in terms of continuity and annotation quality. This information should be noted in the text and in the table.

Response: In response to this comment, we have conducted a thorough review of previously reported source elements (**Tables S19** and **S20**). We have compared these elements with previous large-scale studies that catalogued source elements and evaluated their activity using

transduction tracing in the germline^{7,40}, somatic transduction detection in cancer genomes^{41,42}, and *in vitro* retrotransposition assays^{38,39}.

Upon review, we notably find that only 24% (50/208) of the source L1s and 17% (30/176) of the source SVAs identified in our study have previously been reported to be active in these aforementioned studies. These percentages, however, increase to 80% and 45% when subsetting to the 20 most active L1s and SVAs in our study, respectively. This suggests a high degree of consistency between the top ranking source L1s in our dataset and published catalogues, with two of them, Xp22.2 (1st ranking) and 14q23.1 (8th ranking), previously reported to have strombolian patterns of hot activity in a pan-cancer study⁴¹. On the other hand, we also demonstrate a substantial portion of novel source L1 and source SVAs, compared to these prior comprehensive studies, enabled by the access to low-allele frequency insertion variants in our large long-read resource spanning 1,019 human genomes.

The high degree of novel highly active SVA elements identified in this study probably reflects the fact that this retroelement family is understudied in comparison with L1, with a few published studies on the germline⁷, and none in cancer nor *in vitro*. Nonetheless, the most active source SVA in our resource has been described to be active via germline transduction tracing by Ebert *et al.*⁷. In addition, in order to improve the annotation quality we have also determined whether source elements are present both in GRCh38 and CHM13 references.

We have included our new analyses on source element novelty in the main text, in addition to producing a new version of the main **Figure 4** (included in the response to **Comment #1.12** as **Fig. R25**), which includes activity measurements from previous studies as heatmaps. This information has also been incorporated in the **Tables S19** and **S20**.

Reviewer #3.6: The 8q21.11 source L1 noted as only having 5' transduction events is one we showed does the same thing in the embryonic brain (PMID: 31230816, see Fig. 6) because there's a strong upstream promoter next to it (check the transductions are spliced please, if they are it is the same mechanism). It's very cool to see that this source element 5' transduces in the germline - that could be noted in the main text. In the same study we found the chr13q12.3 source L1 jumps readily in brain and carries 3' transductions. Also see PMID: 34772701 and other papers from the Devine lab taking the time to annotate source elements). Bottom line: a lot of work has gone into the source elements mentioned here by numerous other groups before now and those annotations should be exploited and used.

Response: We are grateful to the reviewer for highlighting this important context. We have undertaken the following revisions to expand and strengthen our manuscript:

1. We have re-examined the 5' transductions from the 8q21.11 source L1 element in our dataset to determine whether these events involve splicing. Our analysis confirms that the transductions are indeed aberrantly spliced, which supports the notion that the same upstream promoter mechanism observed in embryonic brain tissues reported in^{43,63} is also driving these 5' transduction events in the germline. We have included this new finding in the main text and incorporated three panels (c-e) describing the 5' transduction activity for the 8q21.11 source locus in **Figure 4** (included in the response to **Comment #1.12** as **R25**). We now also discuss the sharing of this interesting biological mechanism across different tissues (Discussion section).

2. We fully acknowledge the substantial body of work that has been done by the Devine lab and others in cataloguing active source elements. As described in detail in the response to **Comment #3.5**, we have cross-compared the detected source loci in our study with prior studies and incorporated this information both in the relevant section in the main text and in the updated version of **Fig. 4** (i.e., the heatmaps shown in panels a and b). We have also highlighted in the Discussion the specific cases of 8q21.11 and chr13q12.3 source L1 elements, which display somatic activity in the brain according to prior research^{43,63} kindly highlighted by this reviewer.

Reviewer #3.7: The authors really should cite these two earlier papers (PMID: 10066175, PMID: 7920631 - the latter is the first report of a germline L1 transduction, although they call it a "USC") from the Kazazian lab when talking about L1 transductions.

Response: We thank the reviewer for pointing this out, and have included citations to these seminal papers at the opening sentence of the transduction section.

Reviewer #3.8: Along these lines, please check the references for the sentence "Consistent with prior studies^{5,9,73}, we find that the relative proportion of 3' or 5' transduction events differs considerably between both MEI classes" because reference 73 I doubt says anything about transductions.

Response: We have gone once again through all references in the MEI section, and have incorporated additional relevant references to ensure completeness. For this specific sentence, we have replaced reference 73 by a reference to Damert et al. (2009)⁶⁴, which reports the occurrence of frequent 5' transductions for SVA elements.

Reviewer #3.9: The text says that 96.4% of MEIs show retrotransposition hallmarks. Please provide a table of these MEIs, their genomic coordinates, and the hallmarks (TSD sequence, polyA sequence, integration site EN motif). What is the median TSD size? The cutoff for a polyA to be called (10 bp) is very short considering these tend to be much longer than 10 bp on average. A 1 bp TSD cutoff is also very short. If more stringent TSD sequence feature cutoffs are applied, is the figure still anywhere close to 96.4%? Along those lines, it is a concern that so many of the MEIs (1,115 events) are 3' truncated. This generally won't happen via TPRT ... please show these in the table. Providing this information will make the data more useful to others.

Response: We appreciate the detailed feedback and suggestions for improving the rigour and utility of the MEI resource and analysis. In response, we have undertaken the following actions:

- (1) Our revised manuscript includes a table (**Table S18**) that lists all identified MEIs along with their associated retrotransposition hallmarks, including the sequence and length for the TSD and poly(A) tails, the strand where each insertion occurs, and the description of the conformation for each insertion event. SVA insertions include four additional fields reporting the sequence and length for their internal Hexamer and VNTR repeats.
- (2) Furthermore, we have implemented a new module in our annotation tool, SVAN, to provide detailed information regarding the conformation for each insertion. This is encoded as a string that describes the alignment of the MEI sequence over the corresponding Alu, L1 or SVA consensus sequence. As an example, a conformation of "TRUN+REV+DUP+FOR+POLY+TD+POLY" for an L1 insertion indicates that the L1 has been truncated on the 5' (TRUN), with the remaining 3' piece being composed by two L1 fragments, in reverse (REV) and forward (FOR) orientation, respectively. Both fragments overlap, creating a

duplication (DUP) at their junction. These features are consistent with an integration via the twin priming model²⁸. The L1 insert includes a companion 3' transduced sequence (TD), which is flanked by two poly(A) tails (POLYA).

- (3) Based on SVAN conformation annotations, we have classified the Alu, L1 and SVA insertions as canonical (84.3%; 29,592/35,115) and non-canonical (15.7%; 5,523). Canonical events display a conformation consistent with target-primed reverse transcription or twin priming (only for L1), while non-canonical events include MEIs that are 3' truncated (34.4%; 1,901/5,523), that lack poly(A) tails (66.4%; 3,665/5,523) or have internal rearrangements or potential template switches (52.8%; 2,916/5,523). Non-canonical insertions are potentially the consequence of either subsequent SVs occurring after integration or alternative mechanism of integration⁶⁵⁻⁶⁷. We have expanded the "Mobile elements" subsection in the main text to provide a characterization of canonical and non-canonical events and included four new supplementary figures (**Fig. R30/S41, R31/S42, R32/S43, R33/S44**). In response to **Comment #1.5**, we have also assessed the sensitivity and false discovery rate both for canonical and no-canonical MEIs.
- (4) We acknowledge the reviewer's concern regarding the short cutoffs for TSD (1 bp) and poly(A) tail (10 bp). After increasing the minimum TSD size cutoff to 4 bp, the median TSD is 21 bp, with 98.7% (29,202/29,592) of the canonical and 97.4% (5,378/5,523) of the non-canonical insertion having reported TSDs. The median poly(A) size is 30 bp using a cutoff of 10 bp, with all the canonical and 33.6% (1,858/5,523) of the non-canonical insertion having detected poly(A) tails. Increasing the minimum poly(A) tail length to 20 bp increases the number of canonical and non-canonical events without poly(A) tails to 3,663 and 4,327, respectively. Nonetheless, given the high degree of consistency for canonical events between our MEI callset and orthogonal approaches (response to **Comment #1.5**), we have decided to maintain a poly(A) cutoff of 10 bp to avoid losing sensitivity, given that poly(A) stretches can degenerate and get shortened after the integration. We have included the median values for TSD and poly(A) lengths in the relevant section in the main text, in addition to display their size distributions, as well as for the hexamers and VNTR repeats within SVAs, for each conformation within the new four supplementary figures (**Fig. R30/S41, R31/S42, R32/S43, R33/S44**).

Figure R30 (Corresponds to Figure S41): a) Number of insertions (left) and deletions (right) for each canonical Alu conformation. Conformation components are FOR: forward alignment to consensus sequence, TRUN: Truncation, POLYA. **b)** Length distributions of insertion, PolyA, Target Site Duplication (TSD). Colored by insertion mechanism: TPRT: Target-Primed Reverse Transcription. **c)** Proportion of insertions/deletions of each conformation that overlap each repeat family annotation in the reference. **d)** Bottom: Stacked dot-plots summarizing the alignments of each insertion/deletion against the Alu consensus sequence of the two conformations. Up: histogram of alignment coverage.

Figure R31 (Corresponds to Figure S42): a) Number of insertions (left) and deletions (right) for each canonical L1 conformation. Conformation components are FOR: forward alignment to consensus sequence, REV: reverse alignment, TRUN: Truncation, DEL: deletion, DUP: duplication, TD: transduction, BLUNT: blunt-end inversion breakpoint, POLYA: polyA. **b)** Length distributions of insertion, PolyA, Target Site Duplication (TSD), 5' transduction, and 3' transduction. Colored by insertion mechanism: TPRT: Target-Primed Reverse Transcription. **c)** Proportion of insertions/deletions of each conformation that overlap each repeat family annotation in the reference. **d)** Bottom: Stacked dot-plots summarizing the alignments of each insertion/deletion against the L1 consensus sequence of the four most common conformations (left to right). Up: histogram of alignment coverage.

Figure R32 (Corresponds to Figure S43): a) Number of insertions (left) and deletions (right) for each canonical SVA conformations. Conformation components are Hexamer, Alu-like, VNTR, SINE-R, MAST2, TD: transduction, and POLYA. **b)** Length distributions of insertion, PolyA, Target Site Duplication (TSD), 5' transduction, and 3' transduction. **c)** Proportion of insertions/deletions of each conformation that overlap each repeat family annotation in the reference. **d)** Length distributions of 5'Hexamer (left), and VNTR (right) by conformation. Colored by insertion mechanism: TPRT: Target-Primed Reverse Transcription.

Figure R33 (Corresponds to Figure S44): a) Number of insertions (left) and deletions (right) for each Processed Pseudogene conformation. Conformation components are FOR: forward alignment to source gene, REV: reverse alignment, POLYA: polyA. Colored by insertion mechanism: TPRT: Target-Primed Reverse Transcription. **b)** Distributions of insertion length, PolyA length, Target Site Duplication (TSD) length, and number of exons (left to right). **c)** Proportion of insertions/deletions of each conformation with breakpoints that overlap each repeat family annotation in the reference. **d)** GenomeBrowser screenshot of alignment of two full-length Processed Pseudogenes to the reference genome.

Reviewer #3.10: There are a handful of HERVK insertions reported. These are unexpected. Do they have the correct sized TSDs? Did the authors do a manual inspection of these in the IGV to check they look right? It seems (from Fig 3a) that the HERVKs aren't proviruses or LTRs ... are they just random bits of HERVK? Have they been RT'd by an L1? (TSD length informative here).

Response: We acknowledge the importance of manually inspecting the non-reference insertions involving endogenous retroviral elements, as these are typically rarely found. Using SVAN, we detect a total of 38 such events, which includes 10 insertions classified as HERVK and 28 solo-LTR. The size distribution for the HERVK insertions ranges from 1,453 bp to 25,485 bp, while the solo-LTR size distribution is consistent with the expected size for a LTR5_Hs (median of 974 bp), with the exception of 6 outlier insertions. Following this reviewer's feedback, we further aligned all the inserts using the HERV-K113 complete genome as reference (RefSeq: NC_022518), confirming that 82% (23/28) of solo-LTR correspond to complete LTR5_Hs events, aligning either to the 5' or 3' flanking LTR repeat on the reference. For HERVK, 5 of the insertions encompass the proviral sequence flanked by two LTRs, 4 have a single LTR with the second residing on the human reference (CHM13) and one is heavily truncated on its 5'.

In response to **Comment #3.2** regarding SV site novelty, we further cross compared the 38 insertions involving endogenous retroviral elements detected in our study with those reported by either Wildschutte *et al.* (2016)⁶⁸ or Ewing *et al.* (2020)⁶², in which the authors identified 9 solo-LTR insertions by long-read sequencing of five tissues from two individuals. Notably, while only ~22% (5/23) of the solo-LTR in our study were not previously reported, the majority (70%; 7/10) of HERVK insertions are novel. This underscores the increased sensitivity we achieve with long-reads allowing us to detect and resolve large insertions involving repetitive elements. All these new analyses have been summarized into a new supplementary figure (**Fig. R34/S45**) and highlighted in the main text.

Figure R34 (Corresponds to Figure S45): Length distribution of full-HERVK elements (a) and solo-LTR (c). Number of full-HERVK elements (b) and solo-LTR (d) with insertion breakpoints that overlap each repeat family in the reference. e) Alignments of each full-HERVK (top) and solo-LTR (bottom) to the HERVK-K113 reference genome. ID for each element and whether it has been reported as a non-reference HERV in Wildschutte *et al.* 2016 or Ewing *et al.* 2020 is shown (left). f) UCSC GenomeBrowser screenshots showing examples of two full-length HERVK insertions: Full element is not in reference (up), and one LTR is in the reference (bottom). Yellow bar highlights the insertion site, and zoomed out view highlights HERVK conformation.

References

1. Ren, J., Gu, B. & Chaisson, M. J. P. Vamos: Variable-number tandem repeats annotation using efficient motif sets. *Genome Biol.* **24**, 175 (2023).
2. De Coster, W. *et al.* Visualization and analysis of medically relevant tandem repeats in nanopore sequencing of control cohorts with pathSTR. *Genome Res.* gr.279265.124 (2024).
3. Lansdon, L. A. *et al.* Successful classification of clinical pediatric leukemia genetic subtypes via structural variant detection using HiFi long-read sequencing. *medRxiv* 2024.11.05.24316078 (2024) doi:10.1101/2024.11.05.24316078.
4. Zheng, X. *et al.* STIX: Long-reads based accurate structural variation annotation at population scale. *bioRxiv* 2024.09.30.615931 (2024) doi:10.1101/2024.09.30.615931.
5. Noyvert, B. *et al.* Imputation of structural variants using a multi-ancestry long-read sequencing panel enables identification of disease associations. *medRxiv* 2023.12.20.23300308 (2023) doi:10.1101/2023.12.20.23300308.
6. Reeve, M. P. *et al.* Loss of CFHR5 function reduces the risk for age-related macular degeneration. *medRxiv* 2024.11.11.24317117 (2024) doi:10.1101/2024.11.11.24317117.
7. Ebert, P. *et al.* Haplotype-resolved diverse human genomes and integrated analysis of structural variation. *Science* (2021) doi:10.1126/science.abf7117.
8. Liao, W.-W. *et al.* A draft human pangenome reference. *Nature* **617**, 312–324 (2023).
9. Logsdon, G. A. *et al.* Complex genetic variation in nearly complete human genomes. *bioRxivorg* 2024.09.24.614721 (2024).
10. Beyter, D. *et al.* Long-read sequencing of 3,622 Icelanders provides insight into the role of structural variants in human diseases and other traits. *Nat. Genet.* **53**, 779–786 (2021).
11. Gong, J. *et al.* Long-read sequencing of 945 Han individuals identifies novel structural variants associated with phenotypic diversity and disease susceptibility. *medRxiv* 2024.03.21.24304654 (2024) doi:10.1101/2024.03.21.24304654.
12. Li, H., Feng, X. & Chu, C. The design and construction of reference pangenome graphs with minigraph. *Genome Biol.* **21**, 265 (2020).
13. Sudmant, P. H. *et al.* An integrated map of structural variation in 2,504 human genomes.

- Nature* **526**, 75–81 (2015).
14. Byrska-Bishop, M. *et al.* High-coverage whole-genome sequencing of the expanded 1000 Genomes Project cohort including 602 trios. *Cell* **185**, 3426–3440.e19 (2022).
 15. Hickey, G. *et al.* Pangenome graph construction from genome alignments with Minigraph-Cactus. *Nat. Biotechnol.* (2023) doi:10.1038/s41587-023-01793-w.
 16. Garrison, E. *et al.* Building pangenome graphs. *Nat. Methods* **21**, 2008–2012 (2024).
 17. Halldorsson, B. V. *et al.* The sequences of 150,119 genomes in the UK Biobank. *Nature* **607**, 732–740 (2022).
 18. De Roeck, A. *et al.* An intronic VNTR affects splicing of ABCA7 and increases risk of Alzheimer’s disease. *Acta Neuropathol.* **135**, 827–837 (2018).
 19. Ruggieri, A. *et al.* Multiomic elucidation of a coding 99-mer repeat-expansion skeletal muscle disease. *Acta Neuropathol.* **140**, 231–235 (2020).
 20. Mills, R. E. *et al.* Mapping copy number variation by population-scale genome sequencing. *Nature* **470**, 59–65 (2011).
 21. 1000 Genomes Project Consortium *et al.* An integrated map of genetic variation from 1,092 human genomes. *Nature* **491**, 56–65 (2012).
 22. 1000 Genomes Project Consortium *et al.* A global reference for human genetic variation. *Nature* **526**, 68–74 (2015).
 23. Handsaker, R. E. *et al.* Large multiallelic copy number variations in humans. *Nat. Genet.* **47**, 296–303 (2015).
 24. Benson, G. Tandem repeats finder: a program to analyze DNA sequences. *Nucleic Acids Res.* **27**, 573–580 (1999).
 25. Chaisson, M. J. P. *et al.* Multi-platform discovery of haplotype-resolved structural variation in human genomes. *Nat. Commun.* **10**, 1784 (2019).
 26. Zhao, X. *et al.* Expectations and blind spots for structural variation detection from long-read assemblies and short-read genome sequencing technologies. *Am. J. Hum. Genet.* **108**, 919–928 (2021).
 27. Choo, Z.-N. *et al.* Most large structural variants in cancer genomes can be detected without long reads. *Nat. Genet.* **55**, 2139–2148 (2023).

28. Ostertag, E. M. & Kazazian, H. H., Jr. Twin priming: a proposed mechanism for the creation of inversions in L1 retrotransposition. *Genome Res.* **11**, 2059–2065 (2001).
29. Porubsky, D. *et al.* Recurrent inversion polymorphisms in humans associate with genetic instability and genomic disorders. *Cell* (2022) doi:10.1016/j.cell.2022.04.017.
30. Yoo, D. *et al.* Complete sequencing of ape genomes. *Genomics* (2024).
31. Balachandran, P. *et al.* Transposable element-mediated rearrangements are prevalent in human genomes. *Nat. Commun.* **13**, 7115 (2022).
32. Dwarshuis, N. *et al.* The GIAB genomic stratifications resource for human reference genomes. *Nat. Commun.* **15**, 9029 (2024).
33. Steyaert, W. *et al.* Unravelling undiagnosed rare disease cases by HiFi long-read genome sequencing. *medRxiv* 2024.05.03.24305331 (2024).
34. Jensen, T. D. *et al.* Integration of transcriptomics and long-read genomics prioritizes structural variants in rare disease. *medRxiv* 2024.03.22.24304565 (2024).
35. Höps, W. *et al.* HiFi long-read genomes for difficult-to-detect clinically relevant variants. *Genetic and Genomic Medicine* (2024).
36. Benito-Sanz, S. *et al.* Identification of the first recurrent PAR1 deletion in Léri-Weill dyschondrosteosis and idiopathic short stature reveals the presence of a novel SHOX enhancer. *J. Med. Genet.* **49**, 442–450 (2012).
37. Bunyan, D. J., Baker, K. R., Harvey, J. F. & Thomas, N. S. Diagnostic screening identifies a wide range of mutations involving the SHOX gene, including a common 47.5 kb deletion 160 kb downstream with a variable phenotypic effect. *Am. J. Med. Genet. A* **161A**, 1329–1338 (2013).
38. Brouha, B. *et al.* Hot L1s account for the bulk of retrotransposition in the human population. *Proc. Natl. Acad. Sci. U. S. A.* **100**, 5280–5285 (2003).
39. Beck, C. R. *et al.* LINE-1 retrotransposition activity in human genomes. *Cell* **141**, 1159–1170 (2010).
40. Gardner, E. J. *et al.* The Mobile Element Locator Tool (MELT): population-scale mobile element discovery and biology. *Genome Res.* **27**, 1916–1929 (2017).
41. Rodriguez-Martin, B. *et al.* Pan-cancer analysis of whole genomes identifies driver

rearrangements promoted by LINE-1 retrotransposition. *Nat. Genet.* (2020)

doi:10.1038/s41588-019-0562-0.

42. Tubio, J. M. C. *et al.* Mobile DNA in cancer. Extensive transduction of nonrepetitive DNA mediated by L1 retrotransposition in cancer genomes. *Science* **345**, 1251343 (2014).
43. Evrony, G. D. *et al.* Single-neuron sequencing analysis of L1 retrotransposition and somatic mutation in the human brain. *Cell* **151**, 483–496 (2012).
44. Noguchi, S. *et al.* FANTOM5 CAGE profiles of human and mouse samples. *Sci. Data* **4**, 170112 (2017).
45. Caliskan, M., Cusanovich, D. A., Ober, C. & Gilad, Y. The effects of EBV transformation on gene expression levels and methylation profiles. *Hum. Mol. Genet.* **20**, 1643–1652 (2011).
46. Prodanov, T. *et al.* Locityper: targeted genotyping of complex polymorphic genes. *Bioinformatics* (2024).
47. Begemann, M. *et al.* Maternal variants in NLRP and other maternal effect proteins are associated with multilocus imprinting disturbance in offspring. *J. Med. Genet.* **55**, 497–504 (2018).
48. Xu, J. *et al.* Epigenome-wide methylation haplotype association analysis identified HLA-DRB1, HLA-DRB5 and HLA-DQB1 as risk factors for rheumatoid arthritis. *Int. J. Immunogenet.* **50**, 291–298 (2023).
49. Gustafson, J. A. *et al.* High-coverage nanopore sequencing of samples from the 1000 Genomes Project to build a comprehensive catalog of human genetic variation. *Genome Res.* **34**, 2061–2073 (2024).
50. Gu, W., Zhang, F. & Lupski, J. R. Mechanisms for human genomic rearrangements. *Pathogenetics* **1**, 4 (2008).
51. Waldman, A. S. & Liskay, R. M. Dependence of intrachromosomal recombination in mammalian cells on uninterrupted homology. *Mol. Cell. Biol.* **8**, 5350–5357 (1988).
52. Rubnitz, J. & Subramani, S. The minimum amount of homology required for homologous recombination in mammalian cells. *Mol. Cell. Biol.* **4**, 2253–2258 (1984).
53. Bailey, J. A. & Eichler, E. E. Primate segmental duplications: crucibles of evolution, diversity and disease. *Nat. Rev. Genet.* **7**, 552–564 (2006).

54. Korbelt, J. O. *et al.* Paired-end mapping reveals extensive structural variation in the human genome. *Science* **318**, 420–426 (2007).
55. Conrad, D. F. *et al.* Mutation spectrum revealed by breakpoint sequencing of human germline CNVs. *Nat. Genet.* **42**, 385–391 (2010).
56. Porubsky, D. *et al.* A familial, telomere-to-telomere reference for human de novo mutation and recombination from a four-generation pedigree. *bioRxiv* 2024.08.05.606142 (2024).
57. Abel, H. J. *et al.* Mapping and characterization of structural variation in 17,795 human genomes. *Nature* **583**, 83–89 (2020).
58. Collins, R. L. *et al.* A structural variation reference for medical and population genetics. *Nature* **581**, 444–451 (2020).
59. Dhokarh, D. & Abyzov, A. Elevated variant density around SV breakpoints in germline lineage lends support to error-prone replication hypothesis. *Genome Res.* **26**, 874–881 (2016).
60. Chen, X. *et al.* Genome-wide profiling of highly similar paralogous genes using HiFi sequencing. *bioRxiv* 2024.04.19.590294 (2024) doi:10.1101/2024.04.19.590294.
61. Zhou, W. *et al.* Identification and characterization of occult human-specific LINE-1 insertions using long-read sequencing technology. *Nucleic Acids Res.* **48**, 1146–1163 (2020).
62. Ewing, A. D. *et al.* Nanopore sequencing enables comprehensive transposable element epigenomic profiling. *Mol. Cell* **80**, 915–928.e5 (2020).
63. Sanchez-Luque, F. J. *et al.* LINE-1 Evasion of Epigenetic Repression in Humans. *Mol. Cell* **75**, 590–604.e12 (2019).
64. Damert, A. *et al.* 5'-Transducing SVA retrotransposon groups spread efficiently throughout the human genome. *Genome Res.* **19**, 1992–2008 (2009).
65. Morrish, T. A. *et al.* DNA repair mediated by endonuclease-independent LINE-1 retrotransposition. *Nat. Genet.* **31**, 159–165 (2002).
66. Srikanta, D. *et al.* An alternative pathway for Alu retrotransposition suggests a role in DNA double-strand break repair. *Genomics* **93**, 205–212 (2009).
67. Sen, S. K., Huang, C. T., Han, K. & Batzer, M. A. Endonuclease-independent insertion provides an alternative pathway for L1 retrotransposition in the human genome. *Nucleic Acids Res.* **35**, 3741–3751 (2007).

68. Wildschutte, J. H. *et al.* Discovery of unfixed endogenous retrovirus insertions in diverse human populations. *Proc. Natl. Acad. Sci. U. S. A.* **113**, E2326–34 (2016).

Referee #1 - General Remarks

The authors have performed a great many additional analyses and have reworked the manuscript extensively which I appreciate. While I think the manuscript is substantially strengthened I am still confused by the comparison to genome assemblies. These confusions are to some extent an extension of my original concerns regarding the utility of the dataset overall (R1.1). There does not seem to be a comprehensive analysis of sensitivity and specificity. FDRs are rather confusingly given for different MEI classes and then for “all deletions” (presumably also including the MEIs?) for two different size bins. The question is, what is the True positive rate? Or the false negative rate? This needs to be clearly broken down for different event types shown in Figure 3 (not just MEIs) and importantly considered as a function of allele frequency and size. If there is a singleton SV how likely am I to find it? To be sure, the approach here is very interesting and important, but it seems that given the number of SVs discovered in 1000 individuals vs the number discovered in 10-fold fewer individuals that this is coming at a huge cost to sensitivity. This does not invalidate the interesting results therein to be sure, but is critical to evaluate the utility of this approach for disease studies and highlighted by the manuscript in several places.

Response: Thanks for the encouraging feedback on our revised manuscript. We do agree that a more thorough comparison to genome assemblies is a useful addition to the manuscript. As suggested, we have evaluated the number of true positive, false positive and false negative SV calls as a function of allele frequency and size separately for insertions and deletions for the Giggles genotyped callset and the more stringently filtered final callset (**Fig. S17-S20**) in comparison to high-quality genome assemblies from HGVC3 (Logsdon *et al. BioRxiv*, 2024). A couple of findings are made based on these comparisons:

- First, a more detailed evaluation compared to the HGVC3 assemblies reveals an enrichment of false negative SV sites near telomeres (**Fig. S92**). This can be explained by how our graph augmentation pipeline has been originally designed: This pipeline uses GRCh38 as the backbone sequence for pseudo-haplotype integration. Since telomeres are often not resolved in the GRCh38 reference, these SVs were not integrated into the augmented graph.
- Second, we observe an increased false negative rate for rare SVs (**Fig. S18 and S20**), as may be expected from our study design that uses only intermediate ONT sequencing coverage, and in line with various prior 1000 Genomes Project studies performed using intermediate coverage sequencing (see e.g. 1000 Genomes Project Consortium, *Nature* 2015).
- Third, we observe a slightly higher number of false negatives for insertions compared to deletions. This is likely attributable to the higher ONT error rate and shorter N50 read length compared to high-quality multi-platform assemblies.

- Fourth, we observe more false positive SVs for shorter SVs than for larger SVs. This category of SVs is notably highly enriched for tandem repeats (62.02% for SVs < 250 bp compared to 16.88% for SVs \geq 250 bp). Variations in short tandem repeats (STRs) and variable number of tandem repeats (VNTRs) pose significant challenges for the generation of population SV callsets and their comparison. We believe that in particular, discordances in representation are to be expected when comparing tandem repeat variation based on a pangenome graph representation to a classical assembly-based approach (**Fig. S21**). This can lead to large differences in reported SV size, type and location that are incompletely taken into account by current VCF-based approaches that evaluate SV records one-by-one.

Since SAGA uses minigraph for graph augmentation we also observe a general under-representation of the full allelic spectrum present at these tandem repeat loci, which is since minigraph currently incorporates new alleles only if they differ by at least 50 bp (the effect this may have on representing VNTRs has been noted before by Li *et al. Genome Biol* 2020). As a consequence of this characteristic of minigraph, VNTRs present in a large set of 967 genotyped samples are represented in SAGA with reduced variability (**Fig. S21**), as we had already noted in our prior response letter; this is why we chose to significantly augment our study by using VAMOS as a tool to comprehensively genotype VNTRs across our full cohort.

We emphasise that it remains an open research problem how to represent tandem repeat variations in large population-scale studies – indeed, basepair-level pangenome graphs could become intractable for large cohorts at these loci. At the same time, assembly-based callsets generated via CHM13 (T2T) alignment are likely to face similar challenges, once these approaches are executed at population-scale – i.e. utilised in a similar sample size as in our study. This is because algorithmic over- or under-merging of repeat variations is likely to bias the SV size, location and frequency spectrum present at these highly polymorphic tandem repeat loci.

Finally, in response to the reviewer's request for detailed analyses of false-negative and false-positive rates stratified by SV class, we note that our decision to utilize CHM13/T2T genomic coordinates for our SV resource does constrain the availability of suitable ground truth SV datasets. We emphasise that our approach to focus on previously released SV calls using CHM13/T2T coordinates was intentionally adopted to circumvent liftover procedures, which would otherwise introduce biases and complicate interpretability. In this regard, Logsdon *et al. BioRxiv* 2024 recently provided an annotated mobile element insertion (MEI) callset in the CHM13/T2 coordinate system as part of their HGSVC3 assembly data release. In our recently revised manuscript, we utilized this dataset for an in-depth comparison of specific transposable element classes (Alu, L1, and SVA; see **Figs. S22, S23**). However, Logsdon *et al.* did not specifically annotate other abundant SV classes such as VNTRs, tandem duplications, or non-canonical MEIs – which, unfortunately, prevents similarly detailed comparisons for these event types. We emphasise that going through the process of independently annotating these SV classes in the HGSVC3 assemblies would constitute a potentially circular analysis, and also would extend beyond the scope of our study. Processed pseudogene insertions, NUMTs, and inversion subclasses (including inverted duplications) likewise lack complete discovery set annotation in the

HGSVC3 assemblies; furthermore, these event types occur at fewer than 50 sites per genome, which would impede with a robust comparative analysis such as false negative evaluation stratified by size and/or allele frequency. These limitations in existing ground truth data guided our approach to presenting the newly generated data as shown in **Figs. S17-S20**. These new Supplemental Figures are referred to in the revised “Resource quality assessment” section of our manuscript.

Related to the above, what does this mean: “reflecting altogether 37,834 SVs (43.7%) in our SAGA callset” does this mean that the SAGA callset has 43% of the calls that are detected in the assemblies?

Response: This percentage pertains to the non-tandem repeat associated insertion events, that is, we were giving the percentage of insertions that are (43%) occurring outside of a tandem repeat context. We acknowledge that this part was confusing and we thus do not report this percentage anymore, since we instead have – as recommended by the reviewer – stratified the FDR and TP, FP and FN analysis by SV size and allele frequency. The revised text, from the “Resource quality assessment” section, now reads:

We also compared SVs identified by SAGA to those derived from multi-platform genome assemblies, obtaining a genome-wide FDR of 15.55% for deletions and 15.89% for insertions. These FDRs vary by SV size: SVs ≥ 250 bp show considerably lower FDRs (deletions: 6.91%, insertions: 8.12%) than SVs < 250 bp (deletions: 19.14%, insertions: 19.57%; Fig. S16-S21). The smallest SVs largely comprise tandem repeat variation, whose divergent representation between graph and assembly-based callsets can complicate both SV discovery and comparative analysis, particularly for multiallelic sites. To mitigate this issue, we also pursued genotyping of percentile ranges of repeat units for previously annotated variable number of tandem repeats (VNTRs) in our cohort (see further below). MEIs – an SV class exhibiting well-defined allele architectures – exhibit a particularly low FDR (0.85%–6.75%; Fig. S22), whereas mobile element deletions show 1.94% FDR (Note S3). Moreover, comparison to assemblies yielded autosomal sensitivity estimates, which we find to vary by SV class, allele frequency, and size (Fig. S17-S20, Fig. S23-S24) – ranging from 84.3%–90.6% for MEIs, for example.

Referee #1 - Code Availability

It is open and quite good! they did a lovely job!

Response: We thank the reviewer for their constructive and overall positive comments.

Referee #2 - General Remarks

It was a pleasure to review the revised version of “Long-read sequencing and structural variant characterization in 1,019 samples from the 1000 Genomes Project” by Schloissnig, Pani et al.. The authors have not only addressed the majority of my previous suggestions, or provided logical

argument why it is not possible to address few minor points now. They have also provided a lot of extra analyses, and have further improved their manuscript. There is few very minor remaining suggestions:

Reviewer #2.1: Great the authors updated the SV-Mb sum (previous comment 2.2.). They may consider that their total number of SV per individual and the SV-Mb sum per individual is still significantly lower compared to higher coverage/quality datasets, such as platinum-pedigree (see e.g.: The Platinum Pedigree: A long-read benchmark for genetic variants | bioRxiv), and cite this accordingly in their discussion.

Response: We agree – and have clarified this in the Discussion and inserted the suggested reference.

Reviewer #2.2: Previous comment 2.5: While the full public data sharing is wonderful, I was hoping for a very practical sharing of the SV dataset for non-bioinformaticians. E.g. a bed file with all SVs calls, or an aggregate SV database with allele frequencies within 1kG. this would allow many more scientists to use the SV catalogue to filter their (long-read) genome data to e.g. filter-out common SVs. A UCSC SV track could also be of great added value.

Response: To facilitate practical sharing and browsing of our SV dataset we now provide a VCF comprising the biallelic calls with allele count larger than zero, SVAN annotations and population specific allele frequencies.

Reviewer #2.3: The citation of the newly included dataset by Höps et al. (ref 28) is not accurate. The MedRxiv preprint is available at AJHG, and the same citation in the supplemental note (ref 19) is incorrect.

Response: We have corrected the reference in the revised main text and supplement.

In summary I now fully support publication of the revised manuscript.

Response: We thank the reviewer for their many constructive comments.

Referee #3 - General Remarks

The authors have done a commendable job of addressing my concerns and I have no further criticisms to offer.

Geoff Faulkner (University of Queensland)

Response: Thank you for all the constructive comments provided!

Reply to reviewer 1

Referee #1 (Remarks to the Author):

I appreciate the detailed analyses of true positive, false positive, and false negatives. However, this is largely relegated to the supplement and thus gives the reader an incomplete view of the data. The updated main text should make reference to these numbers as requested. Currently FDR is mentioned, but not TPR. Sensitivity is mentioned, but not specificity. Simply mentioning these numbers in passing is essential.

Response: Thank you for the suggestion. We agree that the graph and assembly-based call set comparison is an important analysis for our manuscript. Therefore, we have decided to highlight the previous Supplemental Figures S17-S20 – showing true positives (TP), false positives (FP) and false negatives (FN) as a function of SV size and minor allele frequency – as an Extended Data Figure (ED Fig. 2a-d). In the main text, we now also provide, in addition to previous FDR and sensitivity estimates, estimates for TPR (64.36% for deletions and 67.33% for insertions). In terms of precision, the positive predictive value (PPV) for deletions is 84.45% and for insertions 84.11%.